# Two distinct host-specialized fungal species cause white-nose disease in bats

Nicola M. Fischer[1,2], Imogen Dumville[2], Benoit Nabholz[2,3], Violeta Zhelyazkova[4], Ruth-Marie Stecker[1], Anna S. Blomberg[5,6], Serena E. Dool[1,7,8], Marcus Fritze[1,9,10], Marie-Ka Tilak[2], Andriy-Taras Bashta[11], Clothilde Chenal[2,12,13], Anna-Sophie Fiston-Lavier[2,3] & Sebastien J. Puechmaille[1,2,3]✉

The emergence of infectious diseases, particularly those caused by fungal pathogens, poses serious threats to public health, wildlife and ecosystem stability[1]. Host–fungus interactions and environmental factors have been extensively examined[2–4]. However, the role of genetic variability in pathogens is often less well-studied, even for diseases such as white-nose in bats, which has caused one of the highest disease-driven death tolls documented in nonhuman mammals[5]. Previous research on white-nose disease has primarily focused on variations in disease outcomes attributed to host traits or environmental conditions[6–8], but has neglected pathogen variability. Here we leverage an extensive reference collection of 5,479 fungal isolates from 27 countries to reveal that the widespread causative agent is not a single species but two sympatric cryptic species, each exhibiting host specialization. Our findings provide evidence of recombination in each species, but significant genetic differentiation across their genomes, including differences in genome organization. Both species contain geographically differentiated populations, which enabled us to identify the species introduced to North America and trace its source population to a region in Ukraine. In light of our discovery of the existence of two cryptic species of the causative agent of white-nose disease, our research underscores the need to integrate the study of pathogen variability into comprehensive disease surveillance, management and prevention strategies. This holistic approach is crucial for enhancing our understanding of diseases and implementing effective measures to prevent their spread.

The emergence of infectious diseases, especially those caused by fungal pathogens, is occurring at an increasing rate[9]. This trend presents substantial and far-reaching threats to public and wildlife health, global food security and the stability of ecosystems worldwide[1]. Among the notorious fungal pathogens is the ascomycete *Pseudogymnoascus destructans*, a bat-specific pathogen responsible for white-nose disease—also commonly referred to as white-nose syndrome[10,11]—which affects hibernating bats[10]. *P. destructans* is native to Eurasia, where it is widely distributed, and has been recently introduced to North America[12,13]. Although the disease occurs across both Eurasia and North America[14–16], mass mortality in bats has been observed solely in North America[15,17]. Previous research has revealed important insights into the interaction between the fungus and environmental conditions and between the fungus and its hosts, with different species of bat showing various levels of susceptibility[18]. However, this body of literature has developed entirely under the postulate that

*P. destructans* is a single clonally expanding isolate in North America and a single fungal species with a broad distribution across the Palaearctic region. Although earlier investigations have documented a genetically diverse population of *P. destructans* in Europe[13,19] and identified three genetically distant isolates from Asia[12], the limited geographical sampling and lack of ecologically relevant data have hindered interpretations of these findings, which have remained overlooked since.

On the basis of an extensive reference collection of 5,479 isolates originating from 264 sites in 27 countries (5,446 from Eurasia and 33 isolates from North America), we use molecular data to demonstrate the presence of two cryptic fungal species that cause the disease and use fine-scale population structure across Europe to identify the probable geographical source population of the North American introduction. We further integrate molecular data with field-based ecological data to investigate differences in ecological niche, including abiotic factors

[1]Zoological Institute and Museum, University of Greifswald, Greifswald, Germany. [2]Institut des Sciences de l'Évolution Montpellier (ISEM), University of Montpellier, CNRS, EPHE, IRD, Montpellier, France. [3]Institut Universitaire de France, Paris, France. [4]National Museum of Natural History, Bulgarian Academy of Sciences, Sofia, Bulgaria. [5]Department of Biology, University of Turku, Turku, Finland. [6]Finnish Museum of Natural History, University of Helsinki, Helsinki, Finland. [7]PHIM Plant Health Institute, University of Montpellier, INRAE, CIRAD, Institut Agro, IRD, Montpellier, France. [8]CIRAD, UMR PHIM, Montpellier, France. [9]German Bat Observatory, Berlin, Germany. [10]Competence Center for Bat Conservation in Saxony Anhalt, South Harz Karst Landscape Biosphere Reserve, Südharz, Germany. [11]Institute of Ecology of the Carpathians, National Academy of Sciences of Ukraine, Lviv, Ukraine. [12]DIADE, University of Montpellier, CIRAD, IRD, Montpellier, France. [13]MIVEGEC, University of Montpellier, CIRAD, IRD, Montpellier, France. ✉e-mail: sebastien.puechmaille@umontpellier.fr

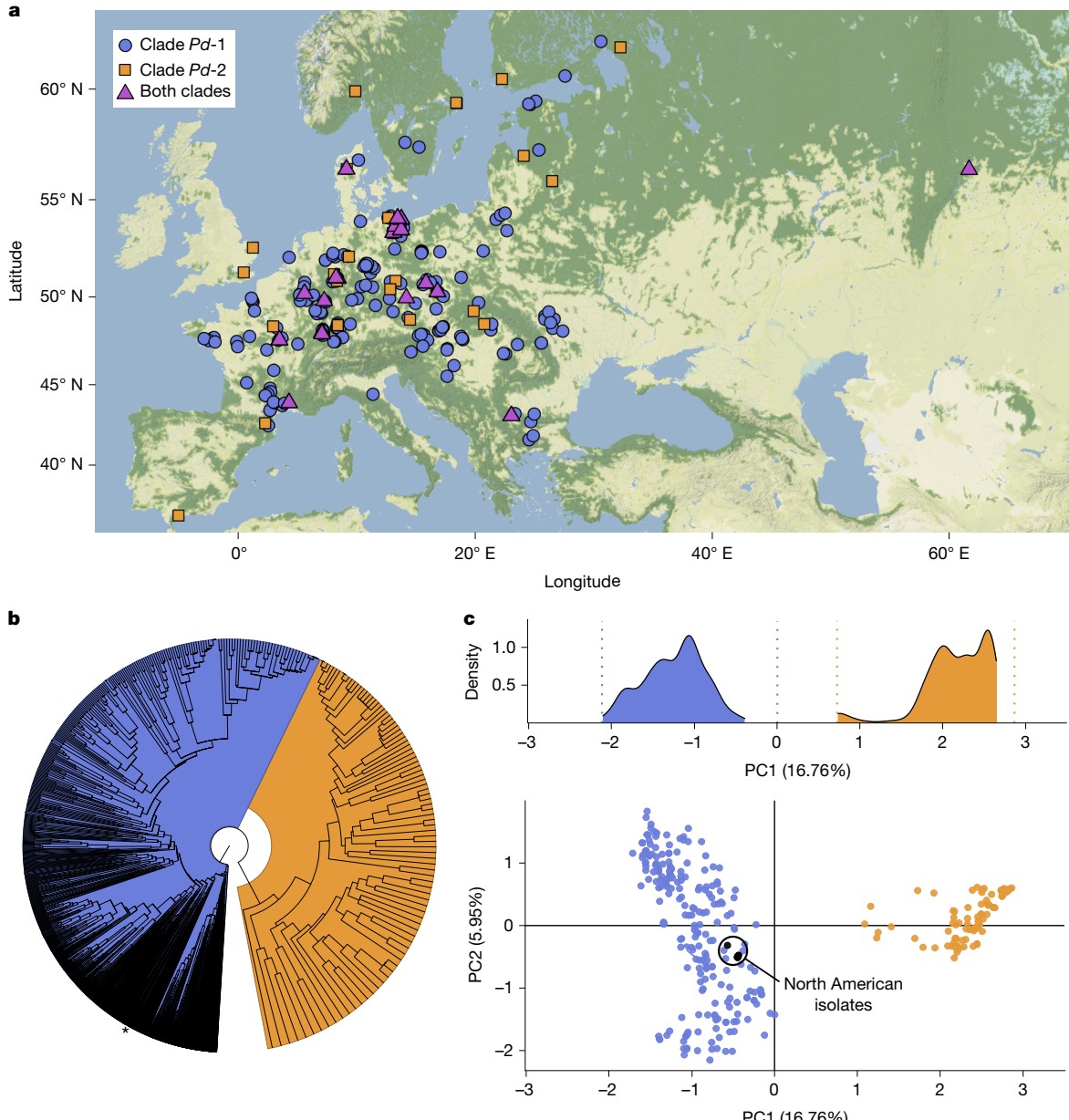

**Fig. 1 | Multilocus microsatellite typing reveals two clades of *P. destructans*.** **a**, Sampling locations in Eurasia (North American sites not shown; *n* = 255). **b**, Phylogenetic tree, based on the $D_A$ distance, representing the relationship between all 1,866 unique multilocus genotypes (North American isolates are indicated with an asterisk) originating from the 5,479 isolates from 264 sites based on 18 microsatellite loci (for the $D_A$ distance, see the section 'Analyses of multilocus genotypes' in the Methods; the monophyly of *Pd*-1 and *Pd*-2 was consistently recovered when jackknifing loci). For better visualization of both clades, the section containing *Pd*-2 was magnified. **c**, Principal component analysis (bottom) of isolates (*Pd*-1 was subsampled to ensure even sampling between clades and to maximize geographical coverage, which resulted in 234 isolates for *Pd*-1 and 92 isolates of *Pd*-2; Supplementary Table 1). Density (top) of principal component 1 (PC1) coefficients after random subsampling of *Pd*-1 to obtain the same number of sampling sites as for *Pd*-2 (51 sites, replicated 100,000 times). Dotted lines represent the absolute edges of the distribution (see the section 'Analyses of multilocus genotypes' in the Methods for details).

(temperature and humidity) and biotic factors (host specialization), between the two pathogen species.

## Two sympatric clades cause the disease

The 5,479 isolates obtained from bats and/or hibernacula environments were genotyped at 18 polymorphic microsatellite loci[20] (Fig. 1a, Supplementary Tables 1–3 and Supplementary Fig. 6). Phylogenetic reconstruction of genotype relationships (Fig. 1b) revealed a clear separation into two clades, a division further corroborated by principal component analysis (Fig. 1c). These clades are hereafter called *Pd*-1 and *Pd*-2, where *Pd*-1 corresponds to *P. destructans sensu stricto*

(Supplementary Figs. 1 and 2). Both *Pd*-1 and *Pd*-2 cause white-nose disease, as both were isolated from bats that exhibited lesions diagnostic of the disease[14,17,21,22]. At the broad scale, these two clades are geographically sympatric and were found present together at 23 of the sites we studied, which makes it unlikely that geographical factors currently play a major part in their differentiation (Fig. 1a). Occasionally, isolates from both clades were syntopic (that is, isolated from the same swab, *n* = 18). The North American isolates, which are known to have a single clonal origin[12,23], clustered with Eurasian isolates of *Pd*-1 (Fig. 1b,c and Supplementary Fig. 2).

Apart from geography, host specialization can act as a driver of population differentiation[24]. Therefore, we investigated the relationship

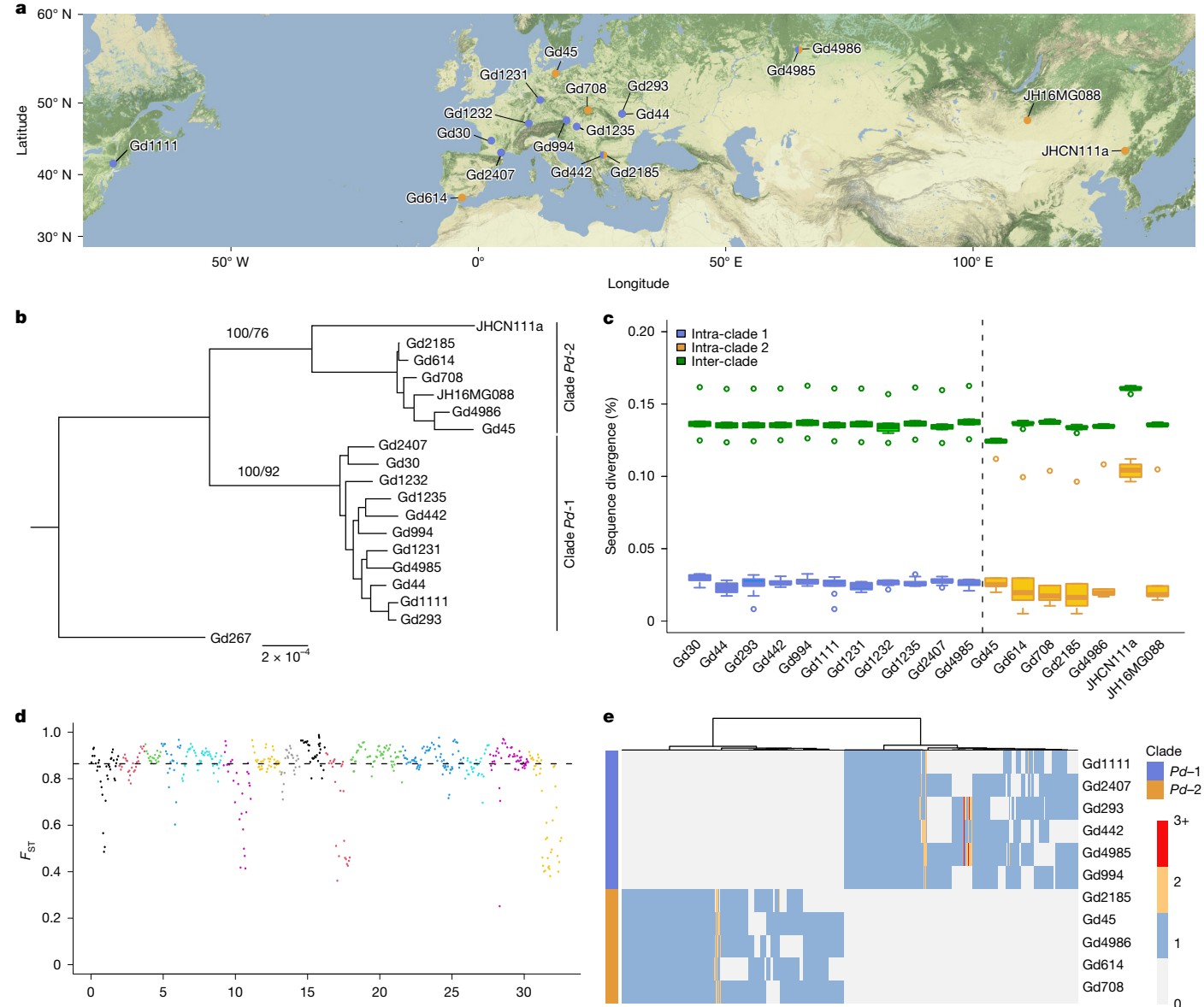

**Fig. 2 | Genomic differentiation between clades. a**, Sampling locations of the 18 isolates used for phylogenetic and sequence divergence analyses. **b**, Phylogenetic tree of 664 BUSCO genes with 1,000 bootstraps and site concordance factors for nodes of interest (as percentages). The branch to the outgroup Gd267 has been shortened for visualization purposes. **c**, Boxplot of the pairwise genetic distances between isolates (each isolate is compared with the 17 other isolates) for the 664 BUSCO genes, partitioned between intra-clade and inter-clade distance. The darker line indicates the median, and the lower and upper hinges represent the first and third quartiles, respectively.

The whiskers extend to 1.5 times the interquartile range, with any data points beyond this range marked as outliers. **d**, Genomic differentiation between pools of 69 and 63 individuals from clades *Pd*-1 and *Pd*-2, respectively, estimated using $F_{ST}$ across a window size of 200 SNPs, using Gd293 (*Pd*-1 clade) as the reference genome, with its 18 contigs successively coloured. **e**, Heatmap of phylogenomic profiling depicting clade-specific synteny network clusters, with isolates in rows and clusters in columns ($n = 1,365$; profile for run 13, Supplementary Table 17). Grey denotes the absence of a gene, whereas other colours indicate the number of gene copies (see the key).

between *Pd*-1 and *Pd*-2 isolates and the species of bat from which they were isolated. Out of the 1,463 swabs collected from a total of 16 bat species sampled at 241 sites across Europe, 87% of those containing *Pd*-1 isolates originated from *Myotis myotis* and *Myotis blythii* (*M. myotis/blythii*) bats, whereas none were found on *Myotis daubentonii*. By contrast, the majority of swabs containing *Pd*-2 isolates originated from *M. daubentonii* (39%), whereas only 34% originated from *M. myotis/blythii* (Extended Data Fig. 1). These contrasting data reveal a significant association between the fungal clades (*Pd*-1 and *Pd*-2) and the bat host species when tested in a Bayesian hierarchical model (Supplementary Table 7). The estimated 95% credible intervals for the probability of different bat species with *Pd*-2 were as follows: $1.07 \times 10^{-42}$–$4.98 \times 10^{-5}$

for *Myotis dasycneme*; $8.07 \times 10^{-26}$–$1.65 \times 10^{-5}$ for *M. myotis/blythii*; and $2.38 \times 10^{-22}$–$0.017$ for the *Myotis nattereri* species complex (*M. nattereri*, *M. escalerai* and *M. crypticus*). *Myotis mystacinus* was equally likely to be infected by both *P. destructans* clades, whereas *M. daubentonii* had a near-certain probability (almost 1.00) of harbouring *Pd*-2 (Supplementary Table 7), which indicated that it had exclusive infection with *Pd*-2. Such host specialization cannot be attributed to species distributions, as *M. daubentonii* occurs throughout the area sampled (Supplementary Fig. 3). Furthermore, on the basis of the bat monitoring data available for the 172 sites studied, we had direct evidence that at least 63% of the sites containing *Pd*-1 isolates are also used as hibernacula by *M. daubentonii*. The latter species of bat is therefore

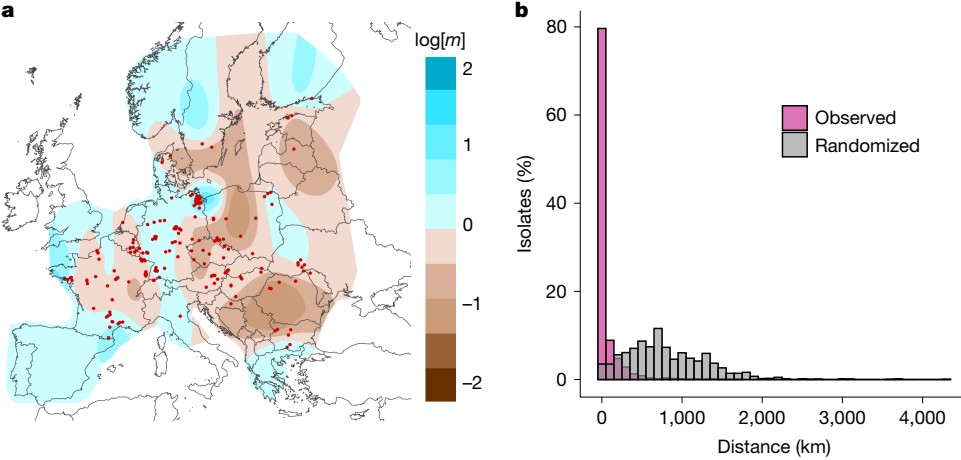

**Fig. 3 | Strong population differentiation in *Pd*-1. a**, Estimation of effective migration (*m*) surfaces based on 2,261 isolates from *Pd*-1 in Europe (all sites excluding Russia and the United States after clone correction, *n* = 225 sites). For visualization, results from eight independent runs (each with 8 million iterations and between 100 and 450 demes) were combined. Different shades of colour represent variable levels of high (blue) or low (brown) effective migration rates. Sampling locations are represented by red dots. **b**, Distribution of the distance between the true and assigned site of each Eurasian isolate of *Pd*-1 (2,191 isolates; dataset limited to 20 isolates per site) for the observed and randomized datasets of DAPCs (bin width, 100 km).

expected to come into contact with *Pd*-1 isolates, yet does not seem to be susceptible to infection by them.

These findings of host specialization highlight the need to discriminate fungal species when elucidating the strategies, such as resistance or tolerance, used by host species when faced with a pathogen. This approach is particularly relevant in the context of diseases such as white-nose, for which disentangling the interactions between hosts and pathogens is crucial for comprehending disease dynamics[10]. Although we do not yet have data linking these characteristics to host specialization, the colouration of agar medium in laboratory cultures was significantly different (based on 45 isolates for *Pd*-1 and 34 isolates for *Pd*-2; two-sided *t*-test *t* = −11.58, d.f. = 60.06, *P* < 0.001) between *P. destructans* clades, with only limited overlap (Extended Data Fig. 2 and Supplementary Table 8). However, *Pd*-1 and *Pd*-2 were encountered in similar abiotic conditions—cold (about 7–8 °C) with high absolute humidity (around 6–8 g m$^{-3}$; Extended Data Fig. 3 and Supplementary Table 3)—and had similar colony expansion rates in culture (based on 45 and 34 isolates for *Pd*-1 and *Pd*-2, respectively; Extended Data Fig. 4 and Supplementary Table 9) with no significant difference in final culture size after 7 weeks (two-sided *t*-test, *t* = −0.044, d.f. = 68.54, *P* = 0.96). These similarities are consistent with the observation that *Pd*-1 and *Pd*-2 share the same geographical range and are often found in the same caves. Therefore, according to existing evidence, host specialization seems to be an important factor that distinguishes their ecological niches. The precise mechanisms that underlie *P. destructans* host specialization may involve a combination of host behavioural differences, immune competence, microbiome composition, skin properties and thermal and hibernation physiology. However, such factors are inherently difficult to disentangle because of their interconnected nature.

## Genomic divergence between clades

To further characterize the extent of genomic divergence between clades, we selected ten Eurasian *P. destructans* isolates (five from each *Pd*-1 and *Pd*-2 clade), one North American *P. destructans* isolate (*Pd*-1) and one *Pseudogymnoascus* sp. outgroup for full-genome long-read sequencing (MinION nanopore; Supplementary Table 10). High quality (>98% complete BUSCO genes) and contiguous genomes (median of 39 contigs; range of 18–132) were assembled using Flye[25] and polished using 150 bp paired-end Illumina reads (Supplementary Table 11). To infer the relationship between isolates, we extracted 664 single-copy BUSCO genes (1,648,545 bp alignment) common to 19 isolates: 12

sequenced in this study (including the outgroup) and 7 previously published[12] (5 from Europe, 1 from Mongolia and 1 from China). The high bootstrap (100%), site concordance factors (92% for *Pd*-1 and 76% for *Pd*-2) and congruence between gene trees (Fig. 2b and Supplementary Fig. 4) confirmed the monophyly of the two clades identified using microsatellites (Fig. 1b,c). This result was further corroborated by a phylogenenetic tree that was reconstructed from partitioning the genome into 10 kb windows (Supplementary Fig. 5). The population differentiation, as measured using the fixation index ($F_{ST}$), was consistently high between clades (median = 0.79; Extended Data Fig. 5). Both clades included samples from Europe and Asia, a result that further confirmed that the separation is not driven by geography. For example, isolate Gd4986 (clade *Pd*-2) from the Ural Mountains in Russia was more similar to isolates from Spain, Mongolia and China than to another isolate from the same cave but from clade *Pd*-1 (Gd4985). Across 664 conserved BUSCO genes, inter-clade sequence divergence (median of 0.14%) was >5 times greater than intra-clade divergence (0.022–0.026%; Fig. 2c), whereas the inter-clade divergence across the full genomes averaged 1.6% (Extended Data Fig. 6). To gain further insights into the genomic differentiation of the clades across the full genome of a larger number of isolates, we sequenced pools of DNA from 69 and 63 isolates from *Pd*-1 and *Pd*-2, respectively (Supplementary Table 15). Short reads of each pool were mapped to the reference genome of *Pd*-1 and *Pd*-2 to estimate the $F_{ST}$ between pools (Supplementary Fig. 6). Irrespective of the reference genome used, the differentiation between clades was strong, with a minimum $F_{ST}$ of 0.25 and a median $F_{ST}$ of 0.88 (mean = 0.85; window size of 200 single-nucleotide polymorphisms (SNPs); Fig. 2d and Extended Data Fig. 7). Less than 2.7% of SNPs were shared between the clades. Approximate Bayesian computation methods were used to model the long-term demographic history of both clades, and a strict-isolation model (that is, no gene flow) was identified as the most likely scenario (Supplementary Table 16). The divergence between the clades was estimated at 750,000 generations (95% credible interval 114,000 to 1.5 million; Supplementary Table 16). Moreover, the widespread recombination detected in the clades (Extended Data Fig. 8 and Supplementary Fig. 1) demonstrated that *P. destructans* displays a degree of non-clonality, which provides strong evidence of genetic exchange among isolates in the clades for which both mating types are common (57% for MAT1-1 and 43% for MAT1-2 in each clade; Supplementary Fig. 1). This genetic exchange in the clades, combined with the strict isolation between sympatric clades, provides compelling evidence that isolation mechanisms other than clonality or

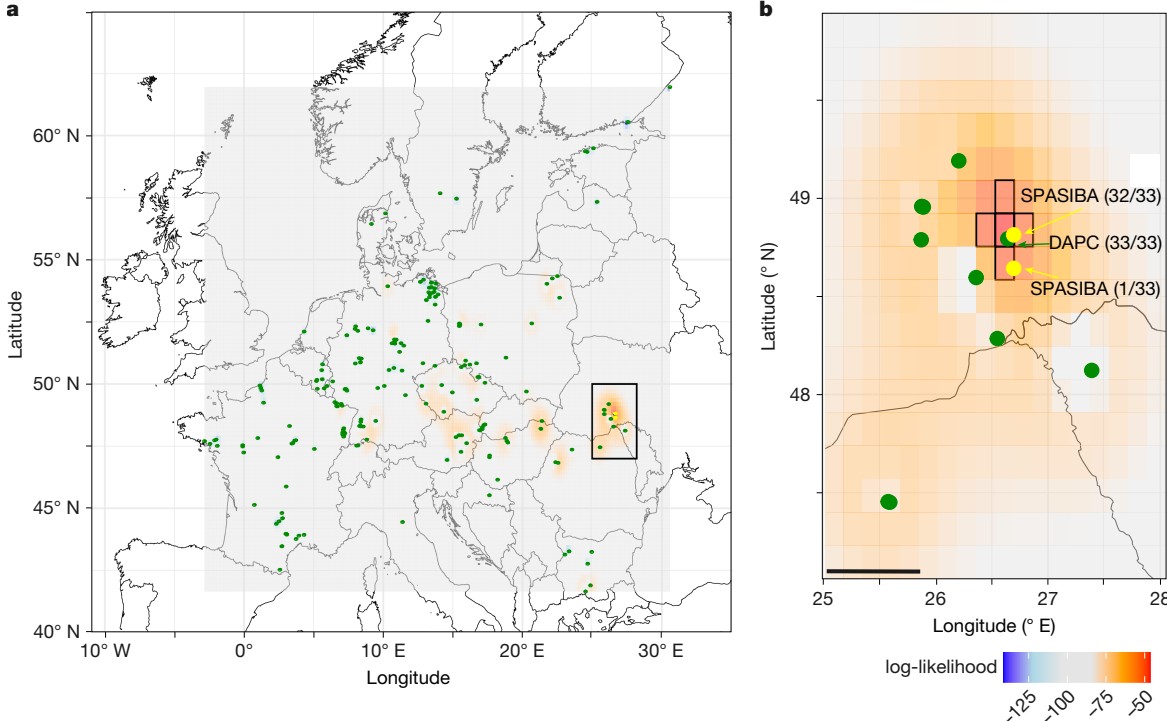

**Fig. 4 | Assignment of the source of the North American introduction of *P. destructans* using Bayesian inference. a**, Bayesian inference (SPASIBA analysis; trained on 5,162 *Pd*-1 isolates) was performed independently for each of the 33 North American isolates across the continuous landscape (>5 million square kilometres). We then calculated the log-likelihood of the assignment across all 33 isolates (Supplementary Table 5), depicted on the map by a colour scale from blue (lowest log-likelihood) to red (highest log-likelihood). Green dots represent sites from which *Pd*-1 samples were collected, and the two yellow dots, in Podillia, Ukraine, indicate the inferred origin of the introduced ancestor of the 33 North American isolates (Supplementary Table 16). The zoomed-in area shown in **b** is represented here by the black rectangle. **b**, Details of the region with the most probable assignment. This area has a 6 × 10⁹ greater likelihood of assignment than any pixel occurring outside this region (Supplementary Table 5). The central black-bordered pixel has the highest likelihood of assignment and contains the site to which the DAPC assigned all 33 North American isolates. The remaining black-bordered pixels have a likelihood within an order of magnitude of the central black-bordered pixel. Scale bar, 50 km.

geography prevent detectable genetic exchange between *Pd*-1 and *Pd*-2 clades.

In addition to sequence divergence, genome organization can reveal key differences between clades. We performed network-based microsynteny analyses on the annotated genomes to reconstruct the phylogenetic relationship between isolates based on microsynteny and to detect and quantify clade-specific syntenic clusters. A typical synteny cluster is a group of syntenic genes shared by individuals, which reflects the phylogenetic relatedness of their genomic architecture. Analyses, replicated with 25 different input settings, recovered synteny for more than 92% of genes, which belonged to 9,136 clusters on average (Supplementary Table 17). A mean of 1,316 of these clusters (14.4%), spread across contigs, were not shared between clades, of which 1,250 contained single-copy genes and 66 multicopy genes (Fig. 2e). The phylogeny reconstructed from binary coding of the presence or absence of clusters consistently recovered both *P. destructans* clades as monophyletic with maximal support (Supplementary Table 17). These analyses reveal substantial genomic structural differences between the clades, differences that have been associated with reproductive isolation in other fungal species[26,27].

This level of genomic differentiation strongly supports the presence of two cryptic *Pseudogymnoascus* sp. according to the established recommendations for fungal species delimitation (for example, refs. 28–30). We further substantiated our conclusion by demonstrating reciprocal monophyly, recombination within clades and the absence of gene flow between the divergent yet sympatric and syntopic clades. These three features are primary indicators of completed speciation. Combining the molecular, ecological (host specialization) and distribution data, we conclude that there are two pathogenic cryptic

species in the *Pseudogymnoascus* genus, native to Eurasia, which are both infecting hibernating bats and causing white-nose. Whether speciation occurred in allopatry or in sympatry, possibly through host specialization as the barrier to gene flow[24], cannot be confirmed with the current dataset. Nevertheless, our data clearly identified *P. destructans* and its cryptic relative as a previously unknown system in which to investigate speciation through host specialization in a mammalian fungal pathogen. In terms of implications for conservation, the presence of another pathogenic fungus with different host specialization that is able to cause white-nose poses a risk to bat conservation, particularly outside its current range. On the basis of a risk assessment conducted in Australia, which concluded that in the next 10 years, the introduction of *P. destructans* is very likely to almost certain, we can confidently infer that *Pd*-1 and *Pd*-2 are highly likely to be introduced to regions where they are currently absent[31]. Microsatellite data provided clear evidence that *Pd*-1 is largely dominant in Europe, accounting for 95% of samples. Despite the limited number of isolates genetically characterized from East Asia (*n* = 4; Supplementary Tables 10 and 14), the exclusive identification of *Pd*-2 in this region implies its potential dominance, with *Pd*-1 yet to be confirmed. Although *Pd*-2 is currently absent from North America, its introduction could pose a serious risk to bat species previously unaffected by *Pd*-1. Furthermore, species recovering from *Pd*-1 exposure may face new challenges if *Pd*-2 is introduced. The emergence of chytrid fungi provides a cautionary example of the consequences of delayed action. For example, *Batrachochytrium dendrobatidis*, a generalist pathogen, was linked to amphibian declines as early as the 1970s. However, at the time, there was no awareness that a second, more specialized species could emerge[32]. When *Batrachochytrium salamandrivorans* was finally identified in 2013, it was already too late

to prevent its introduction to Europe[32]. By contrast, with *Pd*-2, we have the advantage of foresight—an opportunity to strengthen global biosecurity and prevent its spread beyond its native range.

## Population structure and origin

In the dataset from Eurasia (5,446 isolates in total), we detected 1,766 distinct multilocus genotypes (called 'genotypes' hereafter) belonging to *Pd*-1 (5,165 isolates from 227 sites) and 92 genotypes in *Pd*-2 (281 isolates from 51 sites). To test whether an isolate could be genetically assigned to its site of origin among all sampled sites in Europe, we performed discriminant analysis of principal components (DAPC)[33] for each clade separately. As many as 64% of isolates from *Pd*-1 and 69% from *Pd*-2 from Eurasia were successfully reassigned to their exact site of origin. In terms of distances, isolates were reassigned at an average distance of only 52.6 km (*Pd*-1) and 42.9 km (*Pd*-2) from their site of origin, which is very near given that samples originated from sites up to several thousands of kilometres apart. These high reassignment rates cannot be attributed to chance as they sharply contrasted with a 'null-DAPC', whereby information on sites was randomized before running the DAPC, which resulted in only 0.55% and 2.15% of isolates correctly assigned to sites for *Pd*-1 and *Pd*-2, respectively (Fig. 3b and Extended Data Fig. 9). The strong observed population structure, which resulted from the limited effective movement of the fungi across the landscape, could be attributed to either restricted movement of *P. destructans* through bats or the reduced fitness of emigrant *P. destructans* isolates attempting to establish in sites already occupied by other *P. destructans* isolates[34]. To obtain a spatially informed overview of the genetic discontinuities of each clade, we conducted an estimation of effective migration surfaces[35]. The analysis revealed three genetic discontinuities for *Pd*-1: one between the Balkans and the rest of Europe, one dividing Europe in two (from Poland through to Slovenia) and the last one dividing France and Iberia from the rest of Europe (Fig. 3a; see Extended Data Fig. 9 for clade *Pd*-2). Notably, none of these discontinuities was associated with a discontinuity in the main host species *M. myotis* (for examples, see refs. 36,37), which provides evidence that the main host population structure is probably not a key driver of the population structure of the pathogens (see also ref. 34). Moreover, previous studies of *Myotis* sp. hosting *P. destructans* have indicated near panmictic populations on a large scale[38], thereby further supporting the notion that despite them moving across the landscape, bats are not effectively transporting *P. destructans*, even over limited distances of a few hundreds of kilometres[34]. Although both fungal species are native to Eurasia, bats may be at risk from exposure to more virulent strains originating from long-distance inter-specific or intra-specific genetic exchanges. Moreover, such exchange may be exacerbated by human-mediated movements of the fungus, as already observed between eastern and western North America[39].

Comprehensive sequencing of more than 60 isolates from North America has shown that they all originated from a single clonal source[12,23]. Genotypes obtained from North America for this study, clonal descendants of the introduced isolate, unequivocally belonged to *Pd*-1 (Figs. 1b,c and 2b). Furthermore, the strong population structure observed in the native range of *P. destructans* enabled us to trace back the most likely source population of the North American introduction to the region of Podillia in Western Ukraine. This conclusion was strongly supported by results from two independent methods: multivariate DAPC and spatial Bayesian inference (SPASIBA analysis)[40]. Results from multivariate DAPC confidently assigned the origin of all 33 North American isolates to a single site in Podillia (posterior probability of 1; Supplementary Table 5). SPASIBA analysis also identified Podillia as the most probable source of the North American introduction (Fig. 4 and Supplementary Table 18). Both methods identified the source of the North American introduction within a 9 km radius in Podillia

(Fig. 4b), despite the potential assignment zone spanning more than 5 million square kilometres, thereby strengthening confidence in this result. In agreement with these findings, all phylogenetic analyses—whether based on microsatellites (Fig. 1b and Supplementary Fig. 2), BUSCO genes (Fig. 2b) or whole genomes[12] (Supplementary Fig. 5)—consistently identified isolates from Podillia as the closest relatives to those from North America. Podillia is home to some of the longest caves in the world, such as the giant maze caves Optymistychna (longest in Europe; 267 km) and Ozerna (143 km), which are of major international speleological interest. Moreover, since the dissolution of the Soviet Union in 1991, the region has attracted cavers from around the world, including the United States, with notable exchanges involving caving communities from upstate New York[41,42], the area where *Pd*-1 was introduced to North America[16]. Although absolute evidence may be unattainable, the observed pattern of caving exchanges between the source region and the site of introduction provides strong support for the proposed mechanism behind the initial introduction of *Pd*-1 to North America (Supplementary Note 1). This finding highlights the substantial risk posed by international caving activities in the spread of biological agents and underscores the need to improve knowledge of pathogen pollution while raising awareness of essential biosecurity measures[43], such as limiting the movement of caving equipment, particularly when not properly decontaminated[44].

The identification of the region of origin and the discovery of a second causative agent were achievable thanks to the rapid and extensive sampling effort across a substantial part of the native range of the species. Such intensive and synchronous sampling (typically conducted around February–March each year) at a continental scale was only possible through the combined effort of hundreds of volunteers (Supplementary Note 2). Our study demonstrates the potential of tapping into the synergism of citizen engagement for future surveillance of emerging pathogens.

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

## Methods

### Sample and field data collection

Swab samples of *P. destructans* were collected from bat hibernacula. Sampling from hibernating bats was conducted without capture or handling by collecting samples while the bats remained freely hanging. The samples were collected by lightly swabbing the infected areas with a sterile dry swab (Polyester swab 164KS01, Copan). This method is considered minimally invasive or even noninvasive[45,46]. The timing of sample collection was usually between January and April, when the highest numbers of bats with visible infection have been reported[15]. When no bats were present or sampling them was not possible, wall swabs were collected by touching the swab to hibernacula walls (ideally close to where bats usually hang to hibernate; see ref. 34 for more details). Four isolates were also obtained from sediment samples collected from inside hibernacula and 22 were collected from caving gear (that is, caving suits and harnesses), which most probably originated from contact with hibernacula environments[44].

When a sample was taken from a bat or in close proximity (within about 10 cm), the bat species was also recorded. Temperature and relative humidity were measured in the hibernacula. Absolute humidity was then calculated from measures of relative humidity and temperature by applying a previously described formula[47].

Our work adhered to the ethical wildlife research guidelines of the American Society of Mammalogists for the use of wild mammals in research and education[48]. Furthermore, this work was conducted under permission from the following authorities: Italy, Regional Speleological Federation of Emilia-Romagna (FSRER) and the Management Bodies of the Parks of Emilia-Romagna; Poland, Genarny Dyrektor Ochrony Środowiska (General Director for Environmental Protection) and Regional Directorate for Environmental Protection in Gorzów Wielkopolski (Regionalna Dyrekcja Ochrony Środowiska w Gorzowie Wielkopolskim); Switzerland, Kantonaler Fledermausschutz Aargau; Germany, Umweltamt, Veterinäramt, Untere Landschaftsbehörde Siegen-Wittgenstein, Untere Naturschutzbehörde Umweltamt Landkreis Harz und Referat Verbraucherschutz, Veterinärangelegenheiten Landesverwaltungsamt Sachsen-Anhalt, Untere Naturschutzbehörde des Landkreises Vorpommern-Greifswald, Regierung von Unterfranken, Regierung von Mittelfranken, Struktur- und Genehmigungs Direktion Nord/Süd, NLWKN Niedersächsischer Landesbetrieb für Wasserwirtscht, Küsten- und Naturschutz und Region Hannover–Fachbereich Umwelt; Austria, Department of Nature Conservation for Carinthia, Lower Austria, Upper Austria, Salzburg, Styria and Vorarlberg; Hungary, Pest Megyei Kormányhivatal, Országos Környezetvédelmi, Természetvédelmi és Hulladékgazdálkiodási Főosztály (Pest County Government Office, National Department of Environment Protection, Nature Conservation and Waste Management) and the Ministry of Environment and Water; Bulgaria, Bulgarian Ministry of Environment and Water; France, DDTM-Morbihan and DREAL; Republic of Latvia, Nature Conservation Agency; Belgium, Gouvernement Wallon; Denmark, The Nature Agency and Daugbjerg Kalkgruber; Romania, Speleological Heritage Commission; Estonia, Estonian Environmental Board; England, Natural England; Finland, Southwest Finland Centre for Economic Development, Transport and the Environment; Sweden, Uppsala djurförsöksetiska nämnd, Swedish board of Agriculture and the Swedish Environmental Protection Agency; Norway, Miljødirektoratet; Luxemburg, Ministère du Développement durable et des Infrastructures du Luxembourg; Croatia, Croatian Ministry of Environment and Nature; Russian Federation, Game Management Directorate of the Republic of Karelia, Institute of Plant and Animal Ecology and the Ural Division of the Russian Academy of Sciences; Slovak Republic, Ministry of the Environment of the Slovak Republic and the Department of State Administration for Nature and Landscape Protection; the Netherlands, Dutch Ministry of Economic affairs; Republic of Moldova, Government of Republic of Moldova–Ministry of Environment.

### Laboratory materials and methods

**Cultures and genotyping.** Previously published DNA extraction and genotyping protocols were used[20,34], and are briefly outlined here. *P. destructans* was collected using sterile swabs from hibernating bats and the walls of sites where bats hibernate. The collected fungal material was cultured on dextrose peptone yeast agar[49] using classical mycological procedures and sterilization of tools between each use. After observing germination, typically 3–5 days after plating, plates were screened with a microscope to identify germinated single spores (identified as colonies expanding from a single germinating spore). Depending on availability, 1–3 (mean = 3.0, median = 3) and 1–5 (mean = 3.6, median = 2) single spores were typically isolated from bat and wall swabs, respectively. Isolation was performed by excising a plug with a sterile 3-mm biopsy punch and transferring it to a fresh 6-cm Petri dish. The plates were then visually monitored for 1 week to confirm that no additional spores germinated on the plug. Plates were grown at 10–15 °C until there was sufficient material to extract DNA (usually after several weeks to months). Each of these colonies is then referred to as an isolate or a culture. DNA extraction was done using a KingFisher Flex extraction robot (Thermo Scientific) with a MagMAX Plant DNA Isolation kit (Thermo Scientific). After DNA extraction, isolates were genotyped using 18 microsatellite markers and two mating-type markers in four multiplexes. The two mating-type markers were two independent primer sets used to amplify segments of the two mating types (MAT1-1 and MAT1-2) that are different in length and therefore diagnosable through fragment-length analysis[20]. Genotyping was carried out on an ABI 3130 Genetic Analyser (Applied Biosystems), and GeneMapper software (v.5; Applied Biosystems) was used for fragment analysis.

**DNA extraction for MinION and Illumina reads.** Material was collected from *P. destructans* cultures using sterilized tweezers. We used a sorbitol wash buffer (100 mM Tris-HCl pH 8.0, 0.35 M sorbitol, 5 mM EDTA pH 8.0 and 1% (w/v) polyvinylpyrrolidone (PVP-40)) to clean the fungal material and to remove most of the culture medium from the hyphae (the wash was repeated twice with 5 min of incubation at room temperature each time). After removing the sorbitol wash buffer the second time (through centrifugation and removal of the liquid supernatant), 500 µl CTAB lysis buffer (preheated to 65 °C; 0.01 M Tris HCl pH 7.5, 25 mM EDTA pH 8.0, 1.5 M NaCl and 2% CTAB powder (w/v)), 30 µl proteinase K and 5 µl 1 M DTT were added for digestion and incubated overnight at 56 °C, mixing material after the first hour. After letting samples cool for 5 min at room temperature, 4 µl RNase A was added and left to incubate at room temperature for 10 min. One volume chloroform–isoamyl alcohol (24:1 v/v) was added, after which tubes were inverted 30 times and centrifuged for 5 min at maximum speed, keeping the supernatant. We then added a second step of proteinase K (30 µl) and RNase A (4 µl) treatment with an incubation for 30 min at 56 °C, as it was found to reduce the presence of RNA and result in better quality DNA. To remove these enzymes, we performed a second chloroform–isoamyl alcohol (24:1 v/v) extraction step by adding 1 volume, inverting 30 times and centrifuging at maximum speed for 5 min, after which the supernatant was kept. Precipitation of DNA was achieved with the use of 1/10 volume sodium acetate, 2 volumes ethanol (>99% purity) and centrifugation for 20 min at maximum speed. After gentle removal of the sodium acetate–ethanol mixture, the resulting pellet (containing the DNA) was washed twice with 70% ethanol. DNA was then eluted in ddH$_2$O and stored in the fridge. The DNA content was determined using Qubit (ThermoFisher).

**Sample preparation for MinION nanopore sequencing.** We performed long-read Oxford Nanopore Technology (ONT) sequencing of 12 isolates (5 Eurasian per clade, 1 North American and 1 outgroup; Supplementary Table 10) using MinION flowcells (FLO-MIN-106) using

libraries prepared with an ONT Ligation Sequencing kit SQK-LSK109, following the manufacturer's instructions. Statistics of the long-read sequencing and the associated assembled genomes are presented in Supplementary Tables 8 and 9.

**Sample preparation for Illumina sequencing (individual isolates).** Illumina sequencing was performed for all the isolates for which we performed MinION long-read sequencing except for Gd1111, for which Illumina sequences were already available (Gd1111 = 20631-21, sub-culture of the type isolate). For all samples except Gd45 and Gd293, Illumina-indexed libraries were prepared for each isolate according to a previously described protocol[50] with modifications as proposed in a previous study[51]. Libraries were then sequenced (150 bp, paired-end) by Novogene on an Illumina NovaSeq 6000. For Gd45 and Gd293, libraries were prepared using TruSeq DNA PCR Free (350) and TrueSeq Nano DNA (350) kits, respectively, before being sequenced (150 bp, paired-end) by Macrogen on an Illumina HiSeq X.

**Sample preparation for Illumina sequencing (Pool-seq).** We pooled DNA from multiple isolates (details on their origin presented in Supplementary Table 1) in equal concentrations into sample pools for sequencing. A total of four pools were prepared per clade with each isolate appearing in one pool only. For *Pd*-1, a total of 69 isolates were used (pool sizes: 17, 17, 17 and 18 isolates), whereas for *Pd*-2, a total of 63 isolates was used (pool sizes: 15, 16, 16 and 16 isolates). Within clades, isolates were assigned to a pool on the basis of the DNA concentration of their extracts (that is, the 17 and 15 isolates with highest DNA concentration for *Pd*-1 and *Pd*-2, respectively, were pooled together). The strategy of pooling samples into four pools per clade was used to validate the consistency of the results generated by each pool individually and the combined dataset (see the section 'Genotyping'). Isolates were chosen to maximize both the geographical distance among sites and the genotypic richness within each clade. After DNA extraction (and quantification) of each isolate, DNA was combined to result in equal concentrations of isolates with a total of 500 ng DNA in a volume of 60 µl per pool. Illumina-indexed libraries were prepared for each pool (that is, isolates were not individually indexed for Pool-seq) according to a published protocol[50] with modifications as previously proposed[51]. Libraries were then sequenced (150 bp, paired-end) by Novogene on an Illumina NovaSeq 6000.

## Analyses of multilocus genotypes

The analyses of multilocus genotypes (MLGs) were run in R (v.4.1.1)[52], except for estimated effective migration surfaces (EEMS), using packages for specific analyses. Specifically, the package poppr (v.2.9.3)[53] was extensively used as it provides the tools needed for population genetic analyses of haploid species with clonal reproduction (such as *P. destructans*).

MLGs were defined by their unique combination of alleles across the 18 polymorphic microsatellite loci. This set of markers is sufficient to reliably identify the identity of MLGs both among and even within sites, for which MLGs are usually less differentiated[34]. Across all isolates, the allelic richness was high, ranging from 10 to 93 alleles per locus (mean = 37); however, we found that some alleles were fixed in the *Pd*-2 clade (Supplementary Table 6). Only isolates with a maximum of 4 missing alleles (that is, successfully genotyped at 14 microsatellites or more) were used for analyses, which resulted in a dataset comprising 5,479 isolates.

Principal component analysis (PCA) was used to visualize the differentiation among isolates (package adegenet (v.2.1.5)[54]). As the outcome of PCAs depends on sampling intensity[55], it was important to select roughly equal sample sizes among clades to capture their differentiation in Eurasia. To achieve this, we chose 51 sites (the same number of sites in which *Pd*-2 was found) from *Pd*-1 in a way that maximized geographical distance among them (that is, thinning sites) and used up

to 20 isolates per site (again, to ensure that sampling among sites was not markedly uneven). This resulted in a dataset containing 234 and 92 isolates for *Pd*-1 and *Pd*-2, respectively, over 51 sites each (using unique MLGs only). PCA was then performed using this *Pd*-1 dataset subset and the full dataset for *Pd*-2. The results revealed two clusters, which were completely differentiated. The position of the North American isolates on the PCA plot was simply determined a posteriori by projecting or predicting their coordinates using the function 'suprow'. Considering that maximizing distance among sites may also have an influence, we confirmed these findings on the Eurasian dataset by randomly subsampling 51 sites of *Pd*-1 repeatedly (100,000 times, but with the number of isolates per site still capped at 20) without considering the geographical distance between chosen sites. After running these 100,000 subsampled PCAs (with *Pd*-2 unchanged, the geographically thinned PCA run was added, which resulted in a total of 100,001 PCAs), the density of values observed for PC1 was consistent with the values obtained for the thinned dataset, which indicated that the signal of differentiation between *Pd*-1 and *Pd*-2 was strong and independent of geography or identity of the chosen sites (Fig. 1b).

We also used the microsatellite dataset to investigate the presence of population differentiation in each of the discovered clades. For this purpose, we investigated only Eurasian isolates (that is, excluding the 33 isolates from the United States) and treated *Pd*-1 and *Pd*-2 separately.

First, we used DAPC[33] (using the package adegenet) to assign each isolate to one site among all European sites sampled. If populations from different sites are genetically differentiated, one would expect the DAPC to assign isolates to their true site of origin (that is, where they were sampled) more often than expected by chance. Here each isolate was probabilistically assigned to sites based on the observed allele frequencies (no assumptions were made, for example, about the independence of loci). Each isolate was run in an independent DAPC, which resulted in a set of 2,191 and 279 DAPCs for *Pd*-1 and *Pd*-2, respectively (excluding all isolates from North America and with a limited number of 20 isolates per site, see also Supplementary Table 1; 120 PCA axes and 100 discriminant analysis ($D_A$) axes retained in all runs). As some isolates will always be correctly assigned by chance, it was important to quantify the percentage of correct assignments by chance compared with correct assignments based on observed allele frequencies. Hence, we ran the same sets of DAPCs after randomizing the site names (independently for each run) to ascertain the frequency of correct assignments occurring by chance and the expected distances between the isolates' site of origin and their assigned sites if assignment was no better than random (randomized DAPC). The distances between assigned sites and sites of origin are presented in the main article for *Pd*-1 (Fig. 3b) and in Extended Data Fig. 9 for *Pd*-2.

To identify the European sites of origin of the North American introduction, we performed two separate analyses with two different methods: multivariate analyses (DAPC[33]) and spatial Bayesian inferences (SPASIBA[40]). DAPC analyses assign samples to sites already contained in the dataset, whereas the SPASIBA method can perform continuous assignment to any location (that is, coordinates) in the range defined by the user. First, we built a DAPC with all 5,162 *Pd*-1 isolates from 226 sites in Europe (the site in the Ural Mountains was excluded). The aim of this DAPC was to differentiate sites; hence, sites were used as pre-defined groups. We then used the 'predict.dapc' function from the adegenet R package to predict site memberships of the 33 isolates from North America. Running the DAPC while limiting the number of isolates per site to 20 provided identical assignment results. Second, we used the SPASIBA model, a spatial Bayesian inference (SPASIBA) method for geospatial assignment that models the spatial frequency of alleles by a set of spatially autocorrelated random variables with Gaussian distribution. Implementation was carried out in R using the SPASIBA (v.24.6.27) and INLA (v.0.0.4) packages based on the same dataset as described above for the DAPC analysis. That is 5,162 *Pd*-1 isolates from 226 sites in Europe for the reference data ('geno.ref') and

33 isolates from 9 sites in North America as individuals of unknown geographical origin to be assigned ('geno.unknown'). For SPASIBA, we used the function 'SPASIBA.inf' with a ploidy level of 1 and a flat domain (sphere = false). To achieve a uniform spatial resolution, we configured the grid to consist of 198 pixels in longitude and 120 pixels in latitude. This setup ensured a consistent resolution across both dimensions, with each pixel representing an approximate resolution of 0.169°. To identify the most probable source of introduction inferred across the 33 North American isolates, we calculated the average assignment likelihood for each pixel across the 33 isolates (Supplementary Table 5), from which we computed the log-likelihood for representation (Fig. 4).

The visualization of EEMS (https://github.com/dipetkov/eems) was used to evaluate geographical barriers linked to patterns of gene flow[35]. This method differs from PCA in that genetic differentiation is visualized as a function of migration rates rather than through genetic or genotypic distance. This method uses a population genetic model to compare expected pairwise genetic dissimilarities in relation to their geographical distances (that is, under a model of isolation-by-distance) with observed patterns across the sampled area. Specifically, a triangular grid with specific density (number of demes) is built over the area containing geo-referenced samples. For the edge of each grid, the migration parameter is estimated by Bayesian inference and Markov chain Monte Carlo sampling, which means that migration is estimated in an approximated stepping-stone model between neighbouring grid cells. As sampling locations will fall into the same or neighbouring cells depending on the grid cell size, the number of demes (defining the overall density of grid cells and hence their size) influences the outcome of estimated migration rates (particularly at small geographical scales). For this reason, we calculated the EEMS for a range of deme sizes ($n$ = 8, 100–450 demes) in independent runs of 8 million iterations each (after a burn-in of 1 million iterations), as previously recommended[35]. Runs were combined in a single figure for visualization of robust migration rates using the R package reemsplot2 (v.0.1.0)[35]. It should be noted that estimated migration rates are most accurate closer to sampling locations and less accurate in sparsely sampled geographical areas. For this reason, in addition to clone-correction based on site identity (each genotype appears only once per site, additional occurrences are removed), the Russian isolates were excluded before calculating EEMS for both clades. This resulted in a total of 2,261 isolates used for Pd-1 and 107 isolates for Pd-2 (Supplementary Table 1). Markers Pd10 and Pd14 were uninformative for Pd-2 and hence removed for the EEMS (removed for the Pd-2 dataset only), which left a dataset with 16 markers.

Phylogenetic relationships between the 5,479 isolates (or 1,866 MLGs) were reconstructed using Nei's 'Da' genetic distance and Cavalli–Sforza and Edwards Chord distance 'Dch', as both distance measures were found to be the best performing ones to retrieve the relationships between individuals[56]. Genetic distances were then clustered using the UPGMA algorithm as implemented in Phangorn (v.2.11.1)[57] in R. Given that the topology for the nodes of interest was identical for both methods (data not shown), only the results from the Da distance are presented. Analyses were performed in R using functions implemented in hierfstat (v.0.5.11)[58]. For visualization purposes of Fig. 1b, the 'fish eye' function of FigTree (v.1.4.4.)[59] was used to zoom into the section of the tree containing the Pd-2 clade. We jackknifed loci one at a time to test for the support of both Pd-1 and Pd-2 clades monophyly using the 'is.monophyletic' function in ape (v.5.7.1)[60].

### Analyses of genomes de novo assembled from long-reads
**Base calling.** Base calling of fast5 files (from MinION sequencing) was performed on a GPU computer (hosted by the Montpellier Bioinformatics Biodiversity platform) by running the software Guppy (v.5.0.7), a state-of-the-art neural network base caller[61].

**Genome assembly.** To assemble high-quality haploid genomes of *P. destructans* (*P. destructans* is haploid), we carried out adaptor

trimming with Porechop on all base-called reads (https://github.com/rrwick/Porechop)[62]. These were then parsed into Flye (v.2.9)[25] with the --nano-hq flag. Other arguments were left as default (including automatic minimum overlaps). Genomes were polished once using pre-trimmed Illumina reads with HyPo (v.1.0.3) after initial mapping using paired-end mapper Burrows–Wheeler aligner–Maximal Exact Match, bwa-mem (v.0.7.17-r1188)[63,64]. HyPo arguments included approximate genome length of 35 megabases (-s 35m; based on a previous study[65]) and the average read depth of each genome (-c) (calculated with Samtools depth[66]).

To remove contigs that potentially resulted from contamination, contigs with exceptionally low GC content (identified using infoseq EMBOSS (v.6.6.0.0)[67]) were individually compared with the nucleotide BLAST database[68,69]. On the basis of these results, we removed contigs not originating from *P. destructans* from four isolates (which contained clear contamination by *Cellulosimicrobium cellulans*, *Penicillium solitum*, *Pyrenophora teres f. teres* and *Shiraia bambusicola*).

Mitochondrial contigs, with a characteristic lower GC content and known length (about 32 kb), were also identified through BLAST and removed from all further analyses. To remove noise in our genomes resulting from spurious assembly, all contigs below 10,000 bp were removed using SeqKit (v.0.16.1)[70] (-m 10000). Statistics of the assembled genomes are presented in Supplementary Table 11.

**Repeat annotation.** We annotated repeat content of each genome with RepeatModeler and RepeatMasker tools. Build Database was followed by RepeatModeler (v.2.0.1), which we ran individually on all 11 genomes of *P. destructans* isolates to identify repeat regions with default parameters. All novel consensus repeat sequences were combined using CD-HIT-est (v.4.8.1) to remove redundancy in the clustered library[71] (-aS 1 -c 1 -r 1 -g 1 -p 0). All final repeat sequences not annotated were then removed from the repeat library. Furthermore, BLASTN (v.2.9.0+) searches of the repeat library were carried out against the 9,405 annotated protein-coding genes of the *P. destructans* reference genome assembly downloaded from the NCBI (GCF_001641265.1_ASM164126v1). Any repeats with a sequence identity over 80% for over 80% of the query length (-outfmt 6, -perc_identity 80, manual removal of qcovs > 80) were also removed[72]. Finally, we removed duplicated fasta entries with Samtools (faidx). Masked assemblies as used downstream were produced using RepeatMasker (v.4.1.2)[73] (-xsmall) with the curated repeat library (-pa 5 -a -s -gff -no_is). The same pipeline was then used for the outgroup (Gd267), which was treated independently from *P. destructans* isolates.

**Genome annotation.** For gene annotation of all isolates, we chose to use the Funannotate pipeline (v.1.8.15) dedicated specifically for fungal genome annotation[74]. We used the pipeline through the Galaxy Europe cluster (https://usegalaxy.eu). We started by soft masking all the genome assemblies using RepeatMasker tools (v.4.1.2; https://www.repeatmasker.org/RepeatMasker/)[73] with the transposable element (TE) library generated in this study (see the section 'Repeat annotation'). We then used Illumina paired-end RNA sequencing data from European (Sequence Read Archive accessions SRP041673 and SRR1270711) and North American isolates[75] (SRP041668, SRR1270148, SRR1270408 and SRR1270412) in addition to the data from *P. destructans*-infected bats[76] (SRP055976) used for annotation of the published reference sequence[74]. All RNA sequence paired-end reads were then mapped on the soft-masked genome assemblies using the RNA STAR mapping tool (v.2.7.10b)[77] (--genomeSAindexNbases 11bp). Funannotate-predict returns gene models based on read mapping using Augustus (v.3.4.0). It also uses curated databases (UniProtKb/SwissProt databank) for proteins to help predict probable gene structures. For the initial training of predictors, Funannotate-predict also uses BUSCO (v.5.2.2)[78] for initial Augustus species training. For this step, we used the phylogenetically closest species available on the Galaxy server: Fusarium (orthoDB v.10).

We did not use the ab initio predictor dedicated to fungal genomes as the option created fragmented gene models. The results of the Funannotate-predict were combined to generate functional genome annotations using Funannotate-functional. Protein evidence generated was compared with the Funannotate database (v.2022-01-17-193541).

**BUSCO genes.** Genome assembly for each isolate was benchmarked with BUSCO (v.5.2.2) (hmmsearch v.3.1 and metaeuk v.5.34c21f2) using the option -m genome flag for the Kingdom fungi odb10 database from orthoDB (v.10). The fungi database contains 758 orthologous gene sequences[79,80]. Basic statistics of reads were obtained with NanoStat and NanoPlot (v.1.42.0)[81].

**Sequence divergence.** Sequence divergence for BUSCO genes was calculated from the MAFFT alignment (described below) in R, with the function 'dist.dna' from the ape package[60]. For each of the assemblies of full genomes, subcontigs were obtained by deleting the repeated and low-complexity sequences detected using RepeatModeler and RepeatMasker pipelines (as described above). A local alignment of the subcontigs was carried out with NUCmer4 (v.4.0.0)[82] for all the isolates against each other and interpreted using the show-coords program by applying a minimum percentage of sequence identity of 80%. The aligned regions were used to calculate the weighted average identity.

**Synteny.** To gain better insight into the similarities and differences in genomic structure and organization between clades, we performed network-based microsynteny analysis. This enabled us to investigate gene-copy number, to identify synteny conservation, to detect and quantify clade-specific syntenic clusters and to reconstruct phylogenetic relationships between isolates based on microsynteny. The detection of syntenic blocks performs best when using high-contiguity genomes assembled de novo (that is, without a reference). According to a previously published systematic evaluation[83], the characteristics of our genomes (mean N50 of 1.8 Mb; gene density estimated around 270 per Mb: 10,000 genes and 36.9 Mb per genome on average; Supplementary Table 11) should allow for a robust synteny analysis.

Using annotated genes from the 11 *P. destructans* genomes (detailed in the section 'Genome assembly'), all inter-pairwise and intra-pairwise all-vs-all protein similarity searches were conducted using DIAMOND (v.2.1.7), for which the top 5 hits were kept in searches[84]. The output generated by DIAMOND was used as input for the MCScanX algorithm[85] to perform pairwise synteny blocks detection. To verify that parameters used for MCScanX had no impact on our results, we performed microsynteny block detection (and all downstream analyses) under 25 different parameter settings (Supplementary Table 17). These parameter settings involved the two key parameters of MCScanX, namely, the minimum required number of genes (anchors) to call a syntenic block (-s, MACH_SIZE: 10, 15, 20, 25, 30) and the number of upstream and downstream genes to search for anchors (-m, MAX_GAPS: 15, 20, 25, 30, 35). All the syntenic genes identified in the syntenic blocks were used to build a microsynteny network using the Infomap algorithm (implemented in the infer_syntenet function in the R package Syntenet (v.1.8.1)[86], which has been demonstrated to be the best clustering method available for synteny networks[87]. In the synteny network, genes represent nodes, and syntenic relationships between genes are represented by edges connecting the nodes. A median average clustering coefficient of 0.95 was obtained across parameter settings (min–max: 0.94–0.95%; Supplementary Table 17), which revealed that genes tend to form a complete subgraph or cluster with their syntenic neighbours across multiple genomes. Phylogenomic profiling was then performed using the inferred syntenic clusters, which resulted in a matrix $m_{ij}$ of the phylogenomic profiles, which represented the number of genes from cluster $j$ that can be found in genome $i$. To reveal synteny clusters that were conserved and clade-specific clusters, clusters were visualized as a heatmap with clusters clustered using Ward's clustering on a matrix of Euclidean distance (Fig. 2e). The phylogenetic signal present in the synteny network was used to infer a microsynteny-based phylogeny of the 11 genomes. The binarized matrix of phylogenomic profiles was used in IQTREE2 (v.2.0.6) applying the MK + FO + R model with node support evaluated by two methods: 1,000 bootstrap replicates and 1,000 replicates for the SH-like approximate likelihood ratio test. The mid-point method was applied to root the tree using the 'midpoint' function in the 'phangorn' package[57] and the monophyly for the clades *Pd*-1 and *Pd*-2 was evaluated using the 'AssessMonophyly' function in 'MonoPhy' (v.1.3.2)[88].

## Analyses of Illumina reads (individually tagged isolates)

**Data checking and mapping.** Illumina sequences were available for 18 isolates (listed in Supplementary Table 10). Fastp (v.0.23.4) was used to remove bases with a phred quality value lower than 30 ('-q 30'). Subsequently, bwa-mem2 (v.2.2.1) was used to align the Illumina reads to each of the two reference genomes (Gd293 for *Pd*-1 and Gd45 for *Pd*-2). Samtools view (v.1.16.1) was used to only keep reads with mapped (-F 0×4), properly paired ('-f 0×2'), and high-confidence mapping quality (MAPQ values of 60: '-q 60'). Samtools sort and Samtools index were used to respectively sort and index the obtained BAM files. Reads were assigned using Picard (v.2.27.1) and the AddOrReplaceReadGroups tool. The SortSam tool (from Picard) was used to sort the BAM file by queryname (QNAME) before removing duplicate reads using the MarkDuplicates tool (from Picard). SortSam was subsequently used to sort the BAM file by coordinate before indexing using Samtools index.

**Base calling, genotyping and filtering.** We then performed base calling with 'gatk HaplotypeCaller' (gatk v.4.2.6.1) using the '-ERC BP_RESOLUTION' mode and a ploidy of 1. Genotyping was done with 'gatk GenotypeGVCFs'. 'gatk SelectVariants' was then used to extract the non-variable positions in one file and the variable positions (SNP only) in another file (insertions and deletions were discarded). Hard filtering of variable positions that did not meet the criteria 'QUAL < 30.0', 'QUAL/DP < 2.0', 'SOR > 3.0', 'FS > 60.0', 'DP < 20.0' and 'MQ < 40.0' was carried out using 'gatk VariantFiltration' and 'gatk SelectVariants'. Isolate genotypes with a read depth lower than 20× at a given position were set to no-call. 'gatk SortVcf' was then used to merge the filtered variable and non-variable positions into a single VCF file. 'bedtools subtract' was subsequently used to remove masked positions from the reference genome and positions called as heterozygous when considering the isolates as diploid (see the section 'Filtering repetitive DNA').

For each contig of each of the two reference genomes (Gd293 and Gd45), we then checked for the mean read depth, when mapping the 18 isolates to each reference genome, to detect unusually high read depth that would indicate the presence of repetitive sequences in the regions that nevertheless passed our filters described above. For each contig, we also checked for the average number of isolates with missing data per position to identify contigs with high levels of missing data. For Gd45, out of eight contigs that were smaller than 50 kb, two had no reads mapped, and five out of the six remaining had unusual mean read depth (lower or higher than other contigs) and/or a high average number of isolates with missing data (Supplementary Table 12). Therefore, all eight contigs below 50 kb, representing in total only 0.46% of the Gd45 genome were excluded from the analyses. For Gd293, all contigs were larger than 50 kb and a single contig (contig_44) deviated from the values observed for other contigs by having a read depth about 100× higher than other contigs and having on average 13 isolates with missing data per position (Supplementary Table 12). This contig, most likely an accessory chromosome with a high proportion of repetitive sequences, was therefore excluded from further analyses. After all filtering steps were performed (Supplementary Table 13), the average read depths were 237 and 240 when mapping on Gd293 and Gd45, respectively. The correlation between the read depths of the 18 genomes when mapped on Gd293 and Gd45 was 0.996 (Spearman's

rank correlation $\rho$; $P < 0.001$), which highlighted that the mapping worked equally well on either reference genome.

On average, each isolate had a genotype for over 95% of positions (Supplementary Table 13). In the end, we obtained data on 21,609,224 positions when using Gd293 as the reference (including 92,593 biallelic SNPs) and 22,041,192 positions when using Gd45 as the reference (including 95,638 biallelic SNPs). These data were stored in a single VCF file (with 18 samples) per reference genome. Furthermore, to evaluate the quality of our full procedure, we checked the number of differences obtained by mapping the Illumina reads of Gd293 to the de novo-assembled MinION genome (reference genome Gd293, see the section 'Genome assembly' for assembly details). We did a similar analysis for Gd45 mapped to the reference genome Gd45. For Gd293, the number of differences with the reference genome (itself) was 19 variants out of 23,660,591 positions in the final VCF file, which indicated a combined mapping, base calling and genotyping error rate of $8.030231 \times 10^{-7}$; that is 1 error every 1,245,294 bp, which corresponded to a quality score of Q60.95. For Gd45 (mapped on the reference genome Gd45), the error rate was also extremely low: $2.444712 \times 10^{-6}$ (54 variants out of 22,088,490 bp), or 1 error every 409,046 bp (Q56.11). These data provide evidence that our pipeline, which incorporated several stringent filters, recovered highly reliable SNPs while preserving a large amount of data (>22 Mb accounting for around 60% of the reference genomes).

**Diversity and differentiation calculation.** Genetic diversity π was calculated per clade (11 for *Pd*-1 and 7 for *Pd*-2) across 50 kb windows using pixy (v.1.2.7.beta1), which takes into account missing data in calculations and hence provides unbiased estimates[89]. The index of genetic differentiation ($F_{ST}$; Weir and Cockerham's method) between the two clades was calculated using a modified version of vcftools (https://github.com/jydu/vcftools) to allow the computation of statistics with haploid data.

**Phylogenetic relationships.** The relationships between the 18 *P. destructans* isolates (Supplementary Table 10) and the *Pseudogymnoascus* sp. outgroup (isolate 267) were reconstructed using maximum likelihood in IQ-TREE2 (v.2.0.6)[90]. The relationships were reconstructed when considering two methods to partition the genome: (1) BUSCO genes alone, and (2) 10 kb windows across the whole genome. Each of these partition methods were used in combination with each of the two reference genomes (Gd293 and Gd45; see the section 'Data checking and mapping'), which resulted in a total of four datasets. All four datasets converged towards the same topology, recovering *Pd*-1 and *Pd*-2 as two reciprocally monophyletic clades (Fig. 2b and Supplementary Fig. 5).

**BUSCO phylogeny.** To produce the phylogenetic tree using newly sequenced and publicly available BUSCO genes of the isolates, we used all single copy, complete orthologues as identified from the fungi odb10 database, that were common to all 18 isolates (Supplementary Table 11) and the *Pseudogymnoascus* sp. outgroup (isolate Gd267). This resulted in 664 and 662 BUSCO genes when mapping on Gd293 and Gd45, respectively. We extracted the genes from each assembly with BEDTools (v.2.30.0-8)[91] using the command getfasta before individually aligning with MAFFT (v.7.453)[92] (--auto, --adjustdirection). In IQ-TREE2, a concatenation-based species tree with edge-linked proportional partition model with 1,000 ultrafast bootstrap (-B 1000 -T AUTO) was produced[93]. To produce the species concordance factor, which measures how consistent the genealogical relationships are across different loci, the orthologue and species trees were used (--scf 100 --prefix concord -T 10) by IQ-TREE2. The final tree was manually rooted at the outgroup, Gd267, in FigTree (v.1.4.4; http://tree.bio.ed.ac.uk/software/figtree/). To represent conflicting phylogenetic signals between gene trees (for example, owing to recombination and/or incomplete lineage sorting),

we used the function consensusNet from the phangorn R package[94] and computed the consensus network from the splits occurring in the different gene trees. Only splits occurring in at least 10% of trees were represented in the network.

**Whole-genome phylogeny.** The relationship between the 18 *P. destructans* isolates (Supplementary Table 10) and the *Pseudogymnoascus* sp. outgroup (isolate 267) was reconstructed using maximum likelihood in IQ-TREE2 (v.2.0.4)[90]. We used vcftools with the 'missing-site' function to extract missingness on a per-site basis from the VCF file containing the genotypes of the 19 (18 + 1) individuals (see the section 'Base calling, genotyping and filtering'). Then, we calculated the number of positions with no missing data over 10 kb non-overlapping windows and only kept windows with at least 5,000 positions with no missing data across the 19 isolates. For each of the 19 isolates, we then used BEDTools getfasta to extract sequences (FASTA format) from these windows. In each window, only positions without missing data (that is, minimum 5 kb) were kept. We obtained a total of 1,275 and 1,288 genomic partitions when using the reference genomes from each clade, Gd293 and Gd45, respectively. A concatenation-based species tree with edge-linked proportional partition model with 1,000 ultrafast bootstrap (-B 1000 -T AUTO) was then produced. This procedure was applied in parallel to the data mapped on reference genomes from each clade, Gd293 and Gd45 (VCF files).

**Analyses of recombination.** To test for the presence of recombination in each of the two clades, we used two tests, the pairwise homoplasy index (PHI or Φw) test and the four-gamete test (FGT). The Φw test calculates a pairwise similarity index between closely linked sites (situated in $w$ bases) and compares observed values to values obtained after permutation of the sites. Under the null hypothesis of no recombination, the genealogical correlation between adjacent sites remains unchanged even if the order of the sites is shuffled. This is because all sites share the same evolutionary history when recombination is absent. However, when recombination occurs, the order of the sites becomes important as distant sites tend to have weaker genealogical correlations compared with adjacent sites[95]. We used the Φw test as implemented in Splitstree CE (v.6.3.27)[96] to test the null hypothesis of clonality. The FGT test involves counting the number of allelic combinations between any pair of SNPs on the same contig. Assuming an infinite site model, the observation of four combinations is incompatible with the absence of recombination. For example, if two SNPs have the alleles C/T and A/G, then the observation of the haplotypes CA, CG, TA and TG must be the result of a recombination event. We calculated the number of haplotypes using the script FGT.pl (https://github.com/dbsloan/fgt). In each clade, we performed FGT on windows of 100 kb, with at least 20 kb of unmasked nucleotide positions and no missing data. We performed these analyses on the genomic data (Illumina data from individually tagged isolates) from the 18 isolates (11 from *Pd*-1 and 7 from *Pd*-2; described in section 'Analyses of Illumina reads (individually tagged isolates)') mapped on the reference genomes (Gd293 and Gd45) of *Pd*-1 and *Pd*-2 clades, respectively.

Next, we estimated the population recombination rate ($r = 2N_e r$, where $r$ is the recombination rate per bp per generation and $N_e$ is the effective population size) using LDhelmet software[97] (v.1.10; https://github.com/popgenmethods/LDhelmet). We used the parameters -t 0.0005 and -t 0.001 for the population mutation rate ($q = 2N_e m$, where $m$ is the mutation rate per bp per generation) for *Pd*-1 and *Pd*-2, respectively, -w 200, and we estimated the mutation transition matrix using the IQTREE2 substitution model GTR[98]. Default values were used for other parameters following the LDhelmet manual.

Apart from formal recombination tests carried out as explained above, we used the reconstructed phylogenetic tree obtained with microsatellite data (described in the section 'Analyses of multilocus genotypes') to map the two mating types. This analysis demonstrated the presence of both mating types in *Pd*-1 and *Pd*-2 clades, and the presence

of both mating types throughout the phylogenetic tree depicting relationships between MLGs in each clade. These data provide further evidence for recombination in *Pd*-1 and *Pd*-2 clades (Supplementary Fig. 1).

**Modelling demographic history.** To model the evolutionary history of both clades, we implemented Approximate Bayesian Computation comparing two demographic models with and without contemporaneous migration. Specifically, we evaluated the isolation-migration model (IM) and the strict-isolation model (SI). These two models imply a population split at a time $T_{split}$ in the past, but this event is followed by constant migration in the IM model or no migration in the SI model. The ancestral population and the derived population have an independent population size ($N_a$, $N_1$ and $N_2$). Population size is also free to vary within population 1 and 2 independently once at time $T_{dem1}$ and $T_{dem2}$ in the past for population 1 and 2, respectively. Coalescence simulations were generated using msnsam (October 2007 version)[99], which is a modified version of ms[100]. The priors for population size, $T_{split}$ and migration rate were generated using uniform distributions. Priors and summary statistics were computed using scripts taken from DILS[101] (https://github.com/popgenomics/DILS_web). One million simulations were performed for both models using a mutation rate of $1.5 \times 10^{-9}$ mutation per site per generation, a recombination rate half of the mutation rate, a $T_{split}$ between 10 and 10 million generations and an $N$ between 10 and 10 million individuals per population. As performing the analyses on the full genome would be computationally too intensive (for the simulations part), we instead used a representative sample across the genome. To accomplish this, we selected 1 kb sequence windows, spaced 10 kb apart. This resulted in a dataset of 4,832 and 2,881 windows for the datasets when using *Pd*-1 and *Pd*-2 reference genomes, respectively.

Model selection was performed using the random forest method implemented in the R package abcrf (v.1.9)[102] and using the postPr function of the R package abc (v.2.2.1)[103]. Parameter estimations were obtained using the neural network method implemented in the abc package using 5,000 simulations. Finally, we computed a goodness-of-fit statistic for each of the summary statistics as the proportion of simulations of the posterior distributions with a statistic superior or inferior to the observed statistic with this proportion *P* always smaller than 0.5.

The model selection procedure led to the selection of the SI model, with posterior probabilities of 0.77 and 0.75 with the random forest method for references Gd293 and Gd45, respectively. The rejection method led to the selection of the SI model 79% and 76% of the times for reference Gd293 and Gd45, respectively. Therefore, both model-selection methods support the model without contemporaneous migration. The full set of parameters estimated from the posterior distributions of the SI model using the neural network method is presented in Supplementary Table 16. Based on the SI model, the divergence time between *Pd*-1 and *Pd*-2 clades was estimated to have occurred between 114,000 and 1.5 generations ago (Supplementary Table 16).

## Analyses of Illumina reads (Pool-seq data)
**Data checking and mapping.** We checked and mapped the Illumina data (each of the four pools per clade) as detailed in the section 'Data checking and mapping' for Illumina reads above.

**Data pooling.** For each of *Pd*-1 and *Pd*-2, we had four pools (hereafter called 'subpools') of isolates (with 17, 18, 17 and 17 isolates for *Pd*-1 and 16, 16, 16 and 15 isolates for *Pd*-2). Before combining the four subpools per clade, we calculated the number of reads per subpool and subsampled them with Samtools view to have the same number of reads per isolate (1,704,260 reads) in each subpool. We then used Samtools merge to merge BAM files for the four subpools per clade, which resulted in one final pool (pool of four subpools) of 69 isolates for

*Pd*-1 (117,589,122 reads) and one final pool (pool of four subpools) of 63 isolates for *Pd*-2 (107,355,848 reads).

**Genotyping.** Genotyping was carried out separately for each pool (one for *Pd*-1 and one for *Pd*-2), mapped on the reference genome from clade 1 and 2, which resulted in four separate analyses. Samtools mpileup was then used to generate pileup outputs from BAM files. For SNP calling, we applied a Bayesian approach specifically designed for pools (SNAPE-pooled[104,105]) that calculates the posterior probability that a site is polymorphic. This approach has been validated using both simulations and empirical approaches and is among the best performing method currently available[104]. SNAPE-pooled prior parameters were sets as follows: $\theta = 0.0005$, $D = 0.0025$ or $D = 0.0005$ when mapping the pool on the reference from the different versus same clade respectively, prior = 'informative', fold = 'folded', and nchr = number of (haploid) isolates in the pool. We converted the SNAPE-pooled output file to a VCF and as recommended by the program developers, we only considered positions with a posterior probability ≥0.9 as being polymorphic. In all other cases, positions were marked as monomorphic. To avoid including sequencing errors as rare alleles, we adopted two complementary strategies. First, when converting the SNAPE-pooled output to a VCF, we only included alleles that were supported by at least five reads in the pool (that is, the four subpools together). Second, we performed SNAPE-pooled analyses on each subpool (with parameters as described above) to identify alleles that were not supported by at least two reads in two subpools. These alleles were then removed from the VCF file of the pool.

**Data filtering.** To limit false-positive SNPs, we applied further filters to the VCF file to remove the following: (1) regions where repetitive DNA elements were either confirmed or suspected; (2) regions with low or high read depth (based on the Pool-seq data); and (3) regions that were identified as problematic based on read depth of individually tagged isolates (see the section 'Base calling, genotyping and filtering' of Illumina reads).

**Filtering repetitive DNA.** The removal of regions with confirmed or suspected repetitive DNA (including paralogues) that could not be confidently mapped with short-read data was carried out using three complementary approaches.

First, we removed regions that were masked from the reference genome (see the section 'Repeat annotation').

Second, we removed regions suspected to contain hidden repetitive DNA. Although we removed regions that were masked from the reference genome (see above), a single or a set of genomes (18 in our case, see section 'Repeat annotation') is unlikely to harbour the full extent of the repertoire existing in repetitive DNA in a larger number of isolates such as in the Pool-seq dataset. Hidden repetitive DNA can generate spurious heterozygous genotypes that can confound the estimation of genetic differentiation between populations or species (for example, ref. 106). Hence, when using reads from an isolate that was not used to build the repeat library or the filters (for example, Pool-seq data), the unique mapping of its reads to a reference genome does not per se confirm that these reads originated from non-duplicated regions. We used levels of genetic diversity (π) as a proxy to detect hidden repetitive DNA. Repetitive DNA elements sharing a common ancestor accumulate mutations; therefore, when such loci (two or more copies) are erroneously merged together as a single locus, one expects higher levels of genetic diversity provided that everything else remains equal. We therefore removed regions with levels of π greater than 1%. This threshold was obtained by calculating the level of genetic diversity observed across 18 genomes rather than pools of individuals (see the section 'Analyses of Illumina reads (individually tagged isolates)'; Supplementary Table 10) for which only 0.43% and 1.43% of sites for *Pd*-1 and *Pd*-2, respectively, showed π values greater than 1%.

π was calculated per site using pixy (v.1.2.7.beta1), which takes into account missing data in calculations and hence provides unbiased estimates[89]. Site-based calculations from pixy were then averaged in R (function 'runner') over 100 bp sliding windows by summing the raw counts and recomputing the differences/comparisons ratios (https://pixy.readthedocs.io/en/latest/output.html). For the Pool-seq data, genetic diversity (named 'Q1'; see refs. 107,108) was calculated using the function 'computeFST' in the poolfstat (v.2.0.0) package in R. Sites included in sliding windows with genetic diversity greater than 1% were stored in a BED file. Bedops (v.2.4.41) was used to flatten all disjoint, overlapping and adjoining element regions into contiguous, disjoint regions. BEDTools (v.2.30.0-8) merge was used to merge features that were separated by 500 bp maximum.

Third, using the same rationale as the second filter detailed above but with a different approach, we took advantage of the nature of the data (*P. destructans* isolates are haploid) to identify and exclude regions of the genome where mapping would lead to the calling of a heterozygone when performing the analyses of a haploid isolate in diploid mode[109]. Each of the 18 isolates (as listed in Supplementary Table 10 (except the outgroup)) were mapped against the reference genome (as per the section 'Data checking and mapping' for Illumina reads) and analysed in diploid mode (that is, considering that the isolate is diploid). For this, we used the same pipeline as for base calling isolates (with gatk; see the section 'Analyses of Illumina reads (individually tagged isolates)') but with ploidy of 2. Sites scored as heterozygote in at least one isolate were recorded in a BED file. A total of 183,756 heterozygote sites (spread across contigs) were identified in those 18 genomes when using Gd293 as the reference and 177,785 when using Gd45 as the reference. All those heterozygote sites were removed from all 18 genomes, irrespective the isolates they originated from. Although these data alone might indicate that *P. destructans* is diploid, results from the microsatellite analysis firmly refuted this hypothesis. Indeed, we genotyped 5,479 isolates originating from single spore (that is, monosporic isolation) cultures for 18 microsatellite loci (see the section 'Cultures and genotyping'), and out of the 98,622 genotypes (5,479 × 18), we never encountered 2 alleles per locus for any single spore isolate. Two alleles for some loci and some isolates would be expected if *P. destructans* was diploid. Furthermore, such levels of heterozygosity have already been reported when base calling haploids in diploid mode in other fungal species (for example, ref. 109).

The BED files created for the three complementary approaches detailed above were then processed in Bedops to flatten all disjoint, overlapping and adjoining element regions into contiguous, disjoint regions and BEDTools merge was used to merge features that were separated by 500 bp maximum. These regions contained in the final BED file were then excluded from the VCF file in BEDTools substract.

**Filtering based on read depth.** Many species of fungi have accessory chromosomes, and the results from the section 'Base calling, genotyping and filtering' provide strong evidence that this is also the case for both of the *P. destructans* clades. As a result, stringent filtering based on read depth alone was not possible. We therefore only performed light filtering by setting sites with a read depth below 20× or above 600× per pool as missing data. Based on read depth and levels of missing data from individually tagged isolates (section 'Data checking and mapping' for Illumina reads; Supplementary Table 12), we identified a few problematic contigs that were also filtered out from the Pool-seq data.

**Filtering outcomes.** The filtering steps described in the previous two sections resulted in a narrowing of the 95% highest density interval (hdi95) of the read depth for both clades when mapped to the reference genome of both clades (Supplementary Fig. 6). When using reference genome Gd45, *Pd*-1 hdi95 narrowed down from 5–529 to 234–560 whereas *Pd*-2 hdi95 narrowed down from 124–470 to 217–470. When using reference genome Gd293, *Pd*-1 hdi95 narrowed down from

92–512 to 236–523 whereas *Pd*-2 hdi95 narrowed down from 5–511 to 227–542. Positions filtered out were mostly, although not exclusively, positions with read depth less than 200 (Supplementary Fig. 6), which meant that accessory chromosomes constitute a substantial part of the data filtered out. This result was expected, as accessory chromosomes are known to be repeat-rich[110], hence they are expected to be more heavily filtered out than core chromosomes. After filtering, the dataset consisted of 17,110,071 and 16,733,801 positions for *Pd*-1 and *Pd*-2 pools, respectively, mapped to Gd293, and 16,675,338 and 17,286,690 positions for *Pd*-1 and *Pd*-2 pools, respectively, mapped to Gd45. The median read depths were 446 and 382 for *Pd*-1 and *Pd*-2, respectively, mapped on Gd45. The median read depths were 425 and 437 for *Pd*-1 and *Pd*-2, respectively, mapped on Gd293. A similar number of SNPs (55,919 and 57,330) was identified when mapping the pools on Gd293 and Gd45, respectively.

**Calculation of the index of differentiation $F_{ST}$.** The VCF file was then imported into R using poolfstat (v.2.0.0)[108] and the vcf2pooldata function (designating the pool size as 69 and 63 for clade *Pd*-1 and *Pd*-2, respectively, max.cov.per.pool = 600, min.cov.per.pool = 20). The multilocus $F_{ST}$ was then calculated across 200 SNPs with the compute.FST function.

**Variations in growth rates and growth-medium colouration.** To evaluate variations in culture-related properties, a subset of isolates was photographed in a custom-built photobox using the same camera, lens and settings (Canon EOS 600D with 60 mm Canon Macro lens EF-S, shutter speed = 1/6,000, aperture = 3.2, ISO = 400, evaluative metering; see ref. 111), biweekly for 8 weeks. For this purpose, 45 and 34 isolates of *Pd*-1 and *Pd*-2, respectively, were re-cultured on the same day on the same batch of culture medium. Six days later, for each isolate, a single germinating spore was physically moved to a new plate with growth medium (detailed in the section 'Cultures and genotyping') and stored upside down at 10 °C. The identity of isolates is provided in Supplementary Table 1.

**Analysis of growth.** The analysis of the pictures was carried out in R (v.4.1.1)[52] using the packages EBImage (v.4.3)[112], ks (v.1.14.1)[113], adehabitatHR (v.0.4.21)[114,115], sp (v.1.4-6)[116] and adimpro (v.0.9.6)[117]. Images were imported into R using the readImage function that extracts the intensity of each pixel of the red, green and blue channels. Pixels with red intensity above 0.7 were characteristic of cultures, whereas the background (culture medium) was below 0.7. Coordinates (that is, their position in the image) of pixels with red intensity above 0.7 were stored and used to calculate the minimum convex polygon (MCP) using the mcp function of the adehabitatHR package. The MCP was used to outline the edge of the fungal growth and to calculate the number of pixels it covered. To overcome potential issues with dark non-fungal material present on the plate (for example, droplets of water or dirt) that would be included in 100% MCPs and hence artificially increase culture size, we calculated 11 MCPs per isolate, excluding 5–15% of outliers in steps of 1 (11 values; that is, MCP 85 until 95%). We then extrapolated the number of pixels covered by the fungus (for example, for an isolate covering 180 pixels calculated with an MCP 90%: 180/90 × 100 = 200 pixels). For each MCP, pixels were finally transformed to square centimetres with the use of cross multiplication in relation to an object of known size (5.207681 cm²). For each isolate, the average across the 11 MCPs was used, and the quality of the estimates was evaluated using the standard deviation of the estimate made for the 11 MCPs and visual inspection of the edge of the MCPs depicted on top of the original picture. Resulting culture sizes are visualized in Extended Data Fig. 4 and Supplementary Table 9.

**Analysis of culture darkness.** The colouration of culture medium as a result of culture growth was measured from the same pictures taken for

the analysis of growth (see the section 'Analysis of growth'; 45 isolates for *Pd*-1 and 34 isolates for *Pd*-2). Analyses were carried out in R using EBImage (v.4.3)[112]. Colour images were transformed into black and white images, and the pixel density (a value from 0 to 1; 1 being white and 0 being black) was recorded in 3 rectangles distributed on the area showing the culture medium (that is, not touching the edge of the culture and also avoiding the centre with fungal growth). The median pixel density among the three rectangles was used as a proxy for culture darkness. For each isolate, the difference in pixel density between the picture taken after 1 week and that taken after 8 weeks was calculated (8 weeks – 1 week, resulting in a positive value if darkness increased, a negative value when darkness decreased). Results are presented in Extended Data Fig. 2 and Supplementary Table 8.

## Maps and plotting

Unless otherwise stated, figures were produced in R using functions from base R[52] and ggplot2 (v.3.5.0)[118]. Maps (Figs. 1a and 2a) were downloaded as tiles from Stadia Maps (https://stadiamaps.com/) with data by OpenStreetMap (map tiles by Stamen Design under ODbL, under CC BY 4.0) and plotted using the ggmap package (v.3.0.2)[119]. They represent maps of type 'stamen_terrain_background', with colours representing natural vegetation colours and elevation (through shading). Maps for Figs. 3a and 4a,b were obtained from the R packages rworldmap (v.1.3.8)[120] and rworldxtra (v.1.0.1)[121]. Bat species distribution were recovered from the International Union for Conservation of Nature (IUCN) website[122] as shape files (https://www.iucnredlist.org/) and plotted in R. Inkscape[123] (v1.1.1; https://inkscape.org) was used to optimize visualizations.

## Statistical analyses

We tested the relationship between clade identity (*Pd*-1 or *Pd*-2; binary response) and environmental factors, including bat species (nominal variable), latitude and longitude (both numerical variables were scaled). For this analysis, closely related bat species that are challenging to identify during hibernation were pooled together. Furthermore, to improve model convergence and to ensure identifiability of the model, only the most commonly infected species, that is, species from which we isolated *Pd* at ten or more sites were included in the model. This resulted in a dataset comprising 4,295 isolates from 231 sites in Europe, broken down as follows: *M. mystacinus* (28 isolates, 10 sites); *M. dasycneme* (107 isolates, 11 sites); *M. daubentonii* (111 isolates, 17 sites); *Myotis* species complex (322 isolates, 38 sites); and *M. myotis*/*blythii* (3,727 isolates, 187 sites). We fit a Bayesian hierarchical model using clade identity as our response variable and a logit link function. Bat species, latitude and longitude were included as population-level effects (analogous to a fixed effect in a frequentist approach), whereas samples (nested within sites) and sites were included as group effect (analogous to a random effect in a frequentist approach) with random intercept. The group-level effects were used to account for variations between sites and between samples within site. Given that both *P. destructans* clades were found in roughly equal numbers in *M. mystacinus* (13 and 15 isolates for *Pd*-1 and *Pd*-2, respectively), this bat species was used as the baseline category. To estimate the model parameters and to perform Bayesian inference, the model was fitted using 8 chains with 10,000 iterations each. To improve convergence, the 'adapt_delta' parameter, controlling the acceptance rate of the algorithm, was set to 0.99. The first 1,000 iterations of each chain were discarded as warm-up (burn-in) to ensure convergence. Chains were sampled using the NUTS (No-U-Turn Sampler) algorithm in Stan (https://mc-stan. org/) with the brms (v. 2.20.3)[124] package in R. Effective sample size measures (Bulk_ESS and Tail_ESS) were calculated to assess the quality of the draws, and the potential scale reduction factor (Rhat) was used to evaluate the convergence of the chains as previously proposed[125]. Collinearity among the explanatory variables was assessed using the generalized variance inflation factor, computed through the 'check_collinearity' function available in the 'performance' (v.0.12.4)[126] package in R.

## Analyses of barcoding genes

We investigated two universal barcoding genes that are single copy genes, the translation elongation factor 1α (*TEF1*) and the DNA-directed RNA polymerase II subunit B (*RPB2*)[127] (RPB2). We retrieved sequences of these two genes from the full genomes of 18 isolates (based on Illumina read mapped on reference genome Gd293; see the section 'Analyses of Illumina reads (individually tagged isolates)') and from the filtered Pool-seq datasets of each *P. destructans* clade, *Pd*-1 and *Pd*-2 containing 69 and 63 isolates, respectively (also mapped on Gd293; see the section 'Analyses of Illumina reads (Pool-seq data)'). This was done to search for fixed positions that could discriminate the *P. destructans* clades.

Based on sequences from 150 isolates (69 and 63 from Pool-seq data; 18 from full genomes), we identified six substitutions in *TEF1* (at positions 3155612, 3155783, 3156573, 3157304, 3157318 and 3157602 on contig 34 of Gd293) and three in *RPB2* (at positions 523497, 525708 and 526300 on contig 34 of Gd293) that fully discriminated *Pd*-1 and *Pd*-2 clades. These nine substitutions were fixed in clades. Based on the sequences from the 18 isolates mentioned above and the outgroup Gd267, both clades formed monophyletic groups when building a phylogenetic tree for each gene separately (data not shown).

Based on these discriminating sites in *TEF1* and *RPB2*, we searched in published nucleotide sequences in NCBI for *P. destructans* to classify isolates to either clade *Pd*-1 or *Pd*-2. Using this approach, a set of isolates from the Czech Republic (*n* = 3), Portugal (*n* = 13) and South Korea (*n* = 2) could be identified as belonging to clade *Pd*-2 (Supplementary Table 14), for example. These data combined with the data presented in the main text confirmed that *Pd*-1 and *Pd*-2 co-occur in Europe, but thus far, only *Pd*-2 has been found in East Asia (one isolate from China, one from Mongolia, two from South Korea). This result suggests that *Pd*-1 is rarer in East Asia or perhaps even absent.

## Reporting summary

Further information on research design is available in the Nature Portfolio Reporting Summary linked to this article.

## Data availability

The genomic sequences and assembled genomes have been deposited into the Sequence Read Archive under accession numbers SRR30476767–SRR30476787 and SRR30476795–SRR30476796 (Supplementary Table 10), and the Pool-seq data are available under accession numbers SRR30476788–SRR30476794 and SRR30476766 (Supplementary Table 15). These data can be accessed through BioProject PRJNA862744 at the National Center for Biotechnology Information. Metadata along with microsatellite genotypes data are provided in Supplementary Table 1. Temperature, absolute humidity and *M. daubentonii* presence is provided in Supplementary Table 3. Colouration of agar medium is provided in Supplementary Table 8, and colony expansion rates in Supplementary Table 9. All other data are available in the article or the Supplementary information.

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

**Acknowledgements** We thank the 366 volunteers, whose names are presented in Supplementary Note 2, who spent a considerable amount of time to survey and sample for this project; F. Delsuc and R. Allio for their initial help with MinION sequencing and base calling, I. Römer, S. Fregin and A. Magdeleine for help in the laboratory; G. Kerth for providing access to laboratory facilities; M. Bekaert for initial analyses with Illumina sequencing; E. Douzery for discussion about phylogenetic reconstruction; G. Wibbelt, D. Lindner, A. Kubátová and

A. Barlow for providing us with 29, 18, 12 and 2 cultures, respectively; N. Galtier, P. Gladieux and J. Taylor for comments on the manuscript; C. Smadja and P.-A. Gagnaire for discussions on speciation and recombination respectively; and B. Ridush, T. Ermakova and V. Rogozhnikov for information about caving in Podillia. This research was funded, in whole or in part, by Bat Conservation International, Institut Universitaire de France (IUF), Agence Nationale de la Recherche (FunAdapt), Deutsche Forschungsgemeinschaft (PU 527/2-1), National Geographic Society (WW201ER-17), Bulgarian Ministry of Education and Science under the National Research Programme "Young scientists and postdoctoral students" (DCM 577/17.08.2018), Academy of Finland (331515) and the Kone Foundation (201710231).

**Author contributions** Collected samples or data in the field: A.S.B., S.E.D., M.F., V.Z., A.T.B. and S.J.P. Cultured samples: N.M.F., A.S.B., S.E.D., M.F., V.Z., R.-M.S. and S.J.P. Expansion rate and darkness experiment: N.M.F. and S.J.P. Performed molecular work and sequencing: N.M.F., A.S.B., M.-K.T., R.-M.S. and S.J.P. Genome assembly and annotation: I.D., A.-S.F.-L. and S.J.P. Genomic analyses: I.D., B.N., C.C., A.-S.F.-L. and S.J.P. Analyses of microsatellites: N.M.F. and S.J.P. Statistical analyses: N.M.F. and S.J.P. Methodology: N.M.F., A.-S.F.-L. and S.J.P. Validation: N.M.F. and S.J.P. Resources: A.-S.F.-L. and S.J.P. Data curation: N.M.F. and S.J.P. Writing original draft: N.M.F. and S.J.P. Writing, review and editing: N.M.F., I.D., B.N., C.C., A.S.B., S.E.D., M.F., V.Z., R.-M.S., M.-K.T., A.-T.B., A.-S.F.-L. and S.J.P. Visualization: N.M.F. and S.J.P. Supervision: A.-S.F.-L. and S.J.P. Project administration: S.J.P. Funding acquisition: S.J.P. Conceptualization: S.J.P.

**Funding** Open access funding provided by Universität Greifswald.

**Competing interests** The authors declare no competing interests.

## Additional information
**Correspondence and requests for materials** should be addressed to Sebastien J. Puechmaille.

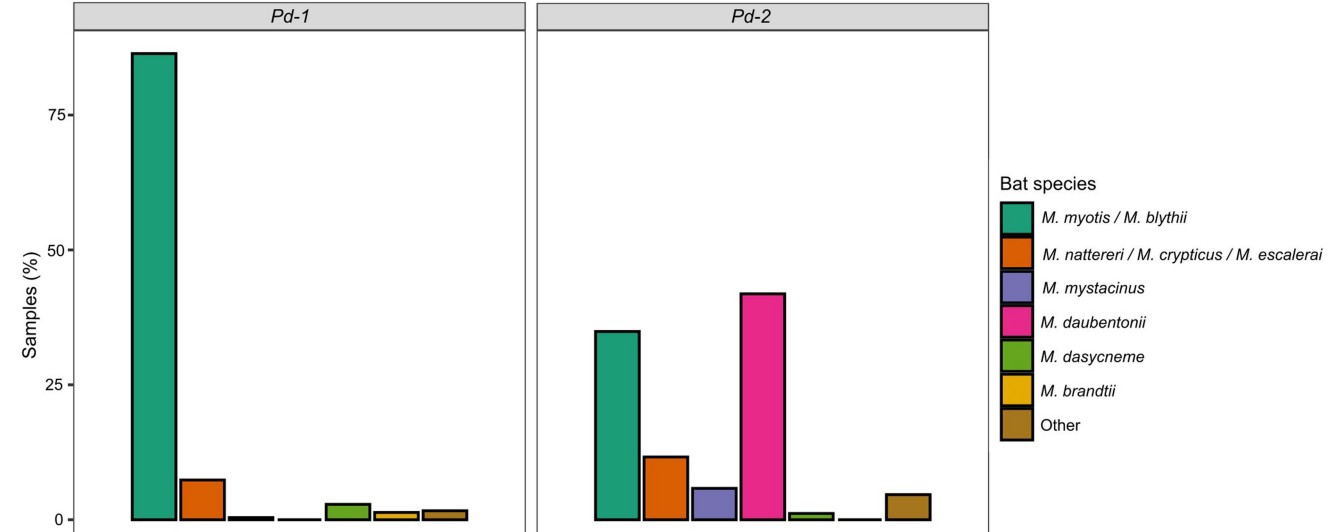

**Extended Data Fig. 1 | Percentages of swab samples collected from the 6 most frequently sampled bat species or combination of bat species.** (*Myotis myotis/M. blythii, Myotis nattereri/M. crypticus/M. escalerai, M. mystacinus, Myotis daubentonii, Myotis dasycneme, M. brandtii*) and all other species or combination of species combined ("Other") per clade (Eurasian sites only). Morphologically cryptic/highly similar species were treated together due to the difficulty of reliable species identification during winter hibernation when bats are not handled to minimise disturbance. Samples from substrates other than bats (*n* = 267) or without bat species information (*n* = 1) were not included in this figure, resulting in data from 1,388 and 92 swabs for *Pd*-1 and *Pd*-2, respectively. Note that 17 swabs out of 1,463 harboured isolates from both clades and are thus used to calculate percentages in both graphs. See the section 'Statistical analyses' in the Methods for statistical analyses formally testing the relationship between clade identity (*Pd*-1 or *Pd*-2) and environmental factors, including bat species.

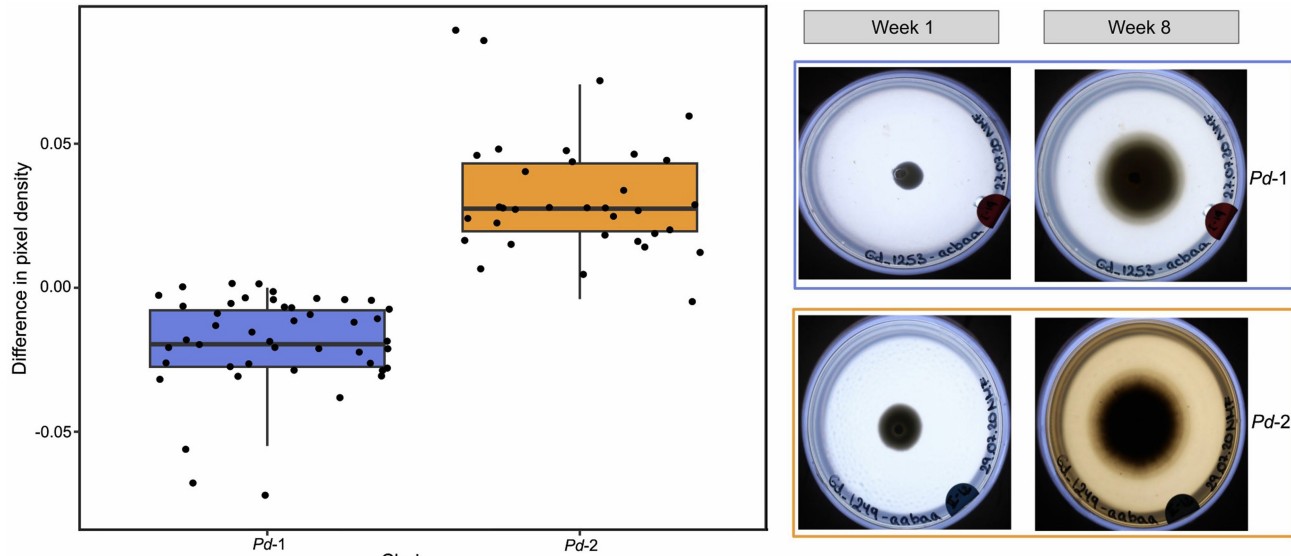

**Extended Data Fig. 2 | Colouration of culture medium by growth of *Pd*-1 and *Pd*-2.** Left: Difference in colour (darkness) of culture medium after 7 weeks using pixel density as a proxy. Photos were taken of 45 and 34 isolates of *Pd*-1 and *Pd*-2 respectively and analysed in R (see section 'Analysis of culture darkness' in the Methods and Table S8). To calculate the difference, the median pixel density at 8 weeks was subtracted from the pixel density at 1 week whereby a positive value indicates an increase in darkness. The black line indicates the median, while the lower and upper hinges represent the first and third quartiles, respectively. The whiskers extend to 1.5 times the interquartile range. Right: Examples of analysed photos (original) for *Pd*-1 and *Pd*-2. Clades *Pd*-1 and *Pd*-2 differ significantly in terms of colouration of the agar medium (Week 1 – Week 8; two-sided t-test: $t = -11.58$, df = 60.06, $p < 0.001$).

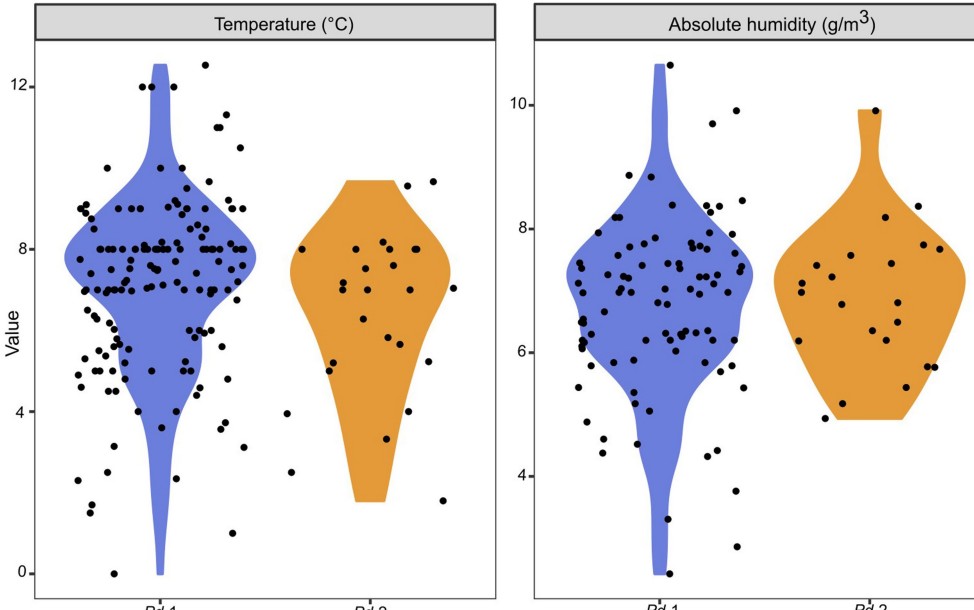

**Extended Data Fig. 3 | Temperature and absolute humidity recorded in Eurasian hibernacula.** Results are shown as raw data (dots) and as violin plots (coloured shading) showing the probability density estimate of the variables per clade. Temperature data were obtained from 152 and 26 sites in which clades *Pd*-1 and *Pd*-2 were sampled while absolute humidity was recorded in 91 and 22 sites per clade. There was no significant difference between the clades, either for temperature (two-sided t-test, $t = 1.65$, df = 35.52, $p = 0.11$) or absolute humidity (two-sided t-test, $t = -0.56$, df = 37.96, $p = 0.58$).

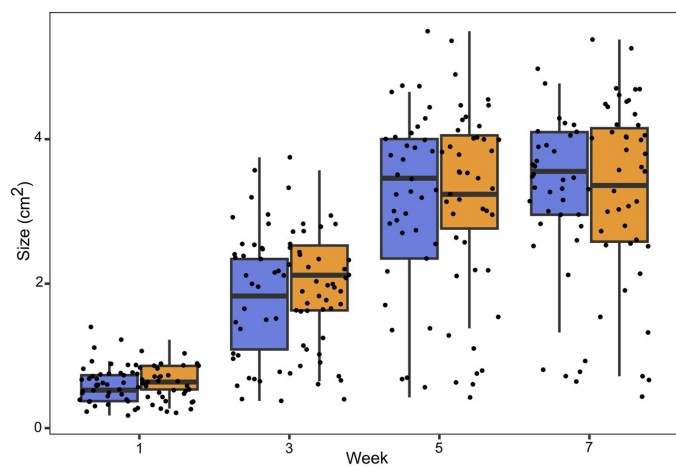

**Extended Data Fig. 4 | Bi-weekly size of cultures belonging to clades *Pd*-1 and *Pd*-2.** Measurement of culture sizes for 45 and 34 isolates of *Pd*-1 and *Pd*-2 respectively recorded for a growth period of 7 weeks (after which point growth ceases) at 15 °C. The size was measured from photos using R software (for more information see section 'Analysis of growth' in the Methods). The black line indicates the median, while the lower and upper hinges represent the first and third quartiles, respectively. The whiskers extend to 1.5 times the interquartile range. There was no significant difference in culture size of isolates belonging to *Pd*-1 compared to *Pd*-2 (two-sided t-test, *p* ranging from 0.07 to 0.97 for each week).

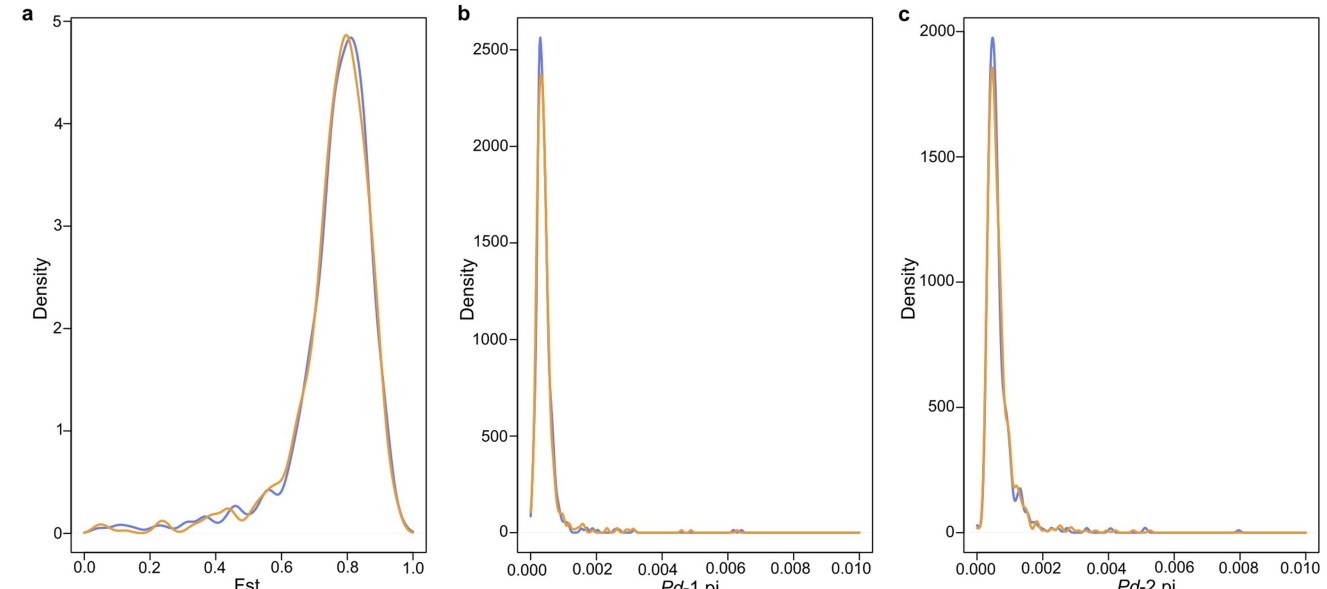

**Extended Data Fig. 5 | Density of multi-locus $F_{ST}$ (a), π for *Pd*-1 (b) and π for *Pd*-2 (c) when mapping the 18 individually tagged isolates (11 *Pd*-1 and 7 *Pd*-2) on Gd293 reference genome (blue) or on Gd45 reference genome (orange).** To better visualize π values, values greater than 0.01 were omitted (representing less than 0.25% of values). 95% highest density intervals for multi-locus $F_{ST}$ are 0.50−0.95 and 0.51−0.95 when using ref Gd293 and ref Gd45 respectively. For π, the means for *Pd*-1 are $4.2 \times 10^{-4}$ and $7.1 \times 10^{-4}$, and the 95% highest density intervals for *Pd*-1 are $6.5 \times 10^{-5} - 8.3 \times 10^{-4}$ and $0 - 0.9 \times 10^{-4}$ when using ref Gd293 and ref Gd45 respectively. For *Pd*-2, the means are $4.6 \times 10^{-4}$ and $7.3 \times 10^{-4}$, and the 95% highest density intervals for π are $2.1 \times 10^{-4} - 1.5 \times 10^{-3}$ and $1.9 \times 10^{-4} - 1.5 \times 10^{-3}$ when using reference genome Gd293 and reference genome Gd45 respectively. See section 'Diversity and differentiation calculation' in the Methods, and Table S10 for further information on methodology and isolates, respectively.

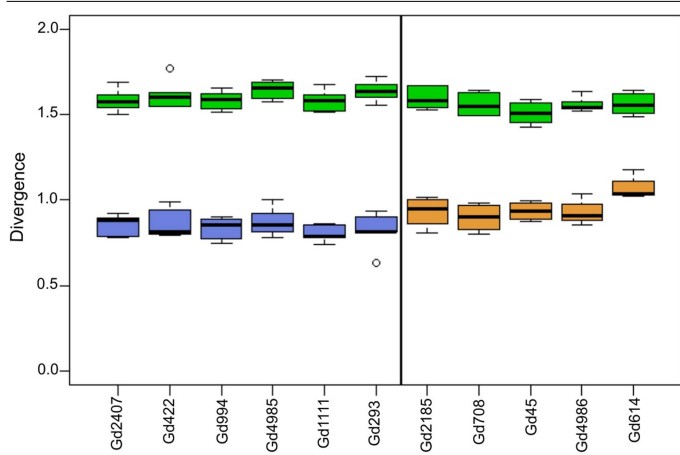

**Extended Data Fig. 6 | Boxplot of the pairwise distance between isolates for 11 full genomes, partitioned between intra- and inter-clade distance.** Intra-clade *Pd*-1 & *Pd*-2 divergence are coloured in blue (*Pd*-1) and orange (*Pd*-2) while inter-clade divergence is coloured in green (as per Fig. 2c). The black line represents the median while the lower and upper hinges correspond to the first and third quartiles and the whiskers extend to 1.5 times the interquartile range with data points beyond this range shown as outlier points.

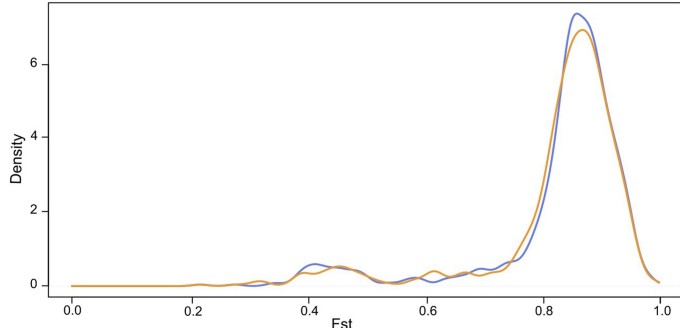

**Extended Data Fig. 7 | Density of multi-locus $F_{ST}$ across the genome when mapping the Pool-seq data on Gd293 reference genome (blue) or on Gd45 reference genome (orange).** Note that the distributions are extremely similar independently of the reference genome. Indeed, the median multi-locus $F_{ST}$ values are 0.88 (95% highest density interval [hdi], 0.50−0.99) and 0.88 (95% hdi, 0.49−0.97) when using reference genome Gd293 and reference genome Gd45 respectively.

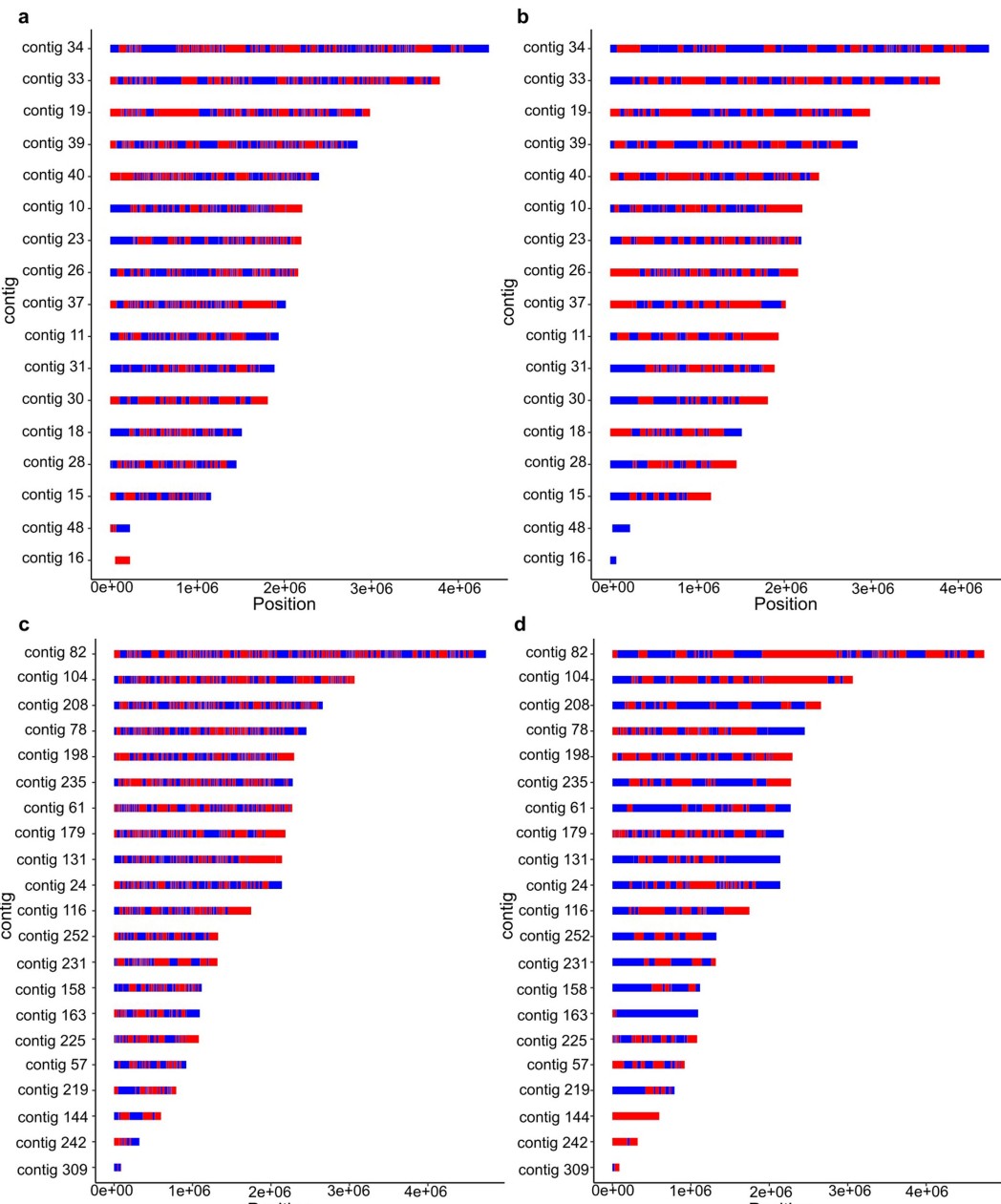

**Extended Data Fig. 8 | Genomic location of within-clade recombination breakpoints.** Based on the four-gamete test using SNPs from clade *Pd*-1 (a, c; *n* = 11 isolates) and *Pd*-2 (b, d, *n* = 7 isolates) when using Gd293 as reference genome (a, b), or Gd45 (c, d). When using Gd293 reference genome: 264 out of 331 and 130 out of 215 windows with at least two SNPs show evidence of recombination in *Pd*-1 and *Pd*-2 respectively. When using Gd45 reference genome: 261 out of 309 and 117 out of 210 windows with at least two SNPs show evidence of recombination in *Pd*-1 and *Pd*-2 respectively. The Φw test of recombination significantly rejected clonality ($p$ = 0.0) in all four instances (within each of the two *Pd* clades, whether considering Gd45 or Gd293 as reference genome). Contiguous regions alternate between blue and red at break points estimated by the four-gamete test. The population recombination rate ($r$ = 2 Ne $r$; see section 'Analyses of recombination' in the Methods) was estimated at $1.0 \times 10^{-4}$ and $4.6 \times 10^{-5}$ for *Pd*-1 and *Pd*-2, respectively. The recombination rate was lower in *Pd*-2 than in *Pd*-1, confirming the result obtained with the FGT test.

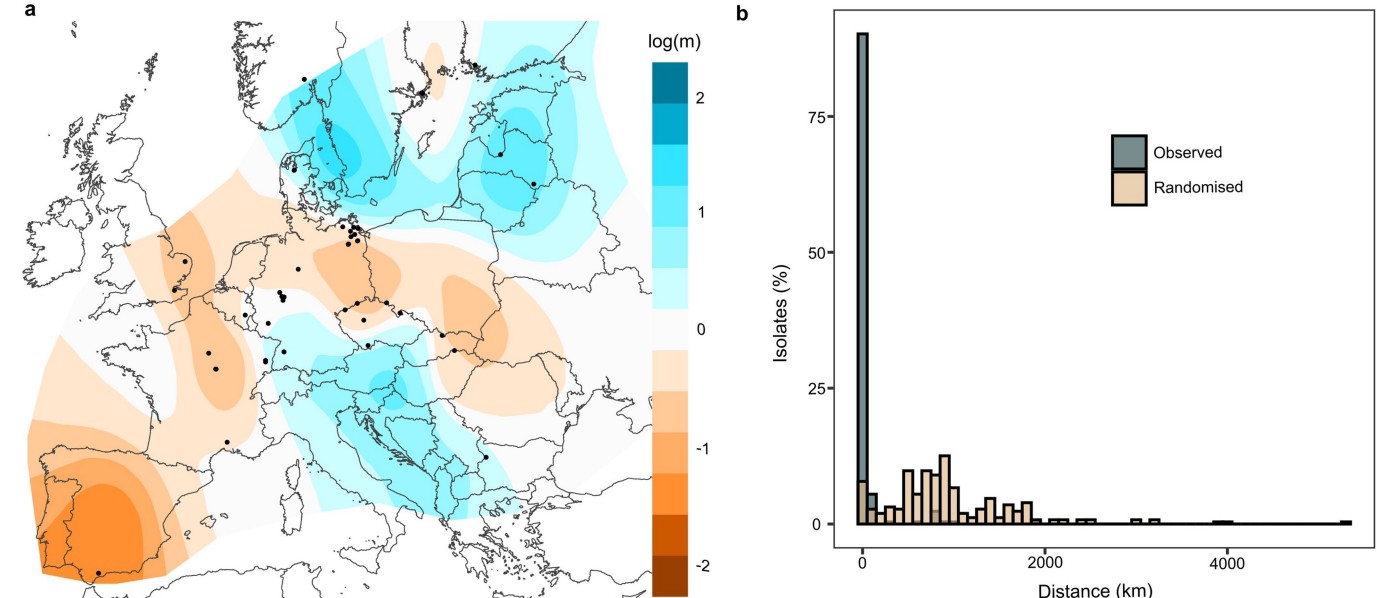

**Extended Data Fig. 9 | Population differentiation in *Pd*-2. a**, Estimation of effective migration surfaces based on 107 isolates from *Pd*-2 in Europe (all sites excluding Russia after clone correction). For visualization, results from eight independent runs (each with 8 million iterations and between 100 and 450 demes), were combined. Different shades of a colour represent variable levels of high (blue) or low (brown) effective migration rates. Sampling locations are represented by black dots. **b**, Distribution of distance between true and assigned site of each isolate (*n* = 279) for the observed and randomized datasets (binwidth = 100 km). The mean distance of assignment was 42.88 km with a median of 0 km. In the Null-DAPC with randomization of sites before assignment, the mean and median were 913.68 km and 774.05 km, respectively.

# Reporting Summary

## Statistics

For all statistical analyses, confirm that the following items are present in the figure legend, table legend, main text, or Methods section.

| n/a | Confirmed | |
|---|---|---|
| ☐ | ☒ | The exact sample size (*n*) for each experimental group/condition, given as a discrete number and unit of measurement |
| ☐ | ☒ | A statement on whether measurements were taken from distinct samples or whether the same sample was measured repeatedly |
| ☐ | ☒ | The statistical test(s) used AND whether they are one- or two-sided *Only common tests should be described solely by name; describe more complex techniques in the Methods section.* |
| ☐ | ☒ | A description of all covariates tested |
| ☐ | ☒ | A description of any assumptions or corrections, such as tests of normality and adjustment for multiple comparisons |
| ☐ | ☒ | A full description of the statistical parameters including central tendency (e.g. means) or other basic estimates (e.g. regression coefficient) AND variation (e.g. standard deviation) or associated estimates of uncertainty (e.g. confidence intervals) |
| ☐ | ☒ | For null hypothesis testing, the test statistic (e.g. *F*, *t*, *r*) with confidence intervals, effect sizes, degrees of freedom and *P* value noted *Give P values as exact values whenever suitable.* |
| ☐ | ☒ | For Bayesian analysis, information on the choice of priors and Markov chain Monte Carlo settings |
| ☐ | ☒ | For hierarchical and complex designs, identification of the appropriate level for tests and full reporting of outcomes |
| ☒ | ☐ | Estimates of effect sizes (e.g. Cohen's *d*, Pearson's *r*), indicating how they were calculated |

*Our web collection on statistics for biologists contains articles on many of the points above.*

## Software and code

Policy information about availability of computer code

| Data collection | No software was used. |
|---|---|
| Data analysis | The following software was used in analyses (full description of analyses provided in supplemental information): Augustus (v3.4.0), Bedops (v2.4.41), BEDTools (v2.30.0-8), BLASTN (v2.9.0+), BUSCO (v5.2.2), bwa-mem (v.0.7.17-r1188), bwa-mem2 (v2.2.1), CD-HIT-est (v4.8.1), DIAMOND (v2.1.7), DILS (unversioned, https://github.com/popgenomics/DILS_web), EEMS (unversioned), EMBOSS (v6.6.0.0), Fastp (v0.23.4), FGT.pl script (unversioned, https://github.com/dbsloan/fgt), FigTree (v1.4.4), Flye (v2.9), funannotate pipeline (v1.8.1), gatk (v4.2.6.1), GeneMapper® (v5), Guppy (v5.0.7), hmmsearch (v3.1), HyPo (v1.0.3), Inkscape (v1.1.1), IQTREE2 (v2.0.6), LDhelmet (v1.10), MAFFT (v7.453), metaeuk (v5.34c21f2), msnsam (october 2007 version), NanoPlot (v1.42.0), NanoStat (unversioned), NUCmer4 (v4.0.0), orthoDB (v10), Picard (v2.27.1), pixy (v1.2.7.beta1), Porechop (unversioned, https://github.com/rrwick/Porechop), RepeatMasker (v.4.1.2), RepeatModeler (v2.0.1), RNA STAR mapping tool (v2.7.10b), Samtools (v1.16.1), SeqKit (v0.16.1), SNAPE-pooled (unversioned, https://github.com/EmanueleRaineri/snape-pooled), Splitstree CE (v6.3.27), vcftools (version modified by J. Dutheil; https://github.com/jydu/vcftools) as well as R software (v4.1.1) with the packages abc (v2.2.1), abcrf (v1.9), adegenet (v2.1.5), adehabitatHR (v0.4.21), adimpro (v0.9.6), ape (v5.7.1), brms (v2.20.3), EBImage (v4.3), ggmap (v3.0.2), ggplot2 (v3.5.0), hierfstat (v0.5.11), INLA (v0.0.4), ks (v1.14.1), MonoPhy (v1.3.2), performance (v0.12.4), Phangorn (v2.11.1), poolfstat (v2.0.0.), poppr (v2.9.3), sp (v1.4-6), SPASIBA (v24.6.27), Syntenet (v1.8.1), reemsplots2 (v0.1.0), rworldxtra( v1.01) and rworldmap (v1.3.8). |

For manuscripts utilizing custom algorithms or software that are central to the research but not yet described in published literature, software must be made available to editors and reviewers. We strongly encourage code deposition in a community repository (e.g. GitHub). See the Nature Portfolio guidelines for submitting code & software for further information.

# Data

Policy information about availability of data

All manuscripts must include a data availability statement. This statement should provide the following information, where applicable:
- Accession codes, unique identifiers, or web links for publicly available datasets
- A description of any restrictions on data availability
- For clinical datasets or third party data, please ensure that the statement adheres to our policy

The genomic sequences and assembled genomes have been deposited under accession numbers SRR30476767–SRR30476787 and SRR30476795–SRR30476796 (see Table S10), while the PoolSeq data are available under accession numbers SRR30476788–SRR30476794 and SRR30476766 (see Table S15). These data can be accessed via BioProject PRJNA862744 at the National Center for Biotechnology Information. Metadata along with microsatellite genotypes data are provided in Table S1 (Exact site locations are not disclosed to protect endangered species and landowners), temperature, absolute humidity and M. daubentonii presence in Table S3, colouration of agar medium in Table S8, and colony expansion rates in Table S9. All other data are available in the manuscript or the supplementary information.

# Research involving human participants, their data, or biological material

Policy information about studies with human participants or human data. See also policy information about sex, gender (identity/presentation), and sexual orientation and race, ethnicity and racism.

| | |
|---|---|
| Reporting on sex and gender | N/A |
| Reporting on race, ethnicity, or other socially relevant groupings | N/A |
| Population characteristics | N/A |
| Recruitment | N/A |
| Ethics oversight | N/A |

Note that full information on the approval of the study protocol must also be provided in the manuscript.

# Field-specific reporting

Please select the one below that is the best fit for your research. If you are not sure, read the appropriate sections before making your selection.

☐ Life sciences    ☐ Behavioural & social sciences    ☒ Ecological, evolutionary & environmental sciences

For a reference copy of the document with all sections, see nature.com/documents/nr-reporting-summary-flat.pdf

# Ecological, evolutionary & environmental sciences study design

All studies must disclose on these points even when the disclosure is negative.

| | |
|---|---|
| Study description | Characterisation of ecological and genetic differentiation across Eurasian and North American samples of the fungus Pseudogymnoascus destructans |
| Research sample | Reference collection of 5,479 isolates of Pseudogymnoascus destructans sampled primarily across Eurasia (33 isolates from North America). In addition, 11 Pseudogymnoascus destructans isolates (10 from Eurasia and 1 from North America) were selected for full genome long-read sequencing and were analysed together with 7 previously published Pseudogymnoascus destructans Genomes (5 from Europe, 1 from Mongolia, 1 from China; Drees et al., 2017). Furthermore, a total of 132 isolates from this study were used for pooled Illumina sequencing (split into two pools). |
| Sampling strategy | We aimed to sample as many sites as feasible throughout Europe while keeping a good geographical representation. In total, we obtain 5,446 samples from 256 sites in 26 countries in Europe, offering an important geographic coverage of the suspected region of origin. For North America, given the clonal origin of the introduction, a much lower sample size (N=9 sites and 33 isolates) was needed. |
| Data collection | Our work adhered to the ethical wildlife research guidelines of the American Society of Mammalogists for the use of wild mammals in research and education (Sikes et al. 2016). Swab samples of Pseudogymnoascus destructans were collected from within bat hibernacula. Sampling from hibernating bats was conducted without capture or handling, by collecting samples while the bats remained freely hanging. The samples were collected by lightly swabbing the infected areas with a sterile dry swab. This method is considered as minimally invasive or even non-invasive. This work was completed with the help from A. Bezard, A. Kubátová, Aleksandra Lange, Alessandra Peron, Alex Lefevre, Alexander Lazarov, Alexandra Telea, Alexandre Cartier, Alina Larion, Alphonse Malpel, Amanda Davies, Andres Beck, Andrew Brinckman, Andriy |

Melnychuk, Andrzej Kepel, Andrzej Wojtaszewski, Angel Ivanov, Àngel Torrent, Ann Lenaerts, Anna Roswag, Anna Suvorova, Anne-Jilke Haarsma, Anne Petzold, Annika Breitfelder, Anthony Lane, Anthony Le Nozahic, Anthony Nickson, Antonia Hubancheva, Artem Tarasov, Ash Murray, Atanas Stavrev, Axel Donning, Axel Griesau, Axel Keusemann, Bart Mulkens, Benjamin Meme-Lafond, Bernd Ohlendorf, Bernhard Walk, Beytullah Özkan, Blanka Lehotská, Boris Petrov, Brian Briggs, Brigitte Meiswinkel, Carlos Ibáñez, Carsten Dense, Catherine Reilly, Chris Vine, Christian Dietz, Christian Jungmann, Christian Sebening, Christoph Treß, Christophe Borel, Christophe Parisot, Christopher Paton, Claudi Gebhart, Clemens Kliesch, Colin Morris, Corentin Le Floch, Csaba Jére, Damian Celiński, Dana Wagemakers, Daniel Eva, Daniela Hamidovic, Daniela Pilgrim, Daniela Schmieder, Daniela Wieser, Dave Hughes, David Anderson, David Aupermann, David Dodds, David Endacott, David García Jiménez, David Hellmann, David Patterson, David Wills, Didier Montfort, Dieter Hülshoff, Dieter Sulzbacher, Dimitar Kunev, Dirk Karoske, Dragoș Bălășoiu, Ebbe Nytors, Eeva-Maria Kyheröinen, Egoitz Salsamendi, Elena Migens Maqueda, Emrah Çoraman, Erich Taube, Ernst Auer, Eva Kriner, Ewa Przepiorka, Fabio Bontadina, Fabio Suppini, Fiona Parker, Florian Gloza-Rausch, Francesco Grazioli, Frank Meisel, Frauke Meier, Frédéric Forget, Frédéric Touzalin, Fulgencio Lison, Gabriella Krivek, Gaël Verat, Gary Shears, Georg Warnke, Gerald Kerth, Gerald Larcher, Giazarian, Goran Rnjak, Gregory Beneux, Grzegorz Apoznański, Grzegorz Lesinski, Guilia Console, Gunars Petersons, Gunther Capo, Gustav Dinger, Gwenaelle Hurpy, Gwendoline Dumenil, H. Bandouchova, H. Seimers, Hannes Köble, Harald Mixanig, Heino Hauf, Helen Miller, Henryk Hörner, Holger Schütt, Hubert Baltus, Iain Hysom, Ian Bond, Ilaria Vaccarelli, Ilona Imoberdorf, Ilze Brila, Inazio Garin, Ingrid Heißen, Ingrid Oftedal, Irbin Manuel Veliz Isidro, Ireneusz Ruczynski, Irina Pocora, Irina Würtele, István Csősz, Ivailo Borissov, Ivan Napotnik, Ivana Budinski, J. Flousek, J. Nogueras, J. Pikula, J. Zukal, J.L. Gathoye, Jamie Shadbolt, Jan Boshmer, Jane Harris, Jane Sedgeley-Strachan, Jasmin Pašić, Jasminko Mulaomerović, Jean-Yves Courtois, Jean Guhring, Jenny Harris, Jens Berg, Jens Krüger, Jens Rydell, Jeroen van der Kooij, Joachim Frömert, John Haddow, Johnny de Jong, Jörn Horn, Jose Siles, Juan R. Boyero, Julia Prüger, Juliane Schatz, Jurgis Suba, Justyna Błesznowska, Karina Jungmann, Karsten Passior, Katharina Bürger, Kathy Warden, Kees Mostert, Kerstin Genz, Klaus Heck, Kristof De Clercq, Krum Sirakov, Krzysztof Piksa, Kseniia Kravchenko, Laura Torrent, Laurence Florian, Laurent Arthur, Lauri Lutsar, Lea Bütje, Lena Godlevska, Lena Grosche, Lide Jimenez, Lilian Girard, Lionel L'Hoste, Lisa Worledge, Llorenc Capella Ripoll, Loic Robert, Lotte Gielis, Lucretia Deplazes, Ludovic Jouve, Luis Vicente, Luisa Rodrigues, Lyn Wells, M. Kubešová, M. Orlova, Magda Milczarska, Maik Korreng, Manfred Keller, Manuel Graf, Manuela Schult, Mara Calvini, Marcin Rusinski, Maria Das Neves Paiva Cardoso, Marion Laprun, Markus Melber, Markus Milchram, Markus Schmidberger, Markus Thies, Martin Biedermann, Martin Harder, Martin Koch, Martin Starrach, Martina Palmer, Mathijs Borms, Matija Perne, Matthias Göttsche, Matthias Hammer, Matthias Weiß, Matthias Zizelmann, Mauro Mucedda, Mechthild Höller, Michael Frede, Miguel Àngel Fuentes Rosua, Mike Debret, Mirna Mazija, Momchil Naydenov, Monika Podgorelec, Morten Elmeros, N. Martinkova, Nataša Sivec, Nia Toshkova, Nick Tribe, Nicolas Cayssiols, Nicolas Fasel, Nicola Fischer, Niklois Jungbluth, Nina Hagner-Wahlsten, Norbert Röse, Nuno Pinto, O. Orlov, Oliver Kalda, Oleksandr Vikyrchak, Olvido Tejedor, Oscar de Paz, P. Blažková, P. Schnitzerová, P. Táje, P. Tájek, Paola Culasso, Pascal Bellion, Pascal Giosa, Pascal Verdeyroux, Patty Briggs, Paul Hope, Paweł Kmiecik, Per IngeVærnesbranden, Peter Busse, Peter Heubes, Peter Holtz, Peter Smith, Petra Gatz, Petra Žvorc, Petro Ploshchanskyi, Philippe Defernez, Philippe Theou, Pierre-Emmanuel Bastien, Piotr Zielinski, Primož Presetnik, Quentin Smits, Radek Lučan, Radostina Tsoneva, Rainer Marcek, Ralf Hansen, Ralf Koch, Rasit Bilgin, Rauno Kalda, Reimund Francke, Reinhard Koch, Rémi Hanotel, Rich Flight, Roberto Toffoli, Robin Moffitt, Ruddy Cors, S. Rebrov, Sabine Lind, Sabine Portig, Sam Dyer, Sándor Boldogh, Sandra Möller, Sebastian Petters, Sebastien Puechmaille, Serbülent Pakzuz, Serena Dool, Serena Magagnoli, Sheelagh Kerry, Shirley Thompson, Simone Pysarczuk, Stamen Dimitrov, Stanimira Deleva, Stefan Schürmann, Steffi Pfeiffer, Stephanie Wohlfahrt, Steve Parker, Stoyan Goranov, Sue Lane, Susan Kerwin, Susanne Rosenau, Szilárd Bücs, T. Juhnke, Tamás Görföl, Tarik Dervović, Tea Knapič, Teodor Jhotev, Thomas Bormann, Thomas Cheyrezy, Thomas Frank, Thomas Kuß, Thomas Le Campion, Thomas Lilley, Tiago Brito, Tina Aughney, Tino Staudt, Todor Karakiev, Tom Hastings, Tom McOwat, Tomasz Kokurewicz, Toni Watt, Tony Lane, Torsten Blohm, Tsvetan Ostromsky, Ulrich Zöpfel, V. Kovacova, V. Lensinger, V.S. Crukov, Vesselin Zhelyazkov, Victor Senderov, Victoria Nistreanu, Viktor Ilyukha, Viorel Pocora, Vita Hommersen, Vitaliy Guckov, Vivien Sottejean, Vladislav Caldari, Vlashenko, Volker Kubisch, Weigert Steen, Wigbert Schorcht, Winfried Krämer, Wolfgang Fiedler, Wolfgang Otremba, Wolfgang Rackow, Wolfgang Strittmatter, Xavier Mestdagh, Yana Dimova, Yann Gager, Yann Le Bris, Yannick Beucher, Yvon Guenescheau, Zuzanna Halat.

| | |
|---|---|
| Timing and spatial scale | The samples were collected between 02.02.2008 and 11.03.2022 based on feasability and availability of bats. The timing of the sampling was of no interest to the study design and adressed research questions. |
| Data exclusions | No data were excluded from the analyses. |
| Reproducibility | In Table S1, we provide a genotype table of all isolates and necessary additional information (metadata) for a full reproducibility of all the analyses. Similarly, we provide GenBank accession numbers of raw sequences and genomes produced in this study. All analyses are described in detail in the Material and Methods section to allow full reproducibility. |
| Randomization | There was no randomisation in the study considering it was an observational study of different individuals and their genotypes and genomes. |
| Blinding | There was no blinding in the study as this was a descriptive study design not an experimental one. |

Did the study involve field work? ☒ Yes ☐ No

# Field work, collection and transport

| | |
|---|---|
| Field conditions | Field work was conducted for sample collections from temperate hibernacula, ususally in the winter season. A total of 264 sites were sampled with conditions varying between different sites and years though cool and humid conditions were typical. |
| Location | 264 sites were sampled across Eurasia and North America (see Table S1). Exact site locations are not disclosed to protect endangered species and landowners. |
| Access & import/export | This work was conducted under permission from the following authorities:<br>Italy: Regional Speleological Federation of Emilia-Romagna (F.S.R.E.R.), and the Management Bodies of the Parks of Emilia-Romagna;<br>Poland: Genarny Dyrektor Ochrony Środowiska (General Director for Environmental Protection); Regional Directorate for |

Environmental Protection in Gorzów Wielkopolski (Regionalna Dyrekcja Ochrony Środowiska w Gorzowie Wielkopolskim); Switzerland: Kantonaler Fledermausschutz Aargau; Germany: Umweltamt, Veterinäramt; Untere Landschaftsbehörde Siegen-Wittgenstein; Untere Naturschutzbehörde Umweltamt Landkreis Harz & Referat Verbraucherschutz, Veterinärangelegenheiten Landesverwaltungsamt Sachsen-Anhalt; Untere Naturschutzbehörde des Landkreises Vorpommern-Greifswald; Regierung von Unterfranken, Regierung von Mittelfranken; Struktur- und Genehmigungs Direktion Nord/Süd, NLWKN Niedersächsischer Landesbetrieb für Wasserwirtschft, Küsten- und Naturschutz, Region Hannover - Fachbereich Umwelt; Austria: Department of nature conservation for Carinthia, Lower Austria, Upper Austria, Salzburg, Styria and Vorarlberg; Hungary: Pest Megyei Kormányhivatal, Országos Környezetvédelmi, Természetvédelmi és Hulladékgazdálkiodási Főosztály (Pest County Government Office, National Department of Environment Protection, Nature Conservation and Waste Management); Ministry of Environment and Water; Bulgaria: Bulgarian Ministry of the Environment and Water; France: DDTM-Morbihan; DREAL; Republic of Latvia: Nature Conservation Agency; Belgium: Gouvernement Wallon; Denmark: The Nature Agency and Daugbjerg Kalkgruber; Romania: Speleological Heritage Commission; Estonia: Estonian Environmental Board; England: Natural England; Finland: Southwest Finland Centre for Economic Development, Transport and the Environment; Sweden: Uppsala djurförsöksetiska nämnd; Swedish board of Agriculture; Swedish Environmental Protection Agency; Norway: Miljødirektoratet; Luxemburg: Ministère du Développement durable et des Infrastructures du Luxembourg; Croatia: Croatian Ministry of Environment and Nature; Russian Federation: Game Management Directorate of the Republic of Karelia; Institute of Plant and Animal Ecology, Ural Division of the Russian Academy of Sciences; Slovak Republic: Ministry of the Environment of the Slovak Republic, Department of State Administration for Nature and Landscape Protection; The Netherlands: Dutch Ministry of Economic affairs; Republic of Moldova: Government of Republic of Moldova - Ministry of Environment.

| Disturbance | Swab samples were collected in a minimally invasive or non-invasive manner, without handling the animals. Disturbance from human presence (e.g., time spent at the site) and noise was kept to a minimum, following the procedures typically used during regular hibernacula counts. |
|---|---|

# Reporting for specific materials, systems and methods

We require information from authors about some types of materials, experimental systems and methods used in many studies. Here, indicate whether each material, system or method listed is relevant to your study. If you are not sure if a list item applies to your research, read the appropriate section before selecting a response.

## Materials & experimental systems

| n/a | Involved in the study |
|---|---|
| ☒ | ☐ Antibodies |
| ☒ | ☐ Eukaryotic cell lines |
| ☒ | ☐ Palaeontology and archaeology |
| ☒ | ☐ Animals and other organisms |
| ☒ | ☐ Clinical data |
| ☒ | ☐ Dual use research of concern |
| ☒ | ☐ Plants |

## Methods

| n/a | Involved in the study |
|---|---|
| ☒ | ☐ ChIP-seq |
| ☒ | ☐ Flow cytometry |
| ☒ | ☐ MRI-based neuroimaging |

## Plants

| Seed stocks | NA |
|---|---|

| Novel plant genotypes | NA |
|---|---|

| Authentication | NA |
|---|---|

