## [Peer Review File · Nature]

Two distinct host-specialised fungal species cause white-nose disease in bats

Corresponding Author: Dr Sébastien Puechmaille

Version 0:

Reviewer comments:

Referee #1

(Remarks to the Author)

I commend the authors for what has been an incredible amount of work that has gone into this study; 5,479 18-locus genotypes from Pd isolated across 27 countries and 1,463 swabs from 16 bat species is a herculean effort. The results of the subsequent population genetic / genomic analyses are well done and biologically insightful. The broader findings are relevant to conservation and biosecurity.

The finding that there are strong indications of host-specificity – at least for the Pd-2 clade and *M. daubentonii* - is pertinent given that fungal diseases trend towards being non-specialist and multi-host on the whole. The comparison here would be with Bd & Bsal which show a tantalisingly similar pattern (Bd is host-generalist, Bsal specialises on caudates), although the Batrachochytrids are far more deeply diverged – it's probably relevant to cite that host-association as a comparator. The bat-host observation (with further research) will lead to a better understanding the evolution of host-pathogen associations in sympatry, and also explains why the more generalist Pd-1 has been such a potent invader of the American bat species. The observation of very profound population genetic structure for WNS in Europe (both within and amongst the cryptic species) compared to the rapid spatially-unconstrained spread in the Americas is an archetype for the havoc wreaked when a pathogen breaks free of its host evolved range (such as was seen with the amphibian chytrids).

Clades:

The microsatellite and genomic analysis compellingly show the occurrence of 2 sympatric clades Pd1 & Pd2 that are non-recombining and fulfill multiple criteria for cryptic species. The F_{st} values – Fig 2D – are approaching fixation for the majority of genes and nearly 97.5% of SNPs; recombination occurs within but not between clades, mating types are in balanced frequency and divergence between the clades equates to ~ 1 -1.5 MyA. The authors show some phenotypic differentiation (eg colouration of media) suggesting that isolation will have led to metabolic divergence between Pd1 & 2 that may in future lead to discovery of the virulence factors leading to host-specialisation and divergence.

Sympatry:

Owing to the depth of the sampling of microsatellite genotypes, and the large number of bat species sampled, some with considerable depth, there is reasonable statistical power to assert sympatry and even to detect co-occurrence on the same host. Hybrids are not detected when co-occurrence occurs in space bolstering the conclusion of strong reproductive isolation.

Given that in regions of sympatry, the major isolating factor is species, suggests a role of bat host species in the potential evolution-in-sympatry of Pd1 & 2. This finding is compelling for Daubentons bat which is seen to exclusively harbor Pd2 despite having a very wide pan-European distribution. I find this association fairly extraordinary and look forward to future research unpacking the biological mechanisms that underpin the association.

Origins of NAM Pd1:

There is unambiguous assignment of the North American invasion of Pd to the Pd1 clade. Given the degree of intra-clade population structuring, genotype assignment suggests that the NAM Pd genotype originated in Eastern Europe – possibly Western Ukraine. This assignment is an important and interesting finding and it is good that the authors went beyond DAPC to use SPASIBA to assign genotype-probabilities (both methods gave the same answer).

My main criticism of the study is simply that it is fairly lengthy and the authors could condense it into a snappier format. That said – they pack in a lot of relevant detail and I wouldn't want to see much of that lost.

Referee #2

(Remarks to the Author)

This manuscript reports that the fungus that causes white-nose syndrome (WNS), which affects bat populations in the US, is not one but two distinct species and identifies the region of Europe that is the likely source of the WNS epidemic.

The work is impressive on multiple fronts. The authors examined ~5,500 isolates from ~260 sites in nearly ~30 countries - using microsatellite loci, they were able to identify two major clades. They then performed genome sequencing on 12 representative isolates and (after adding the genomes of 7 publicly available isolates) performed a series of evolutionary genomic analyses, which strengthened their support for the existence of two distinct species. They also conducted a series of phenotyping assays to identify any potential differences between the two species (they don't seem to have major differences in growth in different abiotic conditions). Rather, the main differentiator seems to be a preference for different bat hosts (host specialization) between the two species.

I found the manuscript very clear and the work convincing, not just for the discovery of a second species of *Pseudogymnoascus*, but also for showcasing so nicely the importance of having ecological and environmental information from a wide diversity of isolates in understanding how fungal epidemics originate and spread.

I have a few comments for the authors to consider:

a) There are quite a few superlatives in the text - just in the abstract, I noted "unprecedented", "notorious", "greatest", and "urgent". Please replace them with more descriptive terms and let your work and data speak for themselves.

b) I found the claim that host specialization is the primary factor and that abiotic factors do not play a role to be somewhat weakly supported. To be fair to the authors, it is very challenging / impossible to test all possible factors, so my suggestion would be that they tone down their inferences from the data at hand. There could be differences in the microclimates that the two species prefer, differences in abundances (in both the fungi and/or their bat hosts), differences in how the fungi interact with other species, and so on. A bit of caution in interpretation will keep readers' minds open to explore this important topic further.

c) Both the DAPC and the SPASIBA analyses identify a small region in Ukraine as the likely source of the population that's causing the WNS epidemic in the US. Another way to test this is to examine the sister lineages of US isolates in the microsatellite phylogeny (Figure 1B) - one would expect that the US isolates would be most closely related to the Ukrainian isolates from the Podillia region. Can you please test if that's the case?

Referee #3

(Remarks to the Author)

This paper reveals two cryptic taxa of *Pseudogymnoascus destructans* (Pd) that appear to show differential abundances on certain bat species. The manuscript also reports that the origin of Pd in North America is likely from eastern Europe.

The authors have done a tremendous amount of work and the conclusions are based on a very large dataset. I am impressed by the number of strains of Pd that the authors isolated for this project and the amount of work involved in genotyping all of these. I want to start my review by congratulating them for all the work they have done! I also found the paper to be very well written and the figures are of high quality. My comments are relatively minor, and I think the authors will be easily able to address them.

Specific Comments

The authors use the term "white-nose disease" to refer to infection with *Pseudogymnoascus destructans*, while the disease is much more commonly referred to as "white-nose syndrome" in much of the scientific literature and popular press. I agree with the authors that "disease" is the medically correct term to use. However, given the readership of *Nature*, it may be more appropriate to use "white-nose syndrome" so that general readership is better able to connect this paper with the disease by its most commonly known name. Disagreements over disease names are frequent (some would argue that this disease should actually be called "pseudogymnoascomycosis." In order for science to be more accessible and easily consumed by diverse audiences, we should not get so hung up on semantics except in instances where terminology might be more important (e.g., distinguishing between disease and syndrome within a medical community, specifically). A good example of this is how AIDS (acquired immunodeficiency syndrome) is used exclusively to refer to outcomes of infection by the human immunodeficiency virus, even though AIDS would be better labeled as a disease rather than a syndrome. Sometimes a disease becomes so well known by both the scientific community and the general public before the cause is identified and the "syndrome" name is retained simply for ease of understanding. I think white-nose syndrome is one such example. Also, "white-nose" should not be capitalized because disease names are only capitalized if named after proper nouns.

With respect to referring to the two *Pseudogymnoascus* as different species: Fungal taxonomy and naming conventions are needlessly complicated for ascomycete fungi. It might be more palatable for much of the mycology community to refer to these as two distinct “taxa” rather than calling them species. I am not familiar with naming conventions for this genus, but there may need to be more evidence of physiological or morphological differences to officially call these species. Personally, I think genetic evidence should be sufficient, but I am not sure if the fungal taxonomy community as a whole would agree.

Abstract

“...remains understudied” is an opinion. I suggest rewording this to “are often less well-studied.” While I agree that pathogen genetic variability tends to be less well-studied, there are numerous diseases for which host strain is well-understood to play an important role in infection ecology and disease outcomes.

Main Text

“...is a single fungal species with a broad distribution range.” I agree that what we know about susceptibility and resistance to Pd infection in North America is based on the presence of the single clonally-expanding strain that occurs there. However, concerns over additional introductions of more strains of Pd has been a concern ever since the fungus was discovered in Eurasia. In other words, I don’t think it is true that the existing literature assumes that Pd is not genetically diverse. This sentence might be better rewritten as “...However, importantly, this literature is based on a single clonally-expanded lineage of Pd and does not take into account the true genetic diversity of Pd that exists across the entire distribution of the species.”

“...lack of ecologically relevant data have impeded interpretations...” I am not sure what the authors are specifically referring to here. “Interpretations” of what? Again, as written, this sounds more like an opinion. I suggest rephrasing this sentence to be more specific and use words like “limit” rather than “impeded.”

“...to identify the source.” Suggest “...to identify the likely geographical source...”

“...differences in ecological niche and host specialization...” State the ecological niche factors that were investigated in this study.

Results and Discussion

Two sympatric *Pseudogymnoascus destructans* clades

Relatively speaking, 18 loci is a very small portion of the genome. How are these microsatellites dispersed on the genome? Is it possible with your data to state that the microsatellites are generally dispersed across the Pd genome and therefore would be likely to capture recombination events that might occur on a particular portion of the genome?

“...and, importantly, both have been demonstrated to cause the lesions diagnostic of the disease.” I suggest avoiding the word “importantly” in most scientific writing as it is subjective. Also, I assume that the citations you provide at the end of the sentence reference papers which describe the data that supports this statement (i.e., that isolates used in the current study originated from bats with confirmed lesions that are mentioned in those cited studies). I suggest stating more clearly that the strains you incorporated in the current project were isolated as parts of previous studies from bats with confirmed lesions. If this is not the case, then the data should be provided in the supplemental materials to demonstrate that there is histopathology to support both strains causing disease.

Figure 1: This is a beautiful figure panel! For panel B, what are the support values for the two main clades?

“...we have direct evidence...” These data need to be included in the supplemental or cited.

“The latter are therefore expected to come into contact with Pd-1 isolates....suggesting that Pd-1 is incompatible with *M. daubentoniid*.” Your methodologies used culture-based methods which are far less sensitive than methods like metagenomics that might have been able to detect smaller amounts of fungus on individual bats. Instead, your results are based on what grew most easily in culture or which colonies were selected for isolation. Therefore, it might be overinterpretation to say that Pd-1 is “incompatible” with *M. daubentoniid*. There also could be behaviors exhibited by *M. daubentoniid* at your study sites that would cause Pd-2 to be enriched in that species even though it could still be susceptible to infection by Pd-1 under certain circumstances.

“...the colouration of agar medium by laboratory cultures can distinguish between the Pd clades.” This is an interesting observation, but I think it would be best to mention that among the strains you examined, there was a significant difference between the clades in coloration. There does appear to be at least one strain of Pd-2 that overlaps Pd-1 and so it may not be a definitive characteristic for distinguishing between the taxa. Furthermore, Khankhet et al. (2014) PLoS ONE provide data that some North American strains of Pd (which would be Pd-1) secrete brown pigment and color the medium similar to what is shown in Figure S5. I can also attest to many strains of Pd from North America causing a similar color change, with variation occurring even within the same isolate. Therefore, it might be best to be more cautious in implying that this feature can be used to distinguish the clades (even if it is a trend).

“...have similar growth rates...” This is a very picky comment, but you might consider using the term “colony expansion rates” rather than “growth rates.” With the colony expansion method, it can be difficult to discern true differences in “growth” because radial hyphal growth is biased toward lateral (versus aerial or penetrative) expansion or hyphal density.

“...host specialization appears to be the primary factor distinguishing their ecological niches, rather than abiotic factors.”

Differential abundances on different hosts could certainly be due to abiotic factors. Caves are not homogenous environments and many bats select microhabitats with different microclimates within caves. It can be very difficult to tease apart biotic and abiotic components because many of them are intertwined. For example, a particular bat species' behavior may subject the pathogen to different abiotic conditions than it experiences on a different host. I think some additional work with the two taxa are necessary to better tease apart whether host specialization is the main factor that causes niche differentiation, especially since Pd-2 is capable of growing on bat species that Pd-1 also colonizes.

Genomic divergence between clades

Remove "the" in front of "664 single-copy BUSCO genes."

"...other than clonality or geography prevent genetic exchange." Replace "prevent" with "prevent detectable."

Continental population structure and origin of the North American introduction

"However, as soon as bats emerge from hibernation...due to the elevated temperature of active bats upon and after emergence." Remove "drastically" and "via grooming" from these sentences. I believe that grooming is only one of many factors that may remove the fungus from the surface of the bat. I would also point out that at least the American strains of the fungus survive periods of warming on bats (see Campbell et al. (2019), J Wildl Dis.) and your hypothesis that spread of the fungus would be limited by these factors is at odds with the rapid spread of Pd in North America. Overall, I do not think this hypothesis is well supported by what we know about Pd. Are their known differences in bat movements in Eurasia that would reduce movement of Pd on the landscape as compared to North America (you mention quite the opposite...that populations are panmictic)? Congregation of bats at hibernating sites to mate in the fall and long-distance travel of male bats during this time is thought to be a key contributor of spread in North America. Do bats in Eurasia have a different breeding ecology? Perhaps they do not use caves in the spring or fall? Perhaps overall lower loads of Pd on Eurasian bat species means there is little fungus being carried on them when they move from site to site? Or perhaps established populations of Pd in hibernacula prevent newly introduced strains from surviving? I think there could be many factors at play to explain the population structure seen in Pd.

A specific region of Ukraine is mentioned as being the likely source of the introduction into the U.S. However, based on Figure 3A, it appears that far eastern Europe is not well represented in the dataset. I am not very familiar with this type of analysis, but wouldn't the lack of samples from other parts of Ukraine and far eastern Europe (e.g. Belarus, western Russia) make it more difficult to pinpoint such a specific location? Nonetheless, it is interesting that Drees et al. also mentioned that an isolate from Ukraine most closely matched North American strains. You may want to mention that paper as it provides additional support for an eastern European origin.

A mean of 3 and 3.6 isolates were obtained for bat and environmental swabs, respectively. Were these typically clonal (you mentioned in a few instances of both strains being isolated from a single bat)? I realize this could be gleaned by looking at the supplemental, but it might be good to mention it in the manuscript proper.

Supplemental

Section 3.2.5

You mention that Onygenales is a sister order to Helotiales (to which Pd belongs). However, these orders are not even within the same class of fungi.

Section 3.4.4.1

"Confidentially" should be "confidently."

Section 3.5

Please describe how a single germinating spore was isolated to inoculate a new plate. How was this transferred? As an agar plug? I did not see this described in more detail in 2.1.

Raw growth data is not provided nor is coloration data. Providing the data you used for your analyses is important for open data access.

It would be helpful to list the individual GenBank accession numbers for each strain in one of the supplemental tables. Currently, I only noticed the project number listed. I apologize if I overlooked where the accession numbers are listed.

Fig S4: The map for *Myotis myotis*/*M. blythii* appears to have two species ranged depicted on the same map, but it is not clear which range represents which species.

Version 1:

Reviewer comments:

Referee #1

(Remarks to the Author)

I appreciate the careful attention that the authors have made in their revision. All the points that I raised have been substantially addressed either in the main text or in the SI.

Although not requested by me, I note the inclusion of historical data linking activity of cavers between the USA and the cave systems of Podillia in Ukraine spanning the period of WNS emergence. Although 'just so' stories can never prove disease transmission, this documented evidence at least details the commencement of activity following the dissolution of the Iron Curtain and further adds to the relevance of the studies implications on transnational biosecurity.

Referee #2

(Remarks to the Author)

The authors have satisfactorily addressed all my comments and concerns. Congratulations on putting together such an impressive body of work.

Referee #3

(Remarks to the Author)

The authors have done a great job responding to the reviewer comments. I think the addition of the sentences comparing the findings to the emergence of Bsal and the information about recreational caving in the vicinity of the region of Ukraine where the North American strain of Pd likely originated have really elevated the paper even more and make a very compelling case for the importance of the work. I do not have any additional comments. Well done!

We have provided a detailed point-by-point response with line numbers corresponding to the MS version with track changes. Our responses are highlighted in blue and marked with '>'.

#####
Referee #1 (Remarks to the Author):
I commend the authors for what has been an incredible amount of work that has gone into this study; 5,479 18-locus genotypes from Pd isolated across 27 countries and 1,463 swabs from 16 bat species is a herculean effort. The results of the subsequent population genetic / genomic analyses are well done and biologically insightful. The broader findings are relevant to conservation and biosecurity.
>We thank the Reviewer for their kind words.
The finding that there are strong indications of host-specificity – at least for the Pd-2 clade and M. daubentonii - is pertinent given that fungal diseases trend towards being non-specialist and multi-host on the whole. The comparison here would be with Bd & Bsal which show a tantalisingly similar pattern (Bd is host-generalist, Bsal specialises on caudates), although the Batrachochytrids are far more deeply diverged – it’s probably relevant to cite that host-association as a comparator. The bat-host observation (with further research) will lead to a better understanding the evolution of host-pathogen associations in sympatry, and also explains why the more generalist Pd-1 has been such a potent invader of the American bat species. The observation of very profound population genetic structure for WNS in Europe (both within and amongst the cryptic species) compared to the rapid spatially-unconstrained spread in the Americas is an archetype for the havoc wreaked when a pathogen breaks free of its host evolved range (such as was seen with the amphibian chytrids).
> Thank you for this suggestion. We have added “The emergence of chytrid fungi provides a cautionary example of the consequences of delayed action. Batrachochytrium dendrobatidis, a generalist pathogen, was linked to amphibian declines as early as the 1970s, yet at the time there was no awareness that a second, more specialised species could emerge. When B. salamandrivorans was finally identified in 2013, it was already too late to prevent its introduction to Europe”; lines 266-270.
Clades:
The microsatellite and genomic analysis compellingly show the occurrence of 2 sympatric clades Pd1 & Pd2 that are non-recombining and fulfil multiple criteria for cryptic species. The Fst values – Fig 2D – are approaching fixation for the majority of genes and nearly 97.5% of SNPs; recombination occurs within but not between clades, mating types are in balanced frequency and divergence between the clades equates to ~.1-1.5 MyA. The authors show some phenotypic differentiation (eg colouration of media) suggesting that isolation will have led to metabolic divergence between Pd1 & 2 that may in future lead to discovery of the virulence factors leading to host-specialisation and divergence.
Sympatry:
Owing to the depth of the sampling of microsatellite genotypes, and the large number of bat species sampled, some with considerable depth, there is reasonable statistical power to assert sympatry and even to detect co-occurrence on the same host. Hybrids are not detected when co-occurrence occurs

in space bolstering the conclusion of strong reproductive isolation.

Given that in regions of sympatry, the major isolating factor is species, suggests a role of bat host species in the potential evolution-in-sympatry of Pd1 & 2. This finding is compelling for Daubentons bat which is seen to exclusively harbor Pd2 despite having a very wide pan-European distribution. I find this association fairly extraordinary and look forward to future research unpacking the biological mechanisms that underpin the association.

Origins of NAM Pd1:

There is unambiguous assignment of the North American invasion of Pd to the Pd1 clade. Given the degree of intra-clade population structuring, genotype assignment suggests that the NAM Pd genotype originated in Eastern Europe – possibly Western Ukraine. This assignment is an important and interesting finding and it is good that the authors went beyond DAPC to use SPASIBA to assign genotype-probabilities (both methods gave the same answer).

My main criticism of the study is simply that it is fairly lengthy and the authors could condense it into a snappier format. That said – they pack in a lot of relevant detail and I wouldn't want to see much of that lost.

>Thank you for your insightful comments. We would be very willing to work on condensing any sections judged to require this once such sections are identified by the Reviewer or Editor.

Referee #2 (Remarks to the Author):

This manuscript reports that the fungus that causes white-nose syndrome (WNS), which affects bat populations in the US, is not one but two distinct species and identifies the region of Europe that is the likely source of the WNS epidemic.

The work is impressive on multiple fronts. The authors examined ~5,500 isolates from ~260 sites in nearly ~30 countries - using microsatellite loci, they were able to identify two major clades. They then performed genome sequencing on 12 representative isolates and (after adding the genomes of 7 publicly available isolates) performed a series of evolutionary genomic analyses, which strengthened their support for the existence of two distinct species. They also conducted a series of phenotyping assays to identify any potential differences between the two species (they don't seem to have major differences in growth in different abiotic conditions). Rather, the main differentiator seems to be a preference for different bat hosts (host specialization) between the two species.

I found the manuscript very clear and the work convincing, not just for the discovery of a second species of *Pseudogymnoascus*, but also for showcasing so nicely the importance of having ecological and environmental information from a wide diversity of isolates in understanding how fungal epidemics originate and spread.

I have a few comments for the authors to consider:

a) There are quite a few superlatives in the text - just in the abstract, I noted "unprecedented", "notorious", "greatest", and "urgent". Please replace them with more descriptive terms and let your work and data speak for themselves.

>Thank you. We have removed superlatives such as "unprecedented", "notorious", and "urgent" from the abstract and introduction and have replaced them by more descriptive terms.

b) I found the claim that host specialization is the primary factor and that abiotic factors do not play a role to be somewhat weakly supported. To be fair to the authors, it is very challenging / impossible to test all possible factors, so my suggestion would be that they tone down their inferences from the data at hand. There could be differences in the microclimates that the two species prefer, differences in abundances (in both the fungi and/or their bat hosts), differences in how the fungi interact with other species, and so on. A bit of caution in interpretation will keep readers' minds open to explore this important topic further.

>We agree with the Reviewer that it is "*very challenging / impossible to test all possible factors*". We therefore followed the Reviewer's suggestion and modified the sentence to highlight the complexity of the host-pathogen interaction system and keep readers' minds open to explore this important topic further (lines 157-160).

c) Both the DAPC and the SPASIBA analyses identify a small region in Ukraine as the likely source of the population that's causing the WNS epidemic in the US. Another way to test this is to examine the sister lineages of US isolates in the microsatellite phylogeny (Figure 1B) - one would expect that the US isolates would be most closely related to the Ukrainian isolates from the Podillia region. Can you please test if that's the case?

>Yes, the Reviewer is absolutely right. Beyond the DAPC and SPASIBA analysis, the US isolates are most closely related to Ukrainian isolates from the Podillia region in all other analyses carried out, including the 18-microsatellites presented Figure 1B (5,479 isolates; the full tree with labels for each isolate is shown in Fig. S2), the phylogeny of the 664 BUSCO genes presented in Figure 2B (Gd1111 [USA] being closest to Gd293 [Podillia region, Ukraine]), or the full genome split in 10 kb windows presented in Fig. S9 (Gd1111 [USA] being closest to Gd293 [Podillia region, Ukraine]). We have now made this explicit in the text [lines 332-334], further strengthening our identification of the Podillia region in Ukraine as the source population of the introduction to North America.

Referee #3 (Remarks to the Author):

This paper reveals two cryptic taxa of *Pseudogymnoascus destructans* (Pd) that appear to show differential abundances on certain bat species. The manuscript also reports that the origin of Pd in North America is likely from eastern Europe.

The authors have done a tremendous amount of work and the conclusions are based on a very large dataset. I am impressed by the number of strains of Pd that the authors isolated for this project and the amount of work involved in genotyping all of these. I want to start my review by congratulating them for all the work they have done! I also found the paper to be very well written and the figures are of high quality. **My comments are relatively minor, and I think the authors will be easily able to address them.**

>We thank the reviewer for their kind appraisal of our work.

Specific Comments

The authors use the term “white-nose disease” to refer to infection with *Pseudogymnoascus destructans*, while the disease is much more commonly referred to as “white-nose syndrome” in much of the scientific literature and popular press. I agree with the authors that “disease” is the medically correct term to use. However, given the readership of Nature, it may be more appropriate to use “white-nose syndrome” so that general readership is better able to connect this paper with the disease by its most commonly known name. Disagreements over disease names are frequent (some would argue that this disease should actually be called “pseudogymnoascomycosis.” In order for science to be more accessible and easily consumed by diverse audiences, we should not get so hung up on semantics except in instances where terminology might be more important (e.g., distinguishing between disease and syndrome within a medical community, specifically). A good example of this is how AIDS (acquired immunodeficiency syndrome) is used exclusively to refer to outcomes of infection by the human immunodeficiency virus, even though AIDS would be better labeled as a disease rather than a syndrome. Sometimes a disease becomes so well known by both the scientific community and the general public before the cause is identified and the “syndrome” name is retained simply for ease of understanding. I think white-nose syndrome is one such example. Also, “white-nose” should not be capitalized because disease names are only capitalized if named after proper nouns.

>Thank you for this insightful perspective. We agree with the Reviewer that “*white-nose disease*” is the medically precise term (as endorsed by the ISHAM/ECMM/FDLC Working Group on Nomenclature of Clinical Fungi to standardize terminology [de Hoog et al. 2024, J. Clin. Microbiol.]), while “*white-nose syndrome*” remains more widely used, particularly in North America.

We recognise the importance of accessibility in ensuring our paper reaches the broadest possible audience. To address this, we have explicitly clarified the relationship between “white-nose syndrome” and “white-nose disease” at their first mention in the main text, allowing readers familiar with either term to make the connection seamlessly and also provide references (lines 57-60). Throughout the rest of the manuscript, we have consistently used “*white-nose*” to maintain clarity and readability for all audiences. We hope this inclusive approach aligns with the Reviewer's expectations. However, if this remains a blocking point, we would be happy to consider alternative solutions.

Reference:

de Hoog S, Walsh TJ, Ahmed SA, Alastruey-Izquierdo A, Arendrup Maiken C, Borman A, Chen S, Chowdhary A, Colgrove RC, Cornely OA, Denning DW, Dufresne PJ, Filkins L, Gangneux J-P, Gené J, Groll AH, Guillot J, Haase G, Halliday C, Hawksworth DL, Hay R, Hoenigl M, Hubka V, Jagielski T, Kandemir H, Kidd SE, Kus JV, Kwon-Chung J, Lockhart SR, Meis JF, Mendoza L, Meyer W, Nguyen MH, Song Y, Sorrell TC, Stielow JB, Vilela R, Vitale RG, Wengenack NL, White PL, Ostrosky-Zeichner L, Zhang SX (2024) Nomenclature for human and animal fungal pathogens and diseases: a proposal for standardized terminology. *Journal of Clinical Microbiology* 62: e00937-00924. <http://dx.doi.org/10.1128/jcm.00937-24>

With respect to referring to the two *Pseudogymnoascus* as different species: Fungal taxonomy and naming conventions are needlessly complicated for ascomycete fungi. It might be more palatable for much of the mycology community to refer to these as two distinct “taxa” rather than calling them species. I am not familiar with naming conventions for this genus, but there may need to be more evidence of physiological or morphological differences to officially call these species. Personally, I think genetic evidence should be sufficient, but I am not sure if the fungal taxonomy community as a whole would agree.

> Thank you for raising this point. We understand that a formal species description is a significant component of fungal taxonomy and some of our co-authors have already described several new species. While *Nature* publishes original research across a wide range of disciplines, including taxonomy, a formal species description is typically considered a taxonomic paper, which is generally more suited to specialized journals such as *MycKeys*, *Persoonia*, *Mycologia*, or *Studies in Mycology*. For this reason, we have opted not to include a formal species description in the current manuscript. We note that when *Pseudogymnoascus (Geomyces) destructans* was first identified, the formal species description was published in a specialized journal (Gargas et al., 2009, *Mycotaxon*), following the initial discovery paper (Blehert et al., 2009, *Science*). We are following a similar approach, consistent with the current norms for this type of research.

Regarding terminology, we have adhered strictly to the conclusions of the most recent review on “*fungal species boundaries in the genomic era*” (Matute & Sepulveda, 2019, *Fungal Genetics and Biology*; cited). These rules align with earlier seminal papers, such as Taylor et al. (2000, *Fungal Genetics and Biology*; cited), which provide clear guidelines for species recognition in fungi based on genetic evidence. Specifically, we have substantiated our decision by demonstrating:

Reciprocal monophyly (Fig. 1B, 1E, S9)

Lower intra-specific differentiation than inter-specific differentiation (Fig. 1C, S11)

Evidence of recombination within clades (Fig. S14)

Absence of gene flow between divergent, yet sympatric and syntopic clades (Table S16)

Deep divergence dates between both species estimated to 750,000 generations (Table S16)

Given the comprehensive genetic evidence we present (in addition to ecological data on host-specialisation), using the term 'taxa' would not accurately reflect the clear species-level distinction we have demonstrated.

Blehert DS, Hicks AC, Behr M, Meteyer CU, Berlowski-Zier BM, Buckles EL, Coleman JT, Darling SR, Gargas A, Niver R, Okoniewski JC, Rudd RJ, Stone WB (2009) Bat white-nose syndrome: an emerging fungal pathogen? *Science* 323: 227. <http://dx.doi.org/10.1126/science.1163874>

Gargas A, Trest MT, Christiensen M, Volk TJ, Blehert DS (2009) *Geomyces destructans* sp. nov. associated with bat white-nose syndrome. *Mycotaxon* 108: 147-154. <http://dx.doi.org/10.5248/108.147>

Matute DR, Sepulveda VE (2019) Fungal species boundaries in the genomics era. *Fungal Genetics and Biology* 131: 103249. <http://dx.doi.org/10.1016/j.fgb.2019.103249>

Taylor JW, Jacobson DJ, Kroken S, Kasuga T, Geiser DM, Hibbett DS, Fisher MC (2000) Phylogenetic species recognition and species concepts in Fungi. *Fungal Genetics and Biology* 31: 21-32. <http://dx.doi.org/10.1006/fgbi.2000.1228>

Abstract

“...remains understudied” is an opinion. I suggest rewording this to “are often less well-studied.” While I agree that pathogen genetic variability tends to be less well-studied, there are numerous diseases for which host strain is well-understood to play an important role in infection ecology and disease outcomes.

>We agree and have revised the text as per the Reviewer's suggestion (line 24-25).

Main Text

“...is a single fungal species with a broad distribution range.” I agree that what we know about susceptibility and resistance to Pd infection in North America is based on the presence of the single clonally-expanding strain that occurs there. However, concerns over additional introductions of more strains of Pd has been a concern ever since the fungus was discovered in Eurasia. In other words, I don't think it is true that the existing literature assumes that Pd is not genetically diverse. This sentence might be better rewritten as “...However, importantly, this literature is based on a single clonally-expanded lineage of Pd and does not take into account the true genetic diversity of Pd that exists across the entire distribution of the species.”

>We have modified the sentences to include the fact that the literature is based on the fact that *Pd* is a single clonally-expanding isolate in North America (lines 65-67).

“...lack of ecologically relevant data have impeded interpretations...” I am not sure what the authors are specifically referring to here. “Interpretations” of what? Again, as written, this sounds more like an opinion. I suggest rephrasing this sentence to be more specific and use words like “limit” rather than “impeded.”

>We agree and have revised the text as per the Reviewer's suggestion (line 70).

“...to identify the source.” Suggest “...to identify the likely geographical source...”

>We agree and have revised the text as per the Reviewer's suggestion (line 74).

“...differences in ecological niche and host specialization...” State the ecological niche factors that were investigated in this study.

>Thank you for this suggestion. We have now revised the sentence to clearly distinguish the abiotic and biotic components of the ecological niche we investigated, specifying the factors examined within each category (lines 76-77)

Results and Discussion

Two sympatric *Pseudogymnoascus destructans* clades

Relatively speaking, 18 loci is a very small portion of the genome. How are these microsatellites dispersed on the genome? Is it possible with your data to state that the microsatellites are generally dispersed across the Pd genome and therefore would be likely to capture recombination events that might occur on a particular portion of the genome?

>Yes, the Reviewer is correct that the 18 loci are well spread across the genome and are likely to capture recombination events. Indeed, the 18 loci are spread across 14 out of the 18 contigs of Gd293. We now report the contig number for each locus based on the most contiguous genome, Gd293 (Table S6).

“...and, importantly, both have been demonstrated to cause the lesions diagnostic of the disease.” I suggest avoiding the word “importantly” in most scientific writing as it is subjective. Also, I assume that the citations you provide at the end of the sentence reference papers which describe the data that supports this statement (i.e., that isolates used in the current study originated from bats with confirmed lesions that are mentioned in those cited studies). I suggest stating more clearly that the strains you incorporated in the current project were isolated as parts of previous studies from bats with confirmed lesions. If this is not the case, then the data should be provided in the supplemental materials to demonstrate that there is histopathology to support both strains causing disease.

>We agree and have changed the text as per the Reviewer’s suggestion, and now clearly state that isolates from each clade were sampled from bats with confirmed lesions diagnostic of the disease (and cite the relevant references; line 86).

Figure 1: This is a beautiful figure panel! For panel B, what are the support values for the two main clades?

>Thank you; we are glad the Reviewer greatly appreciates this figure. We have now added information on the support values for the two main clades in the figure legend (and explained the method used in section 3.1 of Supplementary Material).

“...we have direct evidence...” These data need to be included in the supplemental or cited.

>Thank you for pointing this out. We have added these data in Table S3, in a new column named “Mdau_presence”. Also, we have obtained data from an additional 8 sites, hence the number of sites has increased from 164 to 172, though the percentage of sites with *M. daubentonii* and Pd-1 remains exactly the same (63%).

“The latter are therefore expected to come into contact with Pd-1 isolates....suggesting that Pd-1 is

incompatible with *M. daubentoniid*.” Your methodologies used culture-based methods which are far less sensitive than methods like metagenomics that might have been able to detect smaller amounts of fungus on individual bats. Instead, your results are based on what grew most easily in culture or which colonies were selected for isolation. Therefore, it might be overinterpretation to say that Pd-1 is “incompatible” with *M. daubentoniid*. There also could be behaviors exhibited by *M. daubentoniid* at your study sites that would cause Pd-2 to be enriched in that species even though it could still be susceptible to infection by Pd-1 under certain circumstances.

>We acknowledge that metagenomics may offer greater sensitivity than culture-based methods in detecting organisms, even at low concentrations. However, this advantage comes with a key limitation: metagenomic approaches do not provide direct evidence of whether the detected organisms were viable or actively growing on the host.

We also agree with the Reviewer that our results reflect what grew most readily in culture or what colonies were selected for isolation. However, as shown in Figure S7, both clades exhibit comparable colony expansion rates under the culture conditions used. Thus, there is no reason to expect significant differences in growth dynamics between the two clades that would bias our findings.

To ensure single-spore selection, we adhered to a standardised procedure: following spore germination, we examined plates under a microscope and randomly selected individual spores that were sufficiently isolated to allow for individual isolation (see Section 2.1 for further technical details). Given this method, we do not anticipate significant bias in our selection process.

Moreover, results from our Bayesian hierarchical model (Table S7, main text lines 125-128) indicate that *Myotis daubentoniid* had an almost certain probability (≈ 1.00) of harbouring Pd-2, further reinforcing our conclusions.

Finally, regarding the term “*incompatible*,” we acknowledge the Reviewer’s suggestion that it may not fully exclude the possibility of rare infections. In line with this feedback, we have revised the wording to improve clarity while ensuring our conclusions remain unaffected (line 132).

“...the colouration of agar medium by laboratory cultures can distinguish between the Pd clades.” This is an interesting observation, but I think it would be best to mention that among the strains you examined, there was a significant difference between the clades in coloration. There does appear to be at least one strain of Pd-2 that overlaps Pd-1 and so it may not be a definitive characteristic for distinguishing between the taxa. Furthermore, Khankhet et al. (2014) PLoS ONE provide data that some North American strains of Pd (which would be Pd-1) secrete brown pigment and color the medium similar to what is shown in Figure S5. I can also attest to many strains of Pd from North America causing a similar color change, with variation occurring even within the same isolate. Therefore, it might be best to be more cautious in implying that this feature can be used to distinguish the clades (even if it is a trend).

> Thank you for your thoughtful comments regarding our manuscript. We appreciate the opportunity to clarify our findings.

We acknowledge your concern about the reliability of medium colouration given the differences you and others have observed even within isolates from North America (i.e., Pd-1) or even within the same isolates. However, in the case of Khankhet et al. (2014), the study involved comparisons of phenotypic traits, scored visually by the naked eye, across three growth media (which were different from the

DYPA media we used) and temperatures hence measuring *Pds* response to differing abiotic conditions (n=16 isolates). In our study, we focused on the quantitative measure of phenotypic differences observed under controlled conditions where all isolates (N=79) were grown on the same culture medium (DYPA), at the same temperature (in the very same incubator), and for the same duration.

Our experimental setup allowed us to isolate the effects to the identity of the isolates themselves, which revealed significant differences in medium colouration based on clade identity. That being said, we wanted to make clear to the reader that some slight overlap is present and have added this to the manuscript (lines 138-141). This change does not alter our overall message.

“...have similar growth rates...” This is a very picky comment, but you might consider using the term “colony expansion rates” rather than “growth rates.” With the colony expansion method, it can be difficult to discern true differences in “growth” because radial hyphal growth is biased toward lateral (versus aerial or penetrative) expansion or hyphal density.

> We agree with the Reviewer and have revised the sentence accordingly (line 143).

“...host specialization appears to be the primary factor distinguishing their ecological niches, rather than abiotic factors.” Differential abundances on different hosts could certainly be due to abiotic factors. Caves are not homogenous environments and many bats select microhabitats with different microclimates within caves. It can be very difficult to tease apart biotic and abiotic components because many of them are intertwined. For example, a particular bat species’ behavior may subject the pathogen to different abiotic conditions than it experiences on a different host. I think some additional work with the two taxa are necessary to better tease apart whether host specialization is the main factor that causes niche differentiation, especially since Pd-2 is capable of growing on bat species that Pd-1 also colonizes.

> We agree with the Reviewer that "*It can be very difficult to tease apart biotic and abiotic components because many of them are intertwined.*" In response to this suggestion, we have revised the sentence to more effectively highlight the complexity of the host-pathogen interaction system and the multiple mechanisms that may be involved (lines 157-160). These refinements not only enhance clarity but also open up exciting avenues for future research to better understand resistance to the disease.

Genomic divergence between clades

Remove “the” in front of “664 single-copy BUSCO genes.”

“...other than clonality or geography prevent genetic exchange.” Replace “prevent” with “prevent detectable.”

> We agree with the Reviewer and have revised the sentences accordingly (line 167 and line 208).

Continental population structure and origin of the North American introduction
“However, as soon as bats emerge from hibernation....due to the elevated temperature of active bats upon and after emergence.” Remove “drastically” and “via grooming” from these sentences. I believe that grooming is only one of many factors that may remove the fungus from the surface of the bat. I would also point out that at least the American strains of the fungus survive periods of warming on bats (see Campbell et al. (2019), J Wildl Dis.) and your hypothesis that spread of the fungus would be

limited by these factors is at odds with the rapid spread of *Pd* in North America. Overall, I do not think this hypothesis is well supported by what we know about *Pd*. Are their known differences in bat movements in Eurasia that would reduce movement of *Pd* on the landscape as compared to North America (you mention quite the opposite...that populations are panmictic)? Congregation of bats at hibernating sites to mate in the fall and long-distance travel of male bats during this time is thought to be a key contributor of spread in North America. Do bats in Eurasia have a different breeding ecology? Perhaps they do not use caves in the spring or fall? Perhaps overall lower loads of *Pd* on Eurasian bat species means there is little fungus being carried on them when they move from site to site? Or perhaps established populations of *Pd* in hibernacula prevent newly introduced strains from surviving? I think there could be many factors at play to explain the population structure seen in *Pd*.

>We appreciate the Reviewer's insight and agree that this topic, though beyond the scope of the current paper, warrants further exploration. To maintain focus on the key aspects of our findings, we have condensed this section and included a reference to a relevant study for readers interested in a more detailed examination (lines 287-288). If space is constrained, this sentence can be omitted, as it does not contain critical information for the main message of the paper.

A specific region of Ukraine is mentioned as being the likely source of the introduction into the U.S. However, based on Figure 3A, it appears that far eastern Europe is not well represented in the dataset. I am not very familiar with this type of analysis, but wouldn't the lack of samples from other parts of Ukraine and far eastern Europe (e.g. Belarus, western Russia) make it more difficult to pinpoint such a specific location? Nonetheless, it is interesting that Drees et al. also mentioned that an isolate from Ukraine most closely matched North American strains. You may want to mention that paper as it provides additional support for an eastern European origin.

>The spatial Bayesian inference (SPASIBA) method is specifically designed to handle regions with sparse or no sampling data. The process begins by estimating covariance parameters of allele frequencies from georeferenced genetic data, which are then used to construct geographic maps of allele frequencies for each locus. In the final step, samples of unknown origin are assigned by maximizing the likelihood of their origin across the study area, including unsampled regions.

The study area considered in the SPASIBA analyses (shown in Fig. 4) extends further east than the samples collected in Ukraine and the Republic of Moldova, encompassing regions such as Belarus and areas further into Ukraine. These regions are identified with low likelihood as the origin (greyish colour in Fig. 4A – also reflected in the likelihood values in Table S5). In contrast, the region identified as the origin exhibits a likelihood 6×10^9 times greater than any other considered region (as detailed in the Fig. 4 legend), which strongly supports our conclusions.

Additionally, phylogenetic trees reconstructed from full genomes (e.g., Fig. 1B, Fig. S9) consistently show that Gd1111 [from North America] is closely related to Gd293 [from Podillia, Ukraine], with a branch length much shorter than any other pairs within *Pd*-1, indicating a very close genetic affinity between these two isolates. If the origin were further east, such as in Belarus or Western Russia, Gd1111 would show greater differentiation from Gd293.

Given these multiple lines of evidence, we are confident in our conclusions regarding the region of origin. We have added a sentence to explicitly mention this additional supporting evidence. It is noteworthy that Drees et al. (2017) identified a close relationship between North American *Pd* and an isolate from Ukraine, based on a very restricted sample size from Europe (n=5). Although this work

was already cited in our manuscript, we have now specifically referenced it in this context (line 333). However, it is important to note that Drees et al. were unable to definitively determine the origin due to “[...] low sample sizes, long branch lengths, and poor bootstrap support for the branches between members of the European clade preclude stronger conclusions about the region of Europe that gave rise to North American *P. destructans*.”

A mean of 3 and 3.6 isolates were obtained for bat and environmental swabs, respectively. Were these typically clonal (you mentioned in a few instances of both strains being isolated from a single bat)? I realize this could be gleaned by looking at the supplemental, but it might be good to mention it in the manuscript proper.

>This is an interesting question that, due to space constraints, we were unable to integrate in the manuscript. We considered including this information in the main manuscript but found no suitable section where it would not detract from the core message. However, the Reviewer is correct that the supplemental information includes all raw and processed data (Tables S1 to S5), which can be used to derive this number. Specifically, based on the full dataset, 65.08% of samples with two or more isolates contain at least two MLGs. If the Reviewer suggests an appropriate section to include this number, we would be happy to incorporate it.

Supplemental Section 3.2.5

You mention that Onygenales is a sister order to Helotiales (to which Pd belongs). However, these orders are not even within the same class of fungi.

>Thank you for spotting this. We exclusively used BUSCO for ‘Fungi’ throughout the manuscript hence we now removed this mention of ‘Onygenales’.

Section 3.4.4.1

“Confidentially” should be “confidently.”

>Thank you for spotting this typo that we have now corrected (this section is now part of the main manuscript, line 1060).

Section 3.5

Please describe how a single germinating spore was isolated to inoculate a new plate. How was this transferred? As an agar plug? I did not see this described in more detail in 2.1.

>Yes, that is correct, we used agar plugs to transfer single germinating spores. We have now added this more detailed information in methods section 2.1 (lines 560-564) .

Raw growth data is not provided nor is coloration data. Providing the data you used for your analyses is important for open data access.

> We apologise for the oversight and acknowledge the importance of including the data used in our analyses. Therefore, we have included two new tables (Tables S8 and S9) that present the data used to determine colony expansion rates and medium coloration.

It would be helpful to list the individual GenBank accession numbers for each strain in one of the supplemental tables. Currently, I only noticed the project number listed. I apologize if I overlooked where the accession numbers are listed.

>We now provide the GenBank accession number for each isolate in Supplementary Table S10.

Fig S4: The map for *Myotis myotis*/*M. blythii* appears to have two species ranged depicted on the same map, but it is not clear which range represents which species.

> We agree that it is clearer to display the ranges of both species separately. Therefore, we have now assigned different colours to the species' ranges when more than one species is represented.

*ADDITIONAL COMMENTS (sent in confidence to the editor from one of the referees):

Comment 1a- 1) While this paper provides much more thorough data to support that the Pd strain introduced to North America likely originated from Ukraine, other papers have previously pointed toward a similar conclusion (see Drees et al. 2017, mBio; Hoyt et al. 2021, Nature Reviews Microbiology).

>Short answer.

Both of these fine papers are cited in our work, but it is misleading to state that “*other papers have previously pointed toward a similar conclusion*”, questioning the novelty of our work. Hoyt et al. 2021 is a review, which makes reference to the Drees et al. 2017 study. Due to the limited sampling in Drees et al. (5 isolates), Hoyt et al. make an inference of the origin at the continent scale, not further: “*The P. destructans genotype distributed across North America is a member of the European clade*”.

>Detailed answer.

No other study has determined the origin of the North American *Pd* isolate beyond continental scale.

Drees et al. (2017) identified a close relationship between the North American *Pd* isolate and one from Ukraine, based on just five samples from Europe (whereas we have analysed 5,479 samples from 26 countries collected during the last 16 years), which clearly limited the ability of Drees et al. (2017) to draw any conclusive findings on the country of origin. We have plotted the approximate location of the five samples used in Drees et al. 2017 (left map below). These five samples are between 500 to 800 km distant from each other and no samples come from Northern or Southern Europe. This is in contrast with our sampling (Fig. 1) [map on the right below] where the majority of our 264 sites are within 5-20 km of each other, with sampling covering Northern and Southern Europe. Given the limited sampling in Drees et al. 2017, the authors did not have the possibility to identify the source population of the North American introduction and made no attempt to do so beyond the continental scale.

As the authors acknowledged themselves “*This phylogeny clearly shows the North American clade of *P. destructans* as a member of a greater European clade and distinct from the eastern and central Asian isolates. The tree also shows that North American *P. destructans* isolate is most closely related to a Ukrainian isolate (Gd44) in our sample set and least related to an isolate from France (Gd41). This is in contrast to a previous MLST study (28), which showed the opposite trend using many of the same isolates. However, low sample sizes, long branch lengths, and poor bootstrap support for the branches between members of the European clade preclude stronger conclusions about the region of Europe that gave rise to North American *P. destructans*”.*

An isolate from Ukraine was included in Drees et al. 2017, but due to the limited sampling, it was considered likely that further sample collection would reveal a closer match to the North American genotype: “[...] currently most closely related to an isolate collected from Ukraine (Fig. 1). However, sampling coverage in Eastern Europe and Central Asia is limited (Fig. 1), and a closer match to the North American isolate likely exists but is yet to be collected from this region.” The “Importance” of the work section in Drees et al. 2017 states: “We present new evidence supporting recent introduction of the fungus to North America from a diverse Eurasian population.” This study is cited in our work for this conclusion.

The focus of the Drees et al. 2017 study was on the origin of the introduction to North America considering a large spatial scale. There was very limited geographic sampling of genotypes to address this question (North America n=1, Europe n=5, Asia n=3) meaning that conclusions could at best be drawn at the continental scale (i.e. Europe versus Asia as the geographic source). Again, the conclusions of the study are clearly stated in the abstract: “Although the definitive source for introduction of the North American population has not been conclusively identified, our data support the origin of the North American invasion by *P. destructans* from Europe rather than Asia.”

While Drees et al.'s work was certainly an interesting step forward at the time, it remained inconclusive (regarding the origin), as acknowledged by the authors themselves. Hoyt et al. (2021), in their review, presented no new data compared to Drees et al. (2017), and, given that no new genetic data from Eurasia had been published up until that point, the paper provided no further insight. Despite this, Hoyt et al. speculated that “*The greater diversity of isolates in East*

*Asia in comparison to those in Europe, despite similar geographic distances, may suggest that the fungus first emerged in bats in Far-east Asia and then spread to Europe 21 (Fig. 1)” (ref 21 is Drees et al. 2017). Our analysis, based on 5,479 samples from 264 sites across 26 countries, incorporating multiple genetic markers alongside ecological and environmental data, directly contradicts this interpretation. The isolates from Asia represent a distinct species (*Pd-2*), and this species is also found in Europe (where genetic data were previously unavailable until the present study).*

Reviewer #2 aptly highlighted the importance of comprehensive sampling and the integration of ecological and environmental data in understanding fungal epidemics, stating, *'I found the manuscript very clear and the work convincing, not just for the discovery of a second species of Pseudogymnoascus, but also for demonstrating the critical role of ecological and environmental information from diverse isolates in understanding the origins and spread of fungal epidemics.'* We wholeheartedly agree with this perspective.

Comment 1b: More pressing questions that remain unanswered (and that would have more important conservation implications) that I would like to see answered in a publication that rises to the level of Nature are: a) what was the mechanism by which *Pd* reached North America from Ukraine? b) what is the likelihood of new strains being introduced to North America? c) what management efforts might reduce additional introductions?

>Short answer.

We now answer all three questions ‘a)’, ‘b)’ and ‘c)’.

>Detailed answer.

Determining the origin of *Pd* in North America is the first critical step in understanding how it arrived there, and hence for answering questions a) and c). So, in fact, we have answered the central question (the region of origin) that unlocks the answer to all subsequent pressing questions from the Reviewer. However, although it is a necessary step, answering one question does not automatically provide an easy solution to others (cf. the situation with SARS-CoV-2 origin/mechanisms of emergence), yet such publications are seminal in advancing our understanding of the pathogen's origin.

In our original manuscript, we did not address the mechanisms of introduction (a) as we did not consider this was the key question for this study. However, we have now tackled this issue and added substantial information regarding the mechanisms of introduction and even its likely timing (within a 15 years’ window). The convergence of genetic, historical, ecological, and experimental evidence offers substantial evidence supporting the hypothesis that *Pd* was introduced to North America via caving activities (i.e. contaminated gear). While absolute proof is nearly impossible to obtain (as is often the case with invasive/pathogens’ introduction, including those affecting humans and domestic animals), multiple independent lines of evidence provide a robust and coherent explanation for *Pd*’s transcontinental spread. Here is an overview of the evidence now also succinctly summarised in the paper, with the option to

elaborate on these points if deemed necessary, and additional details now provided as a supplementary text (Supplementary text S1).

1- Genetic evidence (Direct empirical evidence)

- Genetic analyses pinpoint the geographic origin of *Pd*, identifying Podillia as the most likely source (cf. manuscript).

2-Historical and behavioural evidence (Circumstantial evidence)

- Following the collapse of the USSR, previously restricted cave systems in Eastern Europe, including Ukraine, became accessible to international cavers, particularly from the United States (Scheltens, 1991).
- This period (late 1990s–early 2000s) perfectly aligns with the estimated time-frame of *Pd*'s introduction to North America.
- Records we have obtained confirm multiple caving expeditions between Podillia, the suspected region of origin in Ukraine, and North America during this time, including the Albany/Schoharie region in New York State (e.g. Steele, 2005) where the pathogen was introduced sometime before 2006 (Veilleux, 2008).

3-Ecological and environmental evidence (Indirect physical evidence)

- The suspected origin region contains the largest caves in Europe (the largest alone is >260 km), attracting large numbers of cavers coming from many countries. These caves are characterised by extensive systems of narrow corridors where cavers have to crawl to progress through narrow passages (two of our co-authors visited some of these caves). These conditions increase the likelihood of cavers accumulating significant amounts of sediment (and *Pd* spores) on their clothing and equipment.
- *Pd* is sensitive to UV light (including daylight), meaning it cannot survive when exposed to light for extended periods (Palmer et al. 2018). Caving gear provides a perfect, protected environment for the fungus to persist, as it is typically worn only in caves and is usually packed in a bag during travel, offering an ideal mode of transport for spores between caves.

4-Experimental evidence (Direct physical evidence)

- The direct detection of *Pd* on gear from *Pd*-positive sites confirms the fungus's ability to persist on equipment used in contaminated caves (Zhelyazkova et al. 2020).
- Laboratory studies demonstrate that *Pd* spores can survive on caving gear and clothing for several weeks at room temperature (Zhelyazkova et al. 2020), easily allowing viable spores from the pathogen to be transported from one continent to another.

Regarding "*b*) What is the likelihood of new strains being introduced to North America?", this is a difficult/impossible question to answer with certainty. While one could argue that, given the occurrence of past introductions (e.g., from Europe to North America, this study - between Eastern and Western North America, as described by Lorch et al. 2016), the likelihood of future trans-continental or long-distances introductions is high, quantifying this risk is inherently challenging. It is similar to asking, "What is the likelihood of new coronaviruses being passed from wildlife to humans?"

Evidence strongly indicates that Pd has been introduced through human-mediated transport, first from Europe to North America (as demonstrated in this study) and later from the Eastern to the Western United States (Lorch et al. 2016). A risk assessment evaluating Pd's potential introduction to Australia—including its likelihood of entering bat-populated caves and the susceptibility of native bat species—concluded that its introduction within the next 10 years is highly likely to almost certain (Holz et al., 2019). These findings provide evidence that Pd-1 and Pd-2 are highly likely to spread to currently unaffected regions, though the extent of this risk depends on the implementation of preventive measures (see paragraph below about policy).

Finally, with regard to "c) *What management efforts might reduce additional introductions?*" Given that we now provide an answer to 'a)', the solution is clear: avoiding trans-continental movement of caving equipment or washing caving equipment between transcontinental trips is essential, and even between long-distance trips within continents. This simple precaution can significantly reduce the number of viable spores that are transported and hence the risk of further introductions (Zhelyazkova et al., 2020). The answer to the 'b' question largely depends on the effective implementation of such simple measures. We have now included this in the main manuscript.

References cited:

- Holz P, Hufschmid J, Boardman WSJ, Cassey P, Firestone S, Lumsden LF, Prowse TAA, Reardon T, Stevenson M (2019) Does the fungus causing white-nose syndrome pose a significant risk to Australian bats? *Wildlife Research* 46: 657. <http://dx.doi.org/10.1071/WR18194>
- Lorch JM, Palmer JM, Lindner DL, Ballmann AE, George KG, Griffin K, Knowles S, Huckabee JR, Haman KH, Anderson CD, Becker PA, Buchanan JB, Foster JT, Blehert DS, McMahon K (2016) First detection of bat White-Nose Syndrome in Western North America. *mSphere* 1: e00148-00116. <http://dx.doi.org/10.1128/mSphere.00148-16>
- Palmer JM, Drees KP, Foster JT, Lindner DL (2018) Extreme sensitivity to ultraviolet light in the fungal pathogen causing white-nose syndrome of bats. *Nature communications* 9: 35. <http://dx.doi.org/10.1038/s41467-017-02441-z>
- Scheltens, J. (1991). Cowboys, cavers and cossacks. The first NSS field trip to the Soviet Union. *NSS News* July 1991, 198-204.
- Steele B (2005) An Interview with Chris Nicola (and the Priest Cave story). *NSS News* January 2005: 6-12.
- Veilleux JP (2008) Current status of white-nose syndrome in the Northeastern United States. *Bat Research News* 49: 15-17.
- Zhelyazkova VL, Hubancheva A, Radoslavov G, Toshkova NL, Puechmaille SJ (2020) Did you wash your caving suit? Cavers' role in the potential spread of *Pseudogymnoascus destructans*, the causative agent of White-Nose Disease. *International Journal of Speleology* 49: 145-154. <http://dx.doi.org/10.5038/1827-806X.49.2.2326>

Comment 2: 2) The authors mention that more introductions would be potentially detrimental to North America. I agree with this. However, this paper does not provide compelling evidence

that Eurasian strains possess traits that would worsen the impacts of the disease should additional strains be introduced to North America. For instance, there is no demonstration that some of the other strains found in Eurasian cause more severe disease, would be able to colonize American bat species currently considered to be resistant to infection, or that Eurasian strains are able to survive under environmental conditions that would allow them to spread to new regions in North America not occupied by the current strain of Pd that exists in North America. Heightened disease impacts if additional strains are introduced to North America are hypothetical, which may not meet Nature's bar for demonstrating the importance of this work.

>The Reviewer's agreement with the statement that '*more introductions would be potentially detrimental to North America*' indicates their recognition of the existence of a risk for North American bats. Indeed, our study provides evidence that the risk exists. However, in this remark, the Reviewer overlooked the fact that we do not only further characterise *Pd-1* genetic variability, we identify for the first time a second causative agent of white-nose disease, with a distinct host range. As such, our study is the first to highlight the risks associated with this second species for North American bats. This risk is supported by at least two documented instances of successful pathogen introductions: one from Europe to North America and another from Eastern to Western North America, both facilitated by human activity (as detailed in our response to Confidential Comment no. 1b above). Furthermore, the increasing global movement of people and the growing interest in caving activities amplify the argument that this risk should not be underestimated or dismissed (see also the new results regarding the mechanism of introduction; see response to Confidential Comment no. 1b above). As a result, all conservation implications related to this second species are novel. Risk assessment, a particularly challenging task often subject to disputed interpretations (as noted in the Reviewer's comment), is inherently about evaluating the likelihood of an event and does not necessitate certainty about its specific outcomes. This, in fact, addresses the Reviewer's main critique. Obtaining definitive evidence that the second *Pd* species would cause more harm than *Pd-1* in North America, or affect different species, is virtually impossible at this stage.

Our results, which demonstrate the presence of two fungal pathogens and a strong population structure within each species, have significant implications for conservation in Eurasia—implications that had not been previously considered, as the necessary data did not exist before our study. Specifically, while both fungal species are native to Eurasia, bats remain at risk of exposure to more virulent strains arising from long-distance inter- or intra-specific genetic exchanges, a process further exacerbated by human-mediated movements of the fungus (as evidenced with the Chytrid fungus for example). This finding, which is of importance for conservation, appears to have been overlooked by the Reviewer, possibly due to the greater focus on the devastating impact the disease has had on bat populations in North America. However, host-pathogen relationships can be highly dynamic and unstable, even within a species' native range, necessitating continuous monitoring to detect and respond to an ever-changing situation. This need is further amplified by climate change, which will impose additional stress on bat hosts, potentially altering disease dynamics and host susceptibility.

We develop the overall conservation implications of our findings in the response to Confidential Comment no. 3 below.

Comment 3: 3) Because of the two abovementioned points, I do not find the implications to conservation to be particularly novel since these are based on supposition. We have known that there is genetic diversity within *Pd* in Eurasia (and two mating types that can foster sexual recombination) for quite some time and that additional introductions of novel strains into North America could have change disease dynamics. This is also not a novel concept for fungal diseases in wildlife (although I acknowledge that pathogen genetics is often less studied initially than host or environmental factors). Specifically, multiple introductions have been postulated as factors driving disease emergence in chytridiomycosis and ophiidiomycosis.

>Short answer.

>We have provided evidence for the novelty of our findings and revised our manuscript to address all the three 'pressing questions' raised by the Reviewer (see response to Confidential Comment no. 1 above). Additionally, we present evidence of a risk not only to bat populations in North America but also in Eurasia (see response to Confidential Comment no. 2 above).

While we acknowledge the importance of conservation implications, we wish to emphasise that they are not the central conclusion of our study. Rather, our primary focus is the identification of two causative agents of white-nose disease in bats. The discovery of a second fungal pathogen capable of causing white-nose disease presents a risk to bat populations beyond the current range of the fungus, particularly in North America—a risk that appears to have been overlooked by the Reviewer. We believe it is essential to underscore this risk in the context of our study but we do not expand on it beyond a couple of sentences in the main text. While we do not claim that the concept of multiple introductions as a risk factor is novel to our paper, we believe that the failure to address this risk, particularly in relation to bats, would be a significant oversight. The introduction of new fungal species (*Pd-2*) or strains (from *Pd-1*), which could exacerbate the ongoing challenges posed by white-nose, warrants serious consideration and further discussion. We assert that acknowledging these risks is an essential part of the broader understanding of the disease dynamics and the need for proactive conservation strategies. In this regard, we maintain that the inclusion of these points enhances the relevance and urgency of our findings and underscores the potential implications for bat populations in North America, Europe and Asia.

Our discovery of a second causative agent for the white-nose disease with different host-species has more far-reaching implications. Our findings redefine the disease as a multi-pathogen disease rather than a single-species issue. This challenges existing assumptions regarding host specificity, transmission dynamics, and the evolutionary origins of both the pathogens and the disease. This discovery demands a critical re-evaluation of the extensive body of literature on the disease. Such a reassessment is especially vital given that *Pseudogymnoascus* ranks among the top 100 most cited fungal genera (Bhunjun et al., 2024, Stud. Mycol.). Furthermore, the identification of the source population from which the introduction originated represents a major advancement, creating exciting new opportunities to investigate the evolutionary trajectory of the single fungal clone that successfully colonised

North America, adapted to diverse environmental conditions, and infected multiple host species.

We have added responses to three pressing questions raised by the reviewers, further strengthening our findings, which, along with Referee #1's (an expert in fungal epidemiology and genetics/genomics) recognition that "*The broader findings are relevant to conservation and biosecurity*," underscores their novelty in these areas.

Reference: Bhunjun CS, et al. (2024) What are the 100 most cited fungal genera? *Studies in Mycology* 108: 1-411. <http://dx.doi.org/10.3114/sim.2024.108.01>

>Detailed answer.

Conservation is important but is not the central focus of our study.

While the conservation implications of our findings are important, they are not the central focus of our study but rather an ancillary aspect contributing to the broader discussion. To avoid any misinterpretation that conservation is our main conclusion, we have removed the final paragraph on "*Implications for Conservation and Policy*" and integrated its content into the relevant sections of the main text. The key finding of our study is the identification of two distinct fungal pathogens responsible for white-nose disease in bats, rather than a single causative agent. This discovery significantly reshapes our understanding of the disease, including its interaction with bat hosts, the evolution of resistance and tolerance, and broader ecological and epidemiological implications. Due to space limitations, we have only addressed a subset of these broader impacts in the manuscript. However, as the first Reviewer astutely noted, "*let your work and data speak for themselves.*"

From a practical perspective, conservation strategies must now account for multiple fungal pathogens, and the risk they pose, within and outside the range of each of the pathogens. Moreover, disease monitoring, diagnostics, and biosecurity protocols require revision, as current genetic tools (e.g., qPCR) cannot distinguish between the two pathogens. The discovery of a second causative agent not only necessitates updated diagnostic approaches but also provides new insights into fungal disease emergence, pathogen-host co-evolution, and the ecological factors shaping pathogen evolution and distribution.

Major advancements expected from our findings.

Pd is one of the few fungi known to cause severe mortality in immunocompetent mammals, making it a valuable model for studying mammalian immune responses to fungal infections. By combining two key findings—the discovery of a second *Pd* species and the identification of the North American introduction's origin—we overcome two major barriers that have hindered a comprehensive understanding of this host-pathogen system and the evolutionary trajectory of a single fungal clone during its successful invasion of North America.

The first major barrier was the absence of closely related pathogenic species with distinct host specialisation, which previously limited the application of comparative approaches. These approaches are essential for identifying the mechanisms underlying host tolerance and resistance, as they allow researchers to examine how different pathogens interact with various hosts under different ecological and evolutionary pressures. The discovery of a second *Pd* species with a different host range now enables such comparative analyses, offering a unique opportunity to dissect the genetic, immunological, and ecological factors that shape host susceptibility, pathogen virulence, and disease outcomes. Understanding why some bat species succumb to infection while others exhibit resilience will ultimately inform conservation strategies and disease management efforts.

A second major barrier to understanding the successful invasion of North America was the lack of a clearly identified source population in Europe, which is essential for making meaningful comparisons. Without pinpointing the exact origin, any genetic or phenotypic differences observed between North American and European *Pd* populations could be misinterpreted as evolutionary changes following introduction when they might instead reflect natural variation among different European populations. Identifying the true source region provides the most relevant baseline for assessing genetic, ecological, and evolutionary changes post-introduction, allowing researchers to determine whether the North American *Pd* population has undergone selection, adaptation, or other evolutionary shifts since its establishment.

Our identification of the source population in Ukraine represents a major breakthrough—achieved after 16 years of sampling across 264 sites in 26 countries (a “*herculean effort*”, as noted by Reviewer #1). This discovery not only resolves a long-standing gap in understanding *Pd*’s invasion history but also unlocks unique opportunities for comparative studies between North American isolates and their European ancestors. Such comparisons will enable an unprecedented examination of how the pathogen has evolved in its invasive range, providing critical insights into fungal pathogen evolution, host adaptation, and disease dynamics.

Moreover, identifying the source population offers a crucial reference for understanding pathogen-host interactions in *Pd*’s native range. By comparing the genetic diversity, virulence, and ecological dynamics of *Pd* from its source region to its introduced counterpart, researchers can infer how the fungus has adapted to new environments and hosts. These insights have direct implications for predicting future spread, assessing potential changes in virulence, and refining conservation and biosecurity strategies. More broadly, understanding the origins and evolutionary history of invasive fungal pathogens provides essential knowledge for mitigating future introductions and managing emerging fungal diseases in wildlife.

For these reasons, the major advances of our study will reinvigorate research on white-nose disease by establishing new comparative and evolutionary frameworks to investigate host-pathogen interactions, disease dynamics, and pathogen adaptation. More broadly, our findings will advance research on fungal diseases in mammals by providing a rare opportunity to study how fungal pathogens evolve, spread, and interact with their hosts in both native and introduced environments. By overcoming key barriers that have previously constrained the field, our work

paves the way for new insights into fungal disease emergence, mammalian immune responses, and the broader ecological and evolutionary processes that shape pathogen-host interactions.

Comment 4: 4) The materials and methods section of the main manuscript is obviously meant to be a summary and more detailed methods are in the supplemental. However, it seems like this section is perhaps overly brief.

>We have now moved materials and methods from Supplementary Information to the end of the manuscript in the “Materials and Methods” section. This section (15 pages) gives all the necessary details to replicate the study.

Referees' comments:

Referee #1 (Remarks to the Author):

I appreciate the careful attention that the authors have made in their revision. All the points that I raised have been substantially addressed either in the main text or in the SI.

Although not requested by me, I note the inclusion of historical data linking activity of cavers between the USA and the cave systems of Podillia in Ukraine spanning the period of WNS emergence. Although 'just so' stories can never prove disease transmission, this documented evidence at least details the commencement of activity following the dissolution of the Iron Curtain and further adds to the relevance of the studies implications on transnational biosecurity.

Referee #2 (Remarks to the Author):

The authors have satisfactorily addressed all my comments and concerns. Congratulations on putting together such an impressive body of work.

Referee #3 (Remarks to the Author):

The authors have done a great job responding to the reviewer comments. I think the addition of the sentences comparing the findings to the emergence of Bsal and the information about recreational caving in the vicinity of the region of Ukraine where the North American strain of Pd likely originated have really elevated the paper even more and make a very compelling case for the importance of the work. I do not have any additional comments. Well done!

>We thank again all three Reviewers for their support.